# RECYCLING MODEL UPDATES IN FEDERATED LEARNING: ARE GRADIENT SUBSPACES LOW-RANK?

**Sheikh Shams Azam, Seyyedali Hosseinalipour, Qiang Qiu, Christopher Brinton**
School of ECE, Purdue University
{azam1, hosseina, qqiu, cgb}@purdue.edu

## ABSTRACT

In this paper, we question the rationale behind propagating large numbers of parameters through a distributed system during federated learning. We start by examining the rank characteristics of the subspace spanned by gradients across epochs (i.e., the gradient-space) in centralized model training, and observe that this gradient-space often consists of a few leading principal components accounting for an overwhelming majority ($95 - 99\%$) of the explained variance. Motivated by this, we propose the "Look-back Gradient Multiplier" (LBGM) algorithm, which exploits this low-rank property to enable gradient recycling between model update rounds of federated learning, reducing transmissions of large parameters to single scalars for aggregation. We analytically characterize the convergence behavior of LBGM, revealing the nature of the trade-off between communication savings and model performance. Our subsequent experimental results demonstrate the improvement LBGM obtains in communication overhead compared to conventional federated learning on several datasets and deep learning models. Additionally, we show that LBGM is a general plug-and-play algorithm that can be used standalone or stacked on top of existing sparsification techniques for distributed model training.

## 1 INTRODUCTION

Federated Learning (FL) (Konečnỳ et al., 2016) has emerged as a popular distributed machine learning (ML) paradigm. By having each device conduct local model updates, FL substitutes raw data transmissions with model parameter transmissions, promoting data privacy (Shokri & Shmatikov, 2015; Azam et al., 2021) and communication savings (Wang et al., 2020a). At the same time, overparameterized neural networks (NN) are becoming ubiquitous in the ML models trained by FL, e.g., in computer vision (Liu et al., 2016; Huang et al., 2017) and natural language processing (Brown et al., 2020; Liu et al., 2019a). While NNs can have parameters in the range of a few million (VGG (Simonyan & Zisserman, 2014), ResNet (He et al., 2016)) to several billions (GPT-3 (Brown et al., 2020), Turing NLG (tur, 2020)), prior works have demonstrated that a majority of these parameters are often irrelevant (Frankle & Carbin, 2019; Liu et al., 2019b; Han et al., 2015; Li et al., 2017) in optimization and inference. This presents an opportunity to reduce communication overhead by transmitting lighter representations of the model, conventionally achieved through compression/sparsification techniques (Wang et al., 2018; Alistarh et al., 2017; Vogels et al., 2019).

In this work, we investigate the "overparameterization" of NN training, through the lens of rank characteristics of the subspace spanned by the gradients (i.e., the gradient-space) generated during stochastic gradient descent (SGD). We start with the fundamental question: *can we observe the effect of overparameterization in NN optimization directly through the principal components analysis (PCA) of the gradients generated during SGD-based training?* And if so: *can this property be used to reduce communication overhead in FL?* Our main hypothesis is that



**the subspaces spanned by gradients generated across SGD epochs are low-rank.**      **(H1)**



This leads us to propose a technique that instead of propagating a million/billion-dimensional vector (i.e., gradient) over the system in each iteration of FL only requires propagating a single scalar in the majority of the iterations. Our algorithm introduces a new class of techniques based on the concept of reusing/recycling device gradients over time. Our main contributions can be summarized as follows:

- We demonstrate the low-rank characteristics of the gradient-space by directly studying its principal components for several NN models trained on various real-world datasets. We show that

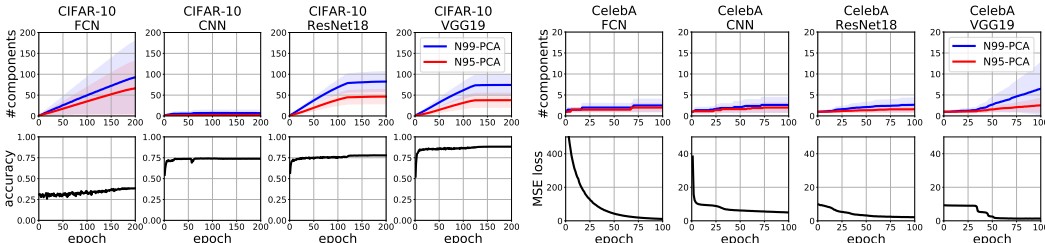

Figure 1: *PCA components progression.* The top row shows the number of components that account for 99% (N99-PCA in blue) and 95% (N95-PCA in red) explained variance of all the gradients generated during gradient descent epochs. The bottom row shows the performance of the model on the test data. The results are presented for: (i) CIFAR-10 classification (left 4 columns), and (ii) CelebA regression (right 4 columns).

- principal gradient directions (i.e., directions of principal components of the gradient-space) can be approximated in terms of actual gradients generated during the model training process.
- Our insights lead us to develop the "Look-back Gradient Multiplier" (LBGM) algorithm to significantly reduce the communication overhead in FL. LBGM recycles previously transmitted gradients to represent the newly-generated gradients at each device with a single scalar. We further analytically investigate the convergence characteristics of this algorithm.
- Our experiments show the communication savings obtained via LBGM in FL both as a standalone solution and a plug-and-play method used with other gradient compression techniques, e.g., top-K. We further reveal that LBGM can be extended to distributed training, e.g., LBGM with SignSGD (Bernstein et al., 2018) substantially reduces communication overhead in multi-GPU systems.

## 2 A GRADIENT-SPACE ODYSSEY

We first start by directly studying the principal component analysis (PCA) of the gradient-space of overparameterized NNs. Given a centralized ML training task (e.g., classification, regression, segmentation), we exploit principal component analysis (PCA) (Pearson, 1901) to answer the following: *how many principal components explain the 99% and 95% variance (termed N99-PCA and N95-PCA, respectively; together N-PCA) of all the gradients generated during model training?*

We start with 4 different NN architectures: (i) fully-connected neural network (FCN), (ii) convolutional neural network (CNN), (iii) ResNet18 (He et al., 2016), and (iv) VGG19 (Simonyan & Zisserman, 2014); trained on 2 datasets: CIFAR-10 (Krizhevsky et al., 2009) and CelebA (Liu et al., 2015), with classification and regression tasks, respectively. We then compute the N-PCA for each epoch by applying PCA on the set of gradients accumulated until that epoch (pseudo-code in Algorithm 2 in **Appendix D.1**). The results depicted in Fig. 1 agree with our hypothesis (**H1**): both N99-PCA and N95-PCA are significantly lower than that the total number of gradients generated during model training, e.g., in Fig. 1* the number of principal components (red and blue lines in top plots) are substantially lower (often as low as 10% of number of epochs, i.e., gradients generated) for both datasets. In **Appendix E.1**, we further find that (**H1**) holds in our experiments using several additional datasets: CIFAR-100 (Krizhevsky et al., 2009), MNIST (LeCun & Cortes, 2010), FMNIST (Xiao et al., 2017), CelebA (Liu et al., 2015), PascalVOC (Everingham et al., 2010), COCO (Lin et al., 2014); models: U-Net (Ronneberger et al., 2015), SVM (Cortes & Vapnik, 1995); and tasks: segmentation and regression. Note that (especially on CIFAR-10) variations in N-PCA across models are not necessarily related to the model performance (CNN performs almost as well as ResNet18 but has much lower N-PCA; Fig. 1 – columns 2 & 3) or complexity (CNN has more parameters than FCN but has lower N-PCA; Fig. 1 – columns 1 & 2). The fact that rank deficiency of the gradient-space is not a consequence of model complexity or performance suggests that the gradient-space of state-of-the-art large-scale ML models could be represented using a few principal gradient directions (PGD).

**N-PCA and FL.** In an "ideal" FL framework, if both the server and the workers/devices have the PGDs, then the newly generated gradients can be transmitted by sharing their projections on the PGDs, i.e., the PGD multipliers (PGM). PGMs and PGDs can together be used to reconstruct the device generated gradients at the server, dramatically reducing communication costs. However, this setting is impractical since: (i) it is infeasible to obtain the PGDs prior to the training, and (ii) PCA is computationally intensive. We thus look for an efficient online approximation of the PGDs.

---

*The addition of a learning rate scheduler (e.g., cosine annealing scheduler (Loshchilov & Hutter, 2016)) has an effect on the PCA of the gradient-space. Careful investigation of this phenomenon is left to future work.

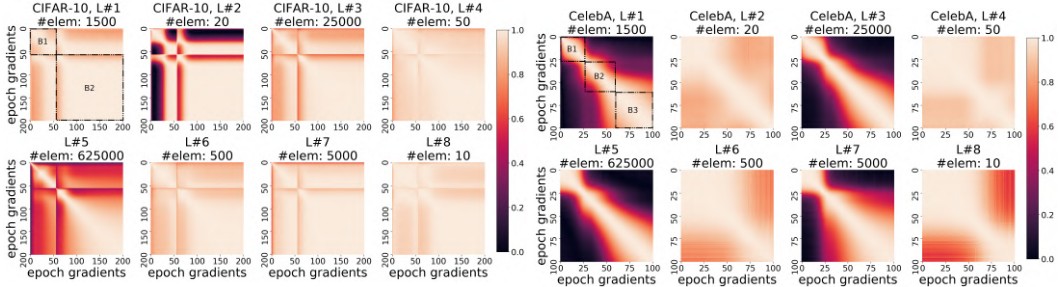

Figure 2: *Overlap of actual and principal gradients.* The heatmap shows the pairwise cosine similarity between actual (epoch) gradients and principal gradient directions (PCA gradients). Epoch gradients have a substantial overlap with one or more PCA gradients and consecutive epoch gradients show a gradual variation. This suggest that there may exist a high overlap between the gradients generated during the NN model training. The results are shown for a CNN classifier trained on CIFAR-10 (left 4 columns) and CelebA (right 4 columns) datasets. Each subplot is marked with #L, the layer number of the CNN and #elem, the number of elements in each layer.

Figure 3: *Similarity among consecutive gradients.* The cosine similarity of consecutive gradients reveals a gradual change in directions of gradients over epochs. Thus, the newly generated gradients can be represented in terms of the previously generated gradients with low approximation error. Reusing/recycling gradients can thus lead to significant communication savings during SGD-based federated optimization.

**Overlap of Actual Gradients and N-PCA.** To approximate PGDs, we exploit an observation made in Fig. 1 (and further in Appendix E.1): N-PCA mostly remains constant over time, suggesting that the gradients change gradually across SGD epochs. To further investigate this, in Fig. 2 (and further in Appendix E.2) we plot the cosine similarity between PGDs and actual gradients as a heatmap.[†] We observe that (i) the similarity of actual gradients to PGDs varies gradually over time, and (ii) actual gradients have a high cosine similarity with one or more PGDs. This leads to our second hypothesis:

**PGDs can be approximated using a subset of gradients generated across SGD epochs.** **(H2)**

**Look-back Gradients.** Our observations above suggest a significant overlap among consecutive gradients generated during SGD epochs. This is further verified in Fig. 3 (and in Appendix E.3[‡]), where we plot the pairwise cosine similarity of consecutive gradients generated during SGD epochs.[†] For example, consider the boxes marked B1, B2, and B3 in layer 1 (L#1 in Fig. 3). Gradients in each box can be used to approximate other gradients within the box. Also, interestingly, the number of such boxes that can be drawn is correlated with the corresponding number of PGDs. Based on (**H2**), we next propose our *Look-back Gradient Multiplier* (LBGM) algorithm that utilizes a subset of actual gradients, termed "look-back gradients", to reuse/recycle gradients transmitted in FL.

## 3   LOOK-BACK GRADIENT MULTIPLIER METHODOLOGY

Federated Learning (FL) considers a system of $K$ workers/devices indexed $1, ..., K$, as shown in Fig. 4. Each worker $k$ possesses a local dataset $\mathcal{D}_k$ with $n_k = |\mathcal{D}_k|$ datapoints. The goal of the system is to minimize the global loss function $F(\cdot)$ expressed through the following problem:

$$\min_{\boldsymbol{\theta} \in \mathbb{R}^M} F(\boldsymbol{\theta}) \triangleq \sum_{k=1}^{K} \omega_k F_k(\boldsymbol{\theta}), \tag{1}$$

---

[†]Refer to Algorithm 2 in Appendix D.1 for the detailed pseudocode.

[‡]2 of 24 experiments (Fig. 52&53 in Appendix E.3) show inconsistent gradient overlaps. However, our algorithm discussed in Sec. 3 still performs well on those datasets and models (see Fig. 60 in Appendix E.3).

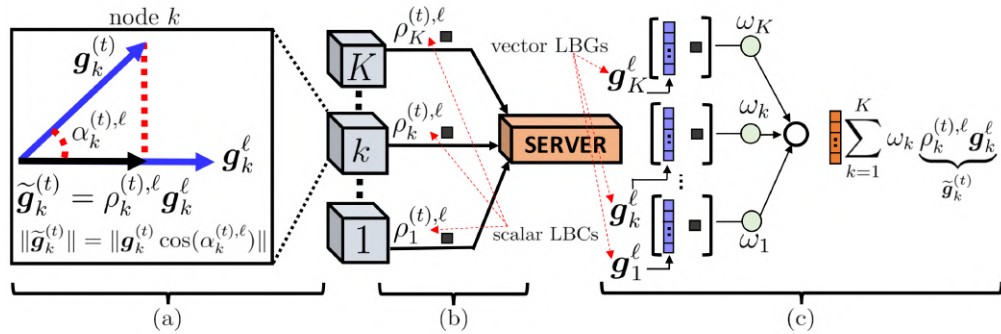

Figure 4: *Look-back gradient multiplier.* (a) The Look-back Coefficients (LBCs) are the projection of accumulated stochastic gradients at the workers on their Look-back Gradients (LBGs). (b) Scalar LBCs, i.e., the $\rho_k^{(t),\ell}$, are transmitted to the server. (c) LBG-based gradient approximations are reconstructed at the server.

where $M$ is the dimension of the model $\boldsymbol{\theta}$, $\omega_k = n_k/N$, $N = \sum_{k=1}^{K} n_k$, and $F_k(\boldsymbol{\theta}) = \sum_{d \in \mathcal{D}_k} f_k(\boldsymbol{\theta}; d)/n_k$ is the local loss at worker $k$, with $f_k(\boldsymbol{\theta}; d)$ denoting the loss function for data sample $d$ given parameter vector $\boldsymbol{\theta}$. FL tackles (1) via engaging the workers in local SGD model training on their own datasets. The local models are periodically transferred to and aggregated at the main server after $\tau$ local updates, forming a global model that is used to synchronize the workers before starting the next round of local model training.

At the start of round $t$, each model parameter $\boldsymbol{\theta}_k^{(t,0)}$ is initialized with the global model $\boldsymbol{\theta}^{(t)}$. Thereafter, worker $k$ updates its parameters $\boldsymbol{\theta}_k^{(t,b)}$ as: $\boldsymbol{\theta}_k^{(t,b+1)} \leftarrow \boldsymbol{\theta}_k^{(t,b)} - \eta \boldsymbol{g}_k(\boldsymbol{\theta}_k^{(t,b)})$, where $\boldsymbol{g}_k(\boldsymbol{\theta}_k^{(t,b)})$ is the stochastic gradient at local step $b$, and $\eta$ is the step size. During a vanilla FL aggregation, the global model parameters are updated as $\boldsymbol{\theta}^{(t+1)} \leftarrow \boldsymbol{\theta}^{(t)} - \eta \sum_{k=1}^{K} \omega_k \boldsymbol{g}_k^{(t)}$, where $\boldsymbol{g}_k^{(t)} = \sum_{b=0}^{\tau-1} \boldsymbol{g}_k(\boldsymbol{\theta}_k^{(t,b)})$ is the accumulated stochastic gradient (ASG) at worker $k$. More generally, this aggregation may be conducted over a subset of workers at time $t$. We define $\nabla F_k(\boldsymbol{\theta}_k^{(t)}) = \sum_{b=0}^{\tau-1} \nabla F_k(\boldsymbol{\theta}_k^{(t,b)})$ and $\boldsymbol{d}_k^{(t)} = \boldsymbol{g}_k^{(t)}/\tau$ as the corresponding accumulated true gradient and normalized ASG, respectively.

**Indexing and Notations.** In *superscripts with parenthesis* expressed as tuples, the first element denotes the global aggregation round while the second element denotes the local update round, e.g., $\boldsymbol{g}_k^{(t,b)}$ is the gradient at worker $k$ at global aggregation round $t$ at local update $b$. *Superscripts without parenthesis* denote the index for look-back gradients, e.g., $\ell$ in $\rho_k^{(t),\ell}$ defined below.

LBGM **Algorithm.** LBGM (see Fig. 4) consists of three main steps: (i) workers initialize and propagate their look-back gradients (LBGs) to the server; (ii) workers estimate their look-back coefficients (LBCs), i.e., the scalar projection of subsequent ASGs on their LBGs, and the look-back phase (LBP), i.e., the angle between the ASG and the LBG; and (iii) workers update their LBGs and propagate them to the server if the LBP passes a threshold, otherwise they only transmit the scalar LBC. Thus, LBGM propagates only a subset of actual gradients generated at the devices to the server. The intermediate global aggregation steps between two LBG propagation rounds only involve transfer of a *single scalar*, i.e., the LBC, from each worker, instead of the entire ASG vector.

In LBGM, the local model training is conducted in the same way as vanilla FL, while model aggregations at the server are conducted via the following rule:

$$\boldsymbol{\theta}^{(t+1)} = \boldsymbol{\theta}^{(t)} - \eta \sum_{k=1}^{K} \omega_k \widetilde{\boldsymbol{g}}_k^{(t)}, \tag{2}$$

where $\widetilde{\boldsymbol{g}}_k^{(t)}$ is the approximation of worker $k$'s accumulated stochastic gradient, given by Definition 1.

**Definition 1.** *(Gradient Approximation in* LBGM*) Given the ASG $\boldsymbol{g}_k^{(t)}$, and the LBG $\boldsymbol{g}_k^\ell$, the gradient approximation $\widetilde{\boldsymbol{g}}_k^{(t)}$ recovered by the server is given by:*

$$\widetilde{\boldsymbol{g}}_k^{(t)} = \rho_k^{(t),\ell} \boldsymbol{g}_k^\ell \ \text{for} \ \left\| \rho_k^{(t),\ell} \boldsymbol{d}_k^\ell \right\| = \left\| \boldsymbol{d}_k^{(t)} \cos(\alpha_k^{(t),\ell}) \right\|, \tag{D1}$$

*where LBC $\rho_k^{(t),\ell} = \langle \boldsymbol{g}_k^{(t)}, \boldsymbol{g}_k^\ell \rangle / \|\boldsymbol{g}_k^\ell\|^2$ is the projection of the accumulated gradient $\boldsymbol{g}_k^{(t)}$ on the LBG $\boldsymbol{g}_k^\ell$, and LBP $\alpha_k^{(t),\ell}$ denotes the angle between $\boldsymbol{g}_k^{(t)}$ and $\boldsymbol{g}_k^\ell$ (see Fig. 4(a)).*

---

**Algorithm 1** `LBGM`: Look-back Gradient Multiplier

---

**Notation:**

$\boldsymbol{\theta}^{(t)}$: global model parameter at global aggregation round $t$.

$\boldsymbol{\theta}_k^{(t,b)}$: model parameter at worker $k$, at global aggregation round $t$ and local update $b$.

$\boldsymbol{g}_k^{(t)}$: accumulated gradient at worker $k$ at global aggregation round $t$.

$\boldsymbol{g}_k^{\ell}$: last full gradient transmitted to server, termed look-back gradient (LBG).

$\alpha_k^{(t),\ell}$: phase between the accumulated gradient $\boldsymbol{g}_k^{(t)}$ and LBG $\boldsymbol{g}_k^{\ell}$, termed look-back phase (LBP).

**Training at worker $k$:**

1: Update local parameters: $\boldsymbol{\theta}_k^{(t,0)} \leftarrow \boldsymbol{\theta}^{(t)}$, and initialize gradient accumulator: $\boldsymbol{g}_k^{(t)} \leftarrow \boldsymbol{0}$.

2: **for** $b = 0$ to $(\tau\text{-}1)$ **do**

3:     Sample a minibatch of datapoints $\mathcal{B}_k$ from $\mathcal{D}_k$ and compute $\boldsymbol{g}_k^{(t,b)} = \sum_{d \in \mathcal{B}_k} \nabla f_k(\boldsymbol{\theta}_k^{(t,b)}; d)/|\mathcal{B}_k|$.

4:     Update local parameters: $\boldsymbol{\theta}_k^{(t,b+1)} \leftarrow \boldsymbol{\theta}_k^{(t,b)} - \eta \cdot \boldsymbol{g}_k^{(t,b)}$, and accumulate gradient: $\boldsymbol{g}_k^{(t)} \leftarrow \boldsymbol{g}_k^{(t)} + \boldsymbol{g}_k^{(t,b)}$.

5: **end for**

6: Calculate the LBP error: $\sin^2(\alpha_k^{(t),\ell}) = 1 - \left( \langle \boldsymbol{g}_k^{(t)}, \boldsymbol{g}_k^{\ell} \rangle / (\|\boldsymbol{g}_k^{(t)}\| \times \|\boldsymbol{g}_k^{\ell}\|) \right)^2$

7: **if** $\sin^2(\alpha_k^{(t),\ell}) \leq \delta_k^{\text{threshold}}$ **then**                              ▷ checking the LBP error

8:     Send scalar LBC to the server: $\boldsymbol{\mu}_k^{(t)} \leftarrow \rho_k^{(t),\ell} = \langle \boldsymbol{g}_k^{(t)}, \boldsymbol{g}_k^{\ell} \rangle / \|\boldsymbol{g}_k^{\ell}\|^2$.

9: **else**                                                                        ▷ updating the LBG

10:     Send actual gradient to the server: $\boldsymbol{\mu}_k^{(t)} \leftarrow \boldsymbol{g}_k^{(t)}$.

11:     Update worker-copy of LBG: $\boldsymbol{g}_k^{\ell} \leftarrow \boldsymbol{g}_k^{(t)}$.

12: **end if**

**Global update at the aggregation server:**

13: Initialize global parameter $\boldsymbol{\theta}^{(0)}$ and broadcast it across workers.

14: **for** $t = 0$ to $(T-1)$ **do**

15:     Receive updates from workers $\{\boldsymbol{\mu}_k^{(t)}\}_{k=1}^K$.

16:     Update global parameters: $\boldsymbol{\theta}^{(t+1)} \leftarrow \boldsymbol{\theta}^{(t)} - \eta \sum_{k=1}^K \omega_k \left[ s_k \cdot \boldsymbol{\mu}_k^{(t)} \cdot \boldsymbol{g}_k^{\ell} + (1-s_k) \cdot \boldsymbol{\mu}_k^{(t)} \right]$,

    where $s_k$ is an indicator function given by, $s_k = \begin{cases} 1, & \text{if } \boldsymbol{\mu}_k^{(t)} \text{ is a scalar} \\ 0, & \text{otherwise, i.e., if } \boldsymbol{\mu}_k^{(t)} \text{ is a vector} \end{cases}$.

17:     Update server-copy of LBGs: $\boldsymbol{g}_k^{\ell} \leftarrow (1-s_k)\boldsymbol{\mu}_k^{(t)} + (s_k)\boldsymbol{g}_k^{\ell}, \; \forall k$.

18: **end for**

---

`LBGM` initializes the LBGs $\boldsymbol{g}_k^{\ell}$ with the first actual gradients propagated by the devices at $t = 1$. For the subsequent aggregations, each worker $k$ shares only its LBC $\rho_k^{(t),\ell}$ with the server if the value of LBP error $\sin^2(\alpha_k^{(t),\ell})$ is below a threshold, i.e., $\sin^2(\alpha_k^{(t),\ell}) \leq \delta_k^{\text{threshold}}$, where $\delta_k^{\text{threshold}} \in [0, 1]$ is a tunable parameter; otherwise it updates the LBG via transmitting the entire gradient to the server.

The details of `LBGM` are given in Algorithm 1. We next conduct convergence analysis for `LBGM`.

**Assumptions:** It is presumed that the local loss functions are bounded below: $\min_{\boldsymbol{\theta} \in \mathbb{R}^M} F_k(\boldsymbol{\theta}) > -\infty, \forall k$. Let $\| \cdot \|$ denote the 2-norm. Our analysis uses the following standard assumptions (Wang et al., 2020a; Friedlander & Schmidt, 2012; Hosseinalipour et al., 2020; Li et al.; Stich, 2019):

1. *Smoothness of Local Loss Functions:* Local loss function $F_k : \mathbb{R}^M \to \mathbb{R}, \forall k$, is $\beta$-smooth:

$$\left\| \nabla F_k(\boldsymbol{\theta}_x) - \nabla F_k(\boldsymbol{\theta}_y) \right\| \leq \beta \left\| \boldsymbol{\theta}_x - \boldsymbol{\theta}_y \right\|, \; \forall \boldsymbol{\theta}_x, \boldsymbol{\theta}_y \in \mathbb{R}^M. \tag{A1}$$

2. *SGD Characteristics:* Local gradients $\boldsymbol{g}_k(\boldsymbol{\theta}), \forall k$, estimated by SGD are unbiased estimators of the true gradients $\nabla F_k(\boldsymbol{\theta})$, and have a bounded variance $\sigma^2 \geq 0$; mathematically:

$$\mathbb{E}\left[ \boldsymbol{g}_k(\boldsymbol{\theta}) \right] = \nabla F_k(\boldsymbol{\theta}), \text{ and } \mathbb{E}\left[ \|\boldsymbol{g}_k(\boldsymbol{\theta}) - \nabla F_k(\boldsymbol{\theta})\|^2 \right] \leq \sigma^2, \; \forall \boldsymbol{\theta} \in \mathbb{R}^M. \tag{A2}$$

3. *Bounded Dissimilarity of Local Loss Functions:* For any realization of weights $\{\omega_k \geq 0\}_{k=1}^K$, where $\sum_{k=1}^K \omega_k = 1$, there exist non-negative constants $\Upsilon^2 \geq 1$ and $\Gamma^2 \geq 0$ such that

$$\sum_{k=1}^K \omega_k \|\nabla F_k(\boldsymbol{\theta})\|^2 \leq \Upsilon^2 \left\| \sum_{k=1}^K \omega_k \nabla F_k(\boldsymbol{\theta}) \right\|^2 + \Gamma^2, \; \forall \boldsymbol{\theta} \in \mathbb{R}^M. \tag{A3}$$

**Theorem 1.** *(General Convergence Characteristic of* LBGM*) Assume* **A1**, **A2**, **A3***, and that* $\eta\beta \leq \min\left\{1/(2\tau), 1/\left(\tau\sqrt{2(1+4\Upsilon^2)}\right)\right\}$. *If the threshold value in step 7 of Algorithm 1 satisfies the condition* $\delta_k^{\mathsf{threshold}} \leq \Delta^2/\|\boldsymbol{d}_k^{(t)}\|^2$, $\forall k$ *, where* $\Delta^2 \geq 0$ *is a constant, then after* $T$ *rounds of global aggregations, the performance of* LBGM *is characterized by the following upper bound:*

$$\frac{1}{T}\sum_{t=0}^{T-1}\mathbb{E}\left[\left\|\nabla F(\boldsymbol{\theta}^{(t)})\right\|^2\right] \leq \frac{8\left[F(\boldsymbol{\theta}^{(0)}) - F^\star\right]}{\eta\tau T} + 16\Delta^2 + 8\eta\beta\sigma^2 + 5\eta^2\beta^2\sigma^2(\tau-1) + 20\eta^2\beta^2\Gamma^2\tau(\tau-1). \quad (3)$$

*Proof.* The proof is provided in Appendix A. ∎

The condition on $\delta_k^{\mathsf{threshold}}$ and LBP error $\sin^2(\alpha_k^{(t),\ell})$ in the above theorem implies that to have a fixed bound in (3), for a fixed $\Delta^2$, a larger gradient norm $\|\boldsymbol{d}_k^{(t)}\|^2$ is associated with a tighter condition on the LBP error $\sin^2(\alpha_k^{(t),\ell})$ and $\delta_k^{\mathsf{threshold}}$. This is intuitive because a larger gradient norm corresponds to a larger estimation error when the gradient is recovered at the server for a given LBP (see Fig. 4). In practice, since the gradient norm $\|\boldsymbol{d}_k^{(t)}\|^2$ does not grow to infinity during model training, the condition on LBP error in Theorem 1, i.e., $\sin^2(\alpha_k^{(t),\ell}) \leq \Delta^2/\|\boldsymbol{d}_k^{(t)}\|^2$, can always be satisfied for any $\Delta^2 \geq 0$, since transmitting actual gradients of worker $k$ makes $\alpha_k^{(t),\ell} = 0$. Given the general convergence behavior in Theorem 1, we next obtain a specific choice of step size and an upper bound on $\Delta^2$ for which LBGM approaches a stationary point of the global loss function (1).

**Corollary 1.** *(Convergence of* LBGM *to a Stationary Point) Assuming the conditions of Theorem 1, if* $\Delta^2 \leq \eta$, *where* $\eta = 1/\sqrt{\tau T}$, *then* LBGM *converges to a stationary point of the global loss function, with the convergence bound characterized below:*

$$\frac{1}{T}\sum_{t=0}^{T-1}\mathbb{E}\left[\left\|\nabla F(\boldsymbol{\theta}^{(t)})\right\|^2\right] \leq \mathcal{O}\left(\frac{1}{\sqrt{\tau T}}\right) + \mathcal{O}\left(\frac{\sigma^2}{\sqrt{\tau T}}\right) + \mathcal{O}\left(\frac{1}{\sqrt{\tau T}}\right) + \mathcal{O}\left(\frac{\sigma^2(\tau-1)}{\tau T}\right) + \mathcal{O}\left(\frac{(\tau-1)\Gamma^2}{T}\right). \quad (4)$$

*Proof.* The proof is provided in Appendix B. ∎

Considering the definition of $\Delta^2$ in Theorem 1 and the condition imposed on it in Corollary 1, the LBP error (i.e., $\sin^2(\alpha_k^{(t),\ell})$) should satisfy $\|\boldsymbol{d}_k^{(t)}\|^2\sin^2(\alpha_k^{(t),\ell}) \leq \eta = 1/\sqrt{\tau T}$ for LBGM to reach a stationary point of the global loss. Since $\|\boldsymbol{d}_k^{(t)}\|^2$ is bounded during the model training, this condition on the LBP error can always be satisfied by tuning the frequency of the full (actual) gradient transmission, i.e., updating the LBG as in lines 10&11 of Algorithm 1. Specifically, since the correlation across consecutive gradients is high and drops as the gradients are sampled from distant epochs (see Fig. 3), a low value of the LBP error can be obtained via more frequent LBG transmissions to the server.

**Main Takeaways from Theorem 1, Corollary 1, and Algorithm 1:**

1. *Recovering Vanilla-FL Bound:* In (3), if the LBGs are always propagated by all the devices, we have $\alpha_k^{(t),\ell} = 0$, $\forall k$, and thus $\Delta^2 = 0$ satisfies the condition on the LBP error. Then, (3) recovers the bound for vanilla FL (Wang et al., 2020a; Stich, 2019; Wang & Joshi, 2018).

2. *Recovering Centralized SGD Bound:* In (3), if the LBGs are always propagated by all the devices, i.e., $\Delta^2 = 0$, the local dataset sizes are equal, i.e., $w_k = 1/K, \forall k$, and $\tau = 1$, then (3) recovers the bound for centralized SGD, e.g., see Friedlander & Schmidt (2012).

3. *Unifying Algorithm 1 and Theorem 1:* The value of $\Delta^2$ in (3) is determined by the value of the LBP error $\sin^2(\alpha_k^{(t),\ell})$, which is also reflected in step 7 of Algorithm 1. This suggests that the performance improves when the allowable threshold on $\sin^2(\alpha_k^{(t),\ell})$ is decreased (i.e., smaller $\Delta^2$), which is the motivation behind introducing the tunable threshold $\delta_k^{\mathsf{threshold}}$ in our algorithm.

4. *Effect of LPB Error on Convergence:* As the value of $\sin^2(\alpha_k^{(t),\ell})$ increases, the term in (3) containing $\Delta^2$ will start diverging (it can become the same order as the gradient $\|\boldsymbol{d}_k^{(t)}\|^2$). The condition in Corollary 1 on $\Delta^2$ avoids this scenario, achieving convergence to a stationary point.

5. *Performance vs. Communication Overhead Trade-off:* Considering step 7 of Algorithm 1, increasing the tolerable threshold on the LBP error increases the chance of transmitting a scalar (i.e., LBC) instead of the entire gradient to the server from each worker, leading to communication savings. However, as seen in Theorem 1 and the condition on $\Delta^2$ in Corollary 1, the threshold on the LBP error cannot be increased arbitrarily since the LBGM may show diverging behavior.

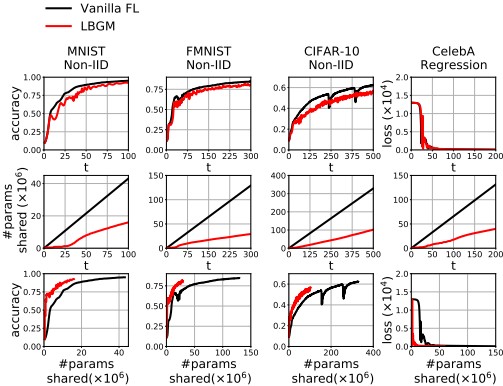

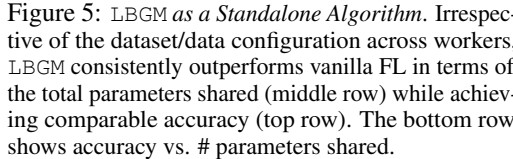

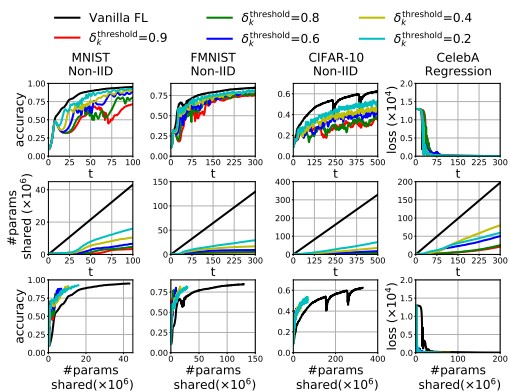

Figure 5: LBGM *as a Standalone Algorithm*. Irrespective of the dataset/data configuration across workers, LBGM consistently outperforms vanilla FL in terms of the total parameters shared (middle row) while achieving comparable accuracy (top row). The bottom row shows accuracy vs. # parameters shared.

Figure 6: *Effect of $\delta_k^{\mathsf{threshold}}$ on* LBGM. As $\delta_k^{\mathsf{threshold}}$ decreases, the training may become unstable. For larger values of $\delta_k^{\mathsf{threshold}}$, LBGM achieves communication benefits (middle row) while maintaining a performance identical to vanilla FL (top row). The bottom row shows accuracy vs. # parameters shared.

# 4 EXPERIMENTS

**Model Settings.** We run experiments on several NN models and datasets. Specifically, we consider: **S1:** CNN on FMNIST, MNIST, CelebA, and CIFAR-10, **S2:** FCN on FMNIST and MNIST, and **S3:** ResNet18 on FMNIST, MNIST, CelebA, CIFAR-10 and CIFAR-100 for both independently and identically distributed (iid) and non-iid data distributions. We present results of **S1** (on non-iid data) in this section and defer the rest (including **S1** on iid data and U-Net on PascalVOC) to Appendix F.

**Properties Studied.** We specifically focus on four properties of LBGM: **P1:** the benefits of gradient recycling by LBGM *as a standalone algorithm*, **P2:** the *effect of $\delta_k^{\mathsf{threshold}}$* on LBGM from Theorem 1, **P3:** practical capabilities of LBGM *as a general plug-and-play algorithm* that can be stacked on top of other gradient compression techniques in FL training, and finally **P4:** generalizability of LBGM *to distributed learning* frameworks, e.g., multi-processor or multi-GPU ML systems.

**Baselines.** For **P1** and **P2**, we compare LBGM with vanilla FL. For **P3**, we stack LBGM on top of top-K and ATOMO (Wang et al., 2018), two state-of-the-art techniques for sparsification and low-rank approximation-based gradient compression, respectively. For **P4**, we stack LBGM on top of SignSGD (Bernstein et al., 2018), a state-of-the-art method in gradient compression for distributed learning.

**Implementation Details.** We consider an FL system consisting of 100 workers. We consider both the iid and non-iid data distributions among the workers. Under the iid setting, each worker has training data from all the labels, while under the non-iid setting each worker has training data only from a subset of all labels (e.g., from 3 of 10 classes in MNIST/FMNIST). The workers train with mini-batch sizes ranging from 128 to 512 based on the choice of dataset. We implement LBGM with uniform $\delta_k^{\mathsf{threshold}}$ across workers. We also use error feedback (Karimireddy et al., 2019) as standard only if top-K sparsification is used in the training. The FL system is simulated using PyTorch (Paszke et al., 2019) and PySyft (Ryffel et al., 2018) and trained on a 48GB Tesla-P100 GPU with 128GB RAM. All of our code and hyperparameters are available at https://github.com/shams-sam/FedOptim. Appendix C.2 details the process of hyperparameter selection for the baselines.

**Complexity.** Compared to other gradient compression techniques, the processing overhead introduced by LBGM is negligible. Considering Algorithm 1, the calculation of LBCs and LBP errors involves inner products and division of scalars, while reconstruction of LBG-based gradient approximations at the server is no more expensive than the global aggregation step: since the global aggregation step requires averaging of local model parameters, it can be combined with gradient reconstruction. This also holds for LBGM as a plug-and-play algorithm, as top-K and ATOMO (Wang et al., 2018) introduce considerable computation overhead. In particular, LBGM has $\mathcal{O}(M)$ complexity, where $M$ is the dimension of the NN parameter, which is inexpensive to plug on top of top-K ($\mathcal{O}(M \log M)$), ATOMO (Wang et al., 2018) ($\mathcal{O}(M^2)$), and SignSGD (Bernstein et al., 2018) ($\mathcal{O}(M)$) methods. The corresponding space complexity of LBGM for the server and devices is discussed in Appendix C.1.

LBGM **as a Standalone Algorithm.** We first evaluate the effect of gradient recycling by LBGM in FL. Fig. 5 depicts the accuracy/loss values (top row) and total floating point parameters transferred

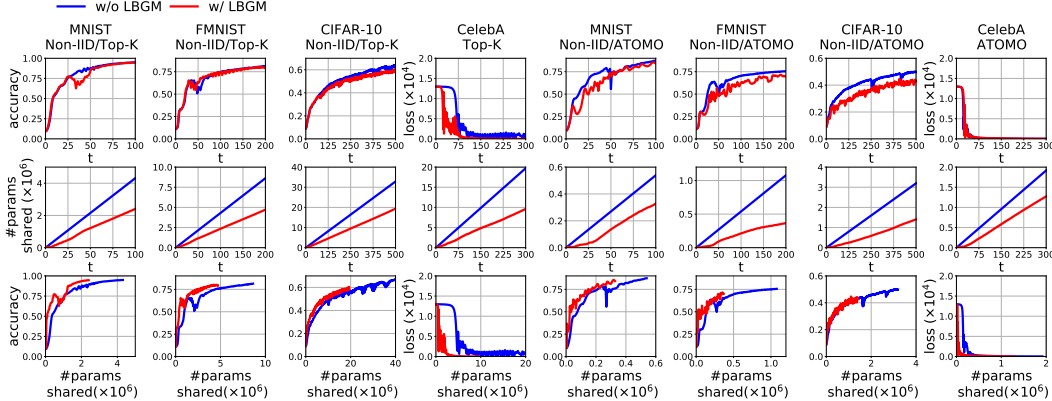

Figure 7: `LBGM` *as a Plug-and-Play Algorithm.* `LBGM` obtains substantial communication benefits when implemented on top of existing gradient compression techniques by exploiting the rank-characteristics of the gradient-space. Top-K and ATOMO are known to achieve state-of-the-art performance of their respective domains of sparsification and low-rank approximation respectively.

over the system (middle row) across training epochs for $\delta_k^{\text{threshold}} = 0.2$, $\forall k$ on diferent datasets. The parameters transferred indicates the communication overhead, leading to a corresponding performance vs. efficiency tradeoff (bottom row). For each dataset, we observe that `LBGM` reduces communication overhead on the order of $10^7$ floating point parameters per worker. Similar results on other datasets and NN models are deferred to Appendix F.1. We also consider `LBGM` under device sampling (see Algorithm 3 in Appendix D.2) and present the results in Appendix F.5, which are qualitatively similar.

**Effect of $\delta_k^{\text{threshold}}$ on Accuracy vs. Communication Savings.** In Fig. 5, the drops in accuracy for the corresponding communication savings are small except for on CIFAR-10. The $14\%$ reduction in accuracy here is a result of the hyperparameter setting $\delta_k^{\text{threshold}} = 0.2$. As noted in takeaway 3 in Sec. 3, a decrease in the allowable threshold on the LBP error improves the accuracy; the effect of threshold value is controlled by changing $\delta_k^{\text{threshold}}$ values in Algorithm 1. Thus, we can improve the accuracy by lowering $\delta_k^{\text{threshold}}$: for $\delta_k^{\text{threshold}} = 0.05$, the accuracy drops by $4\%$ only while still retaining a communication saving of $55\%$, and for $\delta_k^{\text{threshold}} = 0.01$, we get a $22\%$ communication saving for a negligible drop in accuracy (by only $0.01\%$). In Fig. 6, we analyze `LBGM` under different $\delta_k^{\text{threshold}}$ values for different datasets. A drop in model performance can be observed as we increase $\delta_k^{\text{threshold}}$, which is accompanied by an increase in communication savings. This is consistent with takeaway 5 from Sec. 3, i.e., while a higher threshold requires less frequent updates of the LBGs, it reduces the convergence speed. Refer to Appendix F.2 for additional results.

`LBGM` **as a Plug-and-Play Algorithm.** For the plug-and-play setup, `LBGM` follows the same steps as in Algorithm 1, with the slight modification that the output of gradient compression techniques, top-K and ATOMO, are used in place of accumulated gradients $\boldsymbol{g}_k^{(t)}$ and LBGs $\boldsymbol{g}_k^\ell$, $\forall k$. In Fig. 7, we see that `LBGM` adds on top of existing communication benefits of both top-K and ATOMO, on the order of $10^6$ and $10^5$ floating point parameters shared per worker, respectively $(30 - 70\%$ savings across the datasets). The bottom row shows the accuracy/loss improvements that can be obtained for the same number of parameters transferred. While top-K and ATOMO compress gradients through approximation, they do not alter the underlying low-rank characteris-

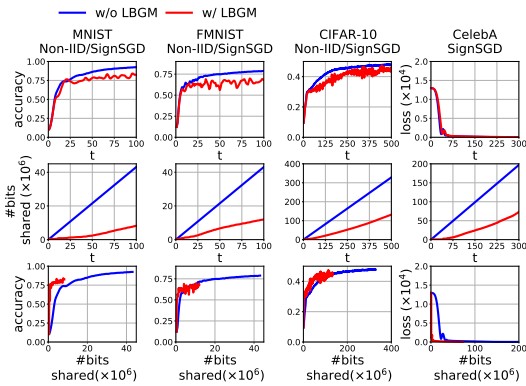

Figure 8: Application of `LBGM` as a plug-and-play algorithm on top of SignSGD in distributed training.

tics of the gradient-space. `LBGM` exploits this property to obtain substantial communication savings on top of these algorithms. Refer to Appendix F.3 for additional experiments.

**Generalizability of `LBGM` to Distributed Training.** `LBGM` can be applied to more general distributed gradient computation settings, e.g., multi-core systems. While heterogeneous (non-iid) data distributions are not as much of a consideration in these settings as they are in FL (since data can

be transferred/allocated across nodes), there is research interest in minimizing parameter exchange among nodes to reduce communication latency. SignSGD (Bernstein et al., 2018) is known to reduce the communication requirements by several order of magnitude by converting floating-point parameters to sign bit communication. In Fig. 8, we apply LBGM as a plug-and-play addition on top of SignSGD and find that LBGM further reduces the overall bits transferred by SignSGD on the order of $10^7$ bits ($60 - 80\%$ savings across the datasets). Refer to Appendix F.4 for additional experiments.

## 5 RELATED WORK

**NN Overparameterization Analysis.** Several prior works on NN overparameterization have focused on Hessian-based analysis. Sagun et al. (2016; 2017) divide the eigenspace of the Hessians into two parts: bulk and edges, and show that increasing network complexity only affects the bulk component. Ghorbani et al. (2019) argues that the existence of large isolated eigenvalues in the Hessian eigenspace is correlated with slow convergence. Gur-Ari et al. (2018) studies the overlap of gradients with Hessians and shows the Hessian edge space remains invariant during training, and thus that SGD occurs in low-rank subspaces. Gur-Ari et al. (2018) also suggest that the edge space cardinality is equal to the number of classification classes, which does not align with our observations in Sec. 2. In contrast to these, our work explores the low-rank property by studying the PCA of the gradient-space directly. Li et al. (2021), a contemporary of ours, employs the spectral decomposition of the NN gradient space to improve centralized SGD training time. Our methodology based on Hypothesis (**H2**) is more suitable for FL since having the resource-constrained workers/devices execute spectral decomposition as a component of the training process would add significant computational burden.

The partitioning of the gradient subspace has also been observed in the domain of continual learning (Chaudhry et al., 2020; Saha et al., 2020). However, the subspace addressed in continual learning is the one spanned by gradient with respect to data samples for the final model, which is different than the subspace we consider, i.e., the subspace of gradient updates generated during SGD epochs.

**Gradient Compression.** Gradient compression techniques can be broadly categorized into (i) sparsification (Wangni et al., 2018; Sattler et al., 2019), (ii) quantization (Seide et al., 2014; Alistarh et al., 2017), and (iii) low-rank approximations (Wang et al., 2018; Vogels et al., 2019; Albasyoni et al., 2020; Haddadpour et al., 2021). Our work falls under the third category, where prior works have aimed to decompose large gradient matrices as an outer product of smaller matrices to reduce communication cost. This idea was also proposed in Konečný et al. (2016), one of the pioneering works in FL. While these prior works study the low-rank property in the context of gradient compression during a single gradient transfer step, our work explores the low rank property of the gradients generated across successive gradient epochs during FL. Existing techniques for gradient compression can also benefit from employing LBGM during FL, as we show in our experiments for top-K, ATOMO, and SignSGD.

**Model Compression.** Model compression techniques have also been proposed to reduce NN complexity, e.g., model distillation (Ba & Caruana, 2014; Hinton et al., 2015), model pruning (LeCun et al., 1990; Hinton et al., 2015), and parameter clustering (Son et al., 2018; Cho et al., 2021). (Li et al., 2020) extends the lottery ticket hypothesis to the FL setting. These methods have the potential to be employed in conjunction with LBGM to reduce the size of the LBGs stored at the server.

**FL Communication Efficiency.** Other techniques have focused on reducing the aggregation frequency in FL. Hosseinalipour et al. (2020); Lin et al. (2021) use peer-to-peer local network communication, while SloMo (Wang et al., 2020b) uses momentum to delay the global aggregations.

## 6 DISCUSSION & CONCLUSIONS

In this paper, we explored the effect of overparameterization in NN optimization through the PCA of the gradient-space, and employed this to optimize the accuracy vs. communication tradeoff in FL. We proposed the LBGM algorithm, which uses our hypothesis that PGDs can be approximated using a subset of gradients generated across SGD epochs, and recycles previously generated gradients at the devices to represent the newly generated gradients. LBGM reduces communication overhead in FL by several orders of magnitude by replacing the transmission of gradient parameter vectors with a single scalars from each device. We theoretically characterized the convergence behavior of LBGM algorithm and experimentally substantiated our claims on several datasets and models. Furthermore, we showed that LBGM can be extended to further reduce latency of communication in large distributed training systems by plugging LBGM on top of other gradient compression techniques. More generally, our work gives a novel insight to designing a class of techniques based on "Look-back Gradients" that can be used in distributed machine learning systems to enhance communication savings.

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

# Appendix

## Table of Contents

## A  PROOF OF THEOREM 1

We start by introducing a lemma, that is used prominently throughout the analysis that follows.

**Lemma 1.** *For a sequence of vectors* $\{\boldsymbol{a}_i\}_{i=1}^N$, *such that* $\mathbb{E}\left[\boldsymbol{a}_i|\boldsymbol{a}_{i-1}, \boldsymbol{a}_{i-2}, \cdots, \boldsymbol{a}_1\right] = \boldsymbol{0}, \forall i$,

$$\mathbb{E}\left[\left\|\sum_{i=1}^N \boldsymbol{a}_i\right\|^2\right] = \sum_{i=1}^N \mathbb{E}\left[\|\boldsymbol{a}_i\|^2\right]. \tag{L2}$$

*Proof.*

$$\mathbb{E}\left[\left\|\sum_{i=1}^N \boldsymbol{a}_i\right\|^2\right] = \sum_{i=1}^N \mathbb{E}\left[\|\boldsymbol{a}_i\|^2\right] + \sum_{i=1}^N \sum_{\substack{j=1 \\ j\neq i}}^N \mathbb{E}\left[\boldsymbol{a}_i^\top \boldsymbol{a}_j\right]. \tag{5}$$

Using the law of total expectation, assuming $i < j$, we get

$$\mathbb{E}\left[\boldsymbol{a}_i^\top \boldsymbol{a}_j\right] = \mathbb{E}\left[\boldsymbol{a}_i^\top \mathbb{E}\left[\boldsymbol{a}_j|\boldsymbol{a}_i, \cdots, \boldsymbol{a}_1\right]\right] = \boldsymbol{0}, \tag{6}$$

which completes the proof. ∎

Next, we define a few auxiliary variables that would be referenced in the proof later: as defined in the main text, $\boldsymbol{g}_k^{(t)} = \sum_{b=0}^{\tau-1} g_k(\boldsymbol{\theta}_k^{(t,b)})$ is the accumulated stochastic gradient at worker $k$, where $b$ ranging from 0 to $\tau - 1$ denotes the rounds of local updates. Using Assumption (A2), error in stochastic gradient approximation $g_k(\boldsymbol{\theta}_k^{(t,b)})$ can be defined as $\boldsymbol{\epsilon}_k^{(t,b)} = g_k(\boldsymbol{\theta}_k^{(t,b)}) - \nabla F_k(\boldsymbol{\theta}_k^{(t,b)})$. Consequently, we can write the stochastic gradient as $g_k(\boldsymbol{\theta}_k^{(t,b)}) = \nabla F_k(\boldsymbol{\theta}_k^{(t,b)}) + \boldsymbol{\epsilon}_k^{(t,b)}$ where $\nabla F_k(\boldsymbol{\theta}_k^{(t,b)})$ is the true gradient. From Assumption (A2) it follows that

$$\mathbb{E}\left[\boldsymbol{\epsilon}_k^{(t,b)}\right] = \boldsymbol{0} \text{ and } \mathbb{E}\left[\left\|\boldsymbol{\epsilon}_k^{(t,b)}\right\|^2\right] \leq \sigma^2. \tag{7}$$

We also introduce the normalized stochastic gradient $\boldsymbol{d}_k^{(t)}$ given by

$$\boldsymbol{d}_k^{(t)} = \frac{\boldsymbol{g}_k^{(t)}}{\tau} = \frac{1}{\tau}\sum_{b=0}^{\tau-1} g_k(\boldsymbol{\theta}_k^{(t,b)}) = \frac{1}{\tau}\sum_{b=0}^{\tau-1} \nabla F_k(\boldsymbol{\theta}_k^{(t,b)}) + \frac{1}{\tau}\sum_{b=0}^{\tau-1} \boldsymbol{\epsilon}_k^{(t,b)} = \boldsymbol{h}_k^{(t)} + \boldsymbol{\epsilon}_k^{(t)}, \tag{8}$$

where we define

$$\text{Cumulative Average of the true gradient: } \boldsymbol{h}_k^{(t)} = \frac{1}{\tau}\sum_{b=0}^{\tau-1} \nabla F_k(\boldsymbol{\theta}_k^{(t,b)}), \text{ and} \tag{9}$$

$$\text{Cumulative Average of the SGD error: } \boldsymbol{\epsilon}_k^{(t)} = \frac{1}{\tau}\sum_{b=0}^{\tau-1} \boldsymbol{\epsilon}_k^{(t,b)}. \tag{10}$$

Next, we evaluate the first and second moment of normalized SGD error, $\boldsymbol{\epsilon}_k^{(t)}$ as follows

$$\mathbb{E}\left[\boldsymbol{\epsilon}_k^{(t)}\right] = \mathbb{E}\left[\frac{1}{\tau}\sum_{b=0}^{\tau-1} \boldsymbol{\epsilon}_k^{(t,b)}\right] = \frac{1}{\tau}\sum_{b=0}^{\tau-1} \mathbb{E}\left[\boldsymbol{\epsilon}_k^{(t,b)}\right] = \boldsymbol{0}, \tag{11}$$

$$\mathbb{E}\left[\left\|\boldsymbol{\epsilon}_k^{(t)}\right\|^2\right] = \mathbb{E}\left[\left\|\frac{1}{\tau}\sum_{b=0}^{\tau-1} \boldsymbol{\epsilon}_k^{(t,b)}\right\|^2\right] = \frac{1}{\tau^2}\sum_{b=0}^{\tau-1} \mathbb{E}\left[\left\|\boldsymbol{\epsilon}_k^{(t,b)}\right\|^2\right] \leq \frac{\sigma^2}{\tau}, \tag{12}$$

where (11) uses Assumption (A2) and (12) uses Lemma 1.

Now we can proceed to the proof of Theorem 1. We start with the update rule of LBGM, where a round of global update is given by

$$\boldsymbol{\theta}^{(t+1)} = \boldsymbol{\theta}^{(t)} - \eta\sum_{k=1}^K \omega_k \widetilde{\boldsymbol{g}}_k^{(t)} = \boldsymbol{\theta}^{(t)} - \tau\eta\sum_{k=1}^K \omega_k \widetilde{\boldsymbol{d}}_k^{(t)}, \tag{13}$$

where $\widetilde{\boldsymbol{g}}_k^{(t)}$ is the approximate gradient shared by worker $k$ with the server and $\widetilde{\boldsymbol{d}}_k^{(t)}$ is given by

$$\widetilde{\boldsymbol{d}}_k^{(t)} = \widetilde{\boldsymbol{g}}_k^{(t)}/\tau = \rho_k^{(t),\ell}\boldsymbol{g}_k^\ell/\tau = \rho_k^{(t),\ell}\boldsymbol{d}_k^\ell \tag{14}$$

where $\boldsymbol{d}_k^\ell = \boldsymbol{g}_k^\ell/\tau$ is the normalized stochastic gradient w.r.t. the last LBG shared by worker $k$. Also, from the trigonometric relationship shown in Fig. 4, we have,

$$\left\| \rho_k^{(t),\ell}\boldsymbol{d}_k^\ell \right\| = \left\| \boldsymbol{d}_k^{(t)}\cos(\alpha_k^{(t),\ell}) \right\| \tag{15}$$

Similar to $\boldsymbol{d}_k^{(t)}$ from (8), $\boldsymbol{d}_k^\ell$ can be split into the cumulative average of true look-back gradient $\boldsymbol{h}_k^\ell$ and the corresponding cumulative average of SGD error $\boldsymbol{\epsilon}_k^\ell$, given by

$$\boldsymbol{d}_k^\ell = \boldsymbol{h}_k^\ell + \boldsymbol{\epsilon}_k^\ell. \tag{16}$$

From Assumption (A1), the global loss function is $\beta$ smooth, which implies:

$$F(\boldsymbol{\theta}^{(t+1)}) - F(\boldsymbol{\theta}^{(t)}) \leq \left\langle \nabla F(\boldsymbol{\theta}^{(t)}), \boldsymbol{\theta}^{(t+1)} - \boldsymbol{\theta}^{(t)} \right\rangle + \frac{\beta}{2}\left\| \boldsymbol{\theta}^{(t+1)} - \boldsymbol{\theta}^{(t)} \right\|^2. \tag{17}$$

Taking expectation over $\boldsymbol{\epsilon}_k^{(t,b)}, k \in \{1, 2, \cdots, K\}, b \in \{0, 1, \cdots, \tau-1\}$,

$$\mathbb{E}\left[ F(\boldsymbol{\theta}^{(t+1)}) \right] - F(\boldsymbol{\theta}^{(t)})$$

$$\leq -\tau\eta \underbrace{\mathbb{E}\left[ \left\langle \nabla F(\boldsymbol{\theta}^{(t)}), \sum_{k=1}^K \omega_k \cdot \widetilde{\boldsymbol{d}}_k^{(t)} \right\rangle \right]}_{\mathbf{Z_1}} + \frac{\tau^2\eta^2\beta}{2} \underbrace{\mathbb{E}\left[ \left\| \sum_{k=1}^K \omega_k\widetilde{\boldsymbol{d}}_k^{(t)} \right\|^2 \right]}_{\mathbf{Z_2}}. \tag{18}$$

We evaluate $\mathbf{Z_1}$ as follows

$$\mathbf{Z_1} = \mathbb{E}\left[ \left\langle \nabla F(\boldsymbol{\theta}^{(t)}), \sum_{k=1}^K \omega_k\widetilde{\boldsymbol{d}}_k^{(t)} \right\rangle \right] = \mathbb{E}\left[ \left\langle \nabla F(\boldsymbol{\theta}^{(t)}), \sum_{k=1}^K \omega_k\left( \boldsymbol{d}_k^{(t)} - \boldsymbol{d}_k^{(t)} + \widetilde{\boldsymbol{d}}_k^{(t)} \right) \right\rangle \right]$$

$$= \underbrace{\mathbb{E}\left[ \left\langle \nabla F(\boldsymbol{\theta}^{(t)}), \sum_{k=1}^K \omega_k\boldsymbol{d}_k^{(t)} \right\rangle \right]}_{\mathbf{Z_{1,1}}} - \underbrace{\mathbb{E}\left[ \left\langle \nabla F(\boldsymbol{\theta}^{(t)}), \sum_{k=1}^K \omega_k\left( \boldsymbol{d}_k^{(t)} - \widetilde{\boldsymbol{d}}_k^{(t)} \right) \right\rangle \right]}_{\mathbf{Z_{1,2}}}, \tag{19}$$

where $\mathbf{Z_{1,1}}$ is given by,

$$\mathbf{Z_{1,1}} \overset{(i)}{=} \mathbb{E}\left[ \left\langle \nabla F(\boldsymbol{\theta}^{(t)}), \sum_{k=1}^K \omega_k\boldsymbol{h}_k^{(t)} \right\rangle \right] + \mathbb{E}\left[ \left\langle \nabla F(\boldsymbol{\theta}^{(t)}), \sum_{k=1}^K \omega_k\boldsymbol{\epsilon}_k^{(t)} \right\rangle \right]$$

$$\overset{(ii)}{=} \frac{1}{2}\left\| \nabla F(\boldsymbol{\theta}^{(t)}) \right\|^2 + \frac{1}{2}\mathbb{E}\left[ \left\| \sum_{k=1}^K \omega_k\boldsymbol{h}_k^{(t)} \right\|^2 \right] - \frac{1}{2}\mathbb{E}\left[ \left\| \nabla F(\boldsymbol{\theta}^{(t)}) - \sum_{k=1}^K \omega_k\boldsymbol{h}_k^{(t)} \right\|^2 \right], \tag{20}$$

where $(i)$ follows from (16) and $(ii)$ uses $2\langle \mathbf{a}, \mathbf{b}\rangle = \|\mathbf{a}\|^2 + \|\mathbf{b}\|^2 - \|\mathbf{a} - \mathbf{b}\|^2$ for any two real vectors $\mathbf{a}$ and $\mathbf{b}$. We next upper bound the term $\mathbf{Z_{1,2}}$ (although $Z_{1,2}$ has a negative sign in (19), $Z_1$ also appears with a negative sign in (18) which allows us to do the upper bound) as follows

$$\mathbf{Z_{1,2}} = \mathbb{E}\left[ \left\langle \nabla F(\boldsymbol{\theta}^{(t)}), \sum_{k=1}^K \omega_k\left( \boldsymbol{d}_k^{(t)} - \widetilde{\boldsymbol{d}}_k^{(t)} \right) \right\rangle \right]$$

$$\overset{(i)}{\leq} \frac{1}{4}\left\| \nabla F(\boldsymbol{\theta}^{(t)}) \right\|^2 + \mathbb{E}\left[ \left\| \sum_{k=1}^K \omega_k\left( \boldsymbol{d}_k^{(t)} - \widetilde{\boldsymbol{d}}_k^{(t)} \right) \right\|^2 \right], \tag{21}$$

where $(i)$ follows from $\langle a, b\rangle \leq (1/4)\|a\|^2 + \|b\|^2$ (result of Cauchy-Schwartz and Young's inequalities). Substituting (20) and (21) back in (19), we get

$$-\mathbf{Z_1} \leq -\frac{1}{4}\left\|\nabla F(\boldsymbol{\theta}^{(t)})\right\|^2 - \frac{1}{2}\mathbb{E}\left[\left\|\sum_{k=1}^{K}\omega_k \boldsymbol{h}_k^{(t)}\right\|^2\right] + \frac{1}{2}\mathbb{E}\left[\left\|\nabla F(\boldsymbol{\theta}^{(t)}) - \sum_{k=1}^{K}\omega_k \boldsymbol{h}_k^{(t)}\right\|^2\right]$$

$$+ \mathbb{E}\left[\left\|\sum_{k=1}^{K}\omega_k\left(\boldsymbol{d}_k^{(t)} - \widetilde{\boldsymbol{d}}_k^{(t)}\right)\right\|^2\right]. \tag{22}$$

We next bound the term $\mathbf{Z_2}$ in (18) as follows

$$\mathbf{Z_2} = \mathbb{E}\left[\left\|\sum_{k=1}^{K}\omega_k \widetilde{\boldsymbol{d}}_k^{(t)}\right\|^2\right] = \mathbb{E}\left[\left\|\sum_{k=1}^{K}\omega_k\left(\boldsymbol{d}_k^{(t)} - \boldsymbol{d}_k^{(t)} + \widetilde{\boldsymbol{d}}_k^{(t)}\right)\right\|^2\right]$$

$$\overset{(i)}{\leq} 2\,\mathbb{E}\underbrace{\left[\left\|\sum_{k=1}^{K}\omega_k \boldsymbol{d}_k^{(t)}\right\|^2\right]}_{\mathbf{Z_{2,1}}} + 2\mathbb{E}\left[\left\|\sum_{k=1}^{K}\omega_k\left(\boldsymbol{d}_k^{(t)} - \widetilde{\boldsymbol{d}}_k^{(t)}\right)\right\|^2\right], \tag{23}$$

where $\mathbf{Z_{2,1}}$ is given by,

$$\mathbf{Z_{2,1}} = \mathbb{E}\left[\left\|\sum_{k=1}^{K}\omega_k \boldsymbol{d}_k^{(t)}\right\|^2\right] \overset{(i)}{=} \mathbb{E}\left[\left\|\sum_{k=1}^{K}\omega_k \boldsymbol{h}_k^{(t)}\right\|^2\right] + \mathbb{E}\left[\left\|\sum_{k=1}^{K}\omega_k \boldsymbol{\epsilon}_k^{(t)}\right\|^2\right]$$

$$\overset{(ii)}{\leq} \mathbb{E}\left[\left\|\sum_{k=1}^{K}\omega_k \boldsymbol{h}_k^{(t)}\right\|^2\right] + \sum_{k=1}^{K}\omega_k\mathbb{E}\left[\left\|\boldsymbol{\epsilon}_k^{(t)}\right\|^2\right] \overset{(iii)}{=} \mathbb{E}\left[\left\|\sum_{k=1}^{K}\omega_k \boldsymbol{h}_k^{(t)}\right\|^2\right] + \frac{\sigma^2}{\tau}, \tag{24}$$

where $(i)$ follows from (8), $(ii)$ uses Jensen's inequality: $\|\sum_{k=1}^{K}\omega_k \mathbf{a}_k\|^2 \leq \sum_{k=1}^{K}\omega_k \|\mathbf{a}_k\|^2$, s.t. $\sum_{k=1}^{K}\omega_k = 1$, and $(iii)$ uses (12). Plugging (24) back into (23), we get

$$\mathbf{Z_2} \leq 2\mathbb{E}\left[\left\|\sum_{k=1}^{K}\omega_k \boldsymbol{h}_k^{(t)}\right\|^2\right] + \frac{2\sigma^2}{\tau} + 2\mathbb{E}\left[\left\|\sum_{k=1}^{K}\omega_k\left(\boldsymbol{d}_k^{(t)} - \widetilde{\boldsymbol{d}}_k^{(t)}\right)\right\|^2\right]. \tag{25}$$

Substituting (22) and (25) back in (18), we get

$$\mathbb{E}\left[F(\boldsymbol{\theta}^{(t+1)})\right] - F(\boldsymbol{\theta}^{(t)}) \leq -\frac{\tau\eta}{4}\left\|\nabla F(\boldsymbol{\theta}^{(t)})\right\|^2 - \frac{\tau\eta}{2}\mathbb{E}\left[\left\|\sum_{k=1}^{K}\omega_k \boldsymbol{h}_k^{(t)}\right\|^2\right]$$

$$+ \frac{\tau\eta}{2}\mathbb{E}\left[\left\|\nabla F(\boldsymbol{\theta}^{(t)}) - \sum_{k=1}^{K}\omega_k \boldsymbol{h}_k^{(t)}\right\|^2\right] + \tau\eta\mathbb{E}\left[\left\|\sum_{k=1}^{K}\omega_k\left(\boldsymbol{d}_k^{(t)} - \widetilde{\boldsymbol{d}}_k^{(t)}\right)\right\|^2\right]$$

$$+ \tau^2\eta^2\beta\mathbb{E}\left[\left\|\sum_{k=1}^{K}\omega_k \boldsymbol{h}_k^{(t)}\right\|^2\right] + \tau\eta^2\beta\sigma^2 + \tau^2\eta^2\beta\mathbb{E}\left[\left\|\sum_{k=1}^{K}\omega_k\left(\boldsymbol{d}_k^{(t)} - \widetilde{\boldsymbol{d}}_k^{(t)}\right)\right\|^2\right]$$

$$= -\frac{\tau\eta}{4}\left\|\nabla F(\boldsymbol{\theta}^{(t)})\right\|^2 - \frac{\tau\eta}{2}(1 - 2\tau\eta\beta)\mathbb{E}\left[\left\|\sum_{k=1}^{K}\omega_k \boldsymbol{h}_k^{(t)}\right\|^2\right] + \frac{\tau\eta}{2}\mathbb{E}\left[\left\|\nabla F(\boldsymbol{\theta}^{(t)}) - \sum_{k=1}^{K}\omega_k \boldsymbol{h}_k^{(t)}\right\|^2\right]$$

$$+ \tau\eta(1 + \tau\eta\beta)\mathbb{E}\left[\left\|\sum_{k=1}^{K}\omega_k\left(\boldsymbol{d}_k^{(t)} - \widetilde{\boldsymbol{d}}_k^{(t)}\right)\right\|^2\right] + \tau\eta^2\beta\sigma^2. \tag{26}$$

Choosing $\tau\eta\beta \leq 1/2$, implies that $-\frac{\tau\eta}{2}(1-2\tau\eta\beta) \leq 0$ and $1+\tau\eta\beta \leq 3/2 < 2$, which results in simplification of (26) as:

$$
\frac{\mathbb{E}\left[F(\boldsymbol{\theta}^{(t+1)})\right] - F(\boldsymbol{\theta}^{(t)})}{\eta\tau} \leq -\frac{1}{4}\left\|\nabla F(\boldsymbol{\theta}^{(t)})\right\|^2 + \frac{1}{2}\mathbb{E}\left[\left\|\nabla F(\boldsymbol{\theta}^{(t)}) - \sum_{k=1}^{K}\omega_k \boldsymbol{h}_k^{(t)}\right\|^2\right]
$$

$$
+ 2\mathbb{E}\left[\left\|\sum_{k=1}^{K}\omega_k\left(\boldsymbol{d}_k^{(t)} - \widetilde{\boldsymbol{d}}_k^{(t)}\right)\right\|^2\right] + \eta\beta\sigma^2
$$

$$
\overset{(i)}{\leq} -\frac{1}{4}\left\|\nabla F(\boldsymbol{\theta}^{(t)})\right\|^2 + \frac{1}{2}\sum_{k=1}^{K}\omega_k\mathbb{E}\left[\left\|\nabla F_k(\boldsymbol{\theta}_k^{(t)}) - \boldsymbol{h}_k^{(t)}\right\|^2\right]
$$

$$
+ 2\sum_{k=1}^{K}\omega_k\underbrace{\mathbb{E}\left[\left\|\boldsymbol{d}_k^{(t)} - \widetilde{\boldsymbol{d}}_k^{(t)}\right\|^2\right]}_{\mathbf{Z_3}} + \eta\beta\sigma^2, \tag{27}
$$

where $(i)$ follows from Jensen's inequality $\|\sum_{k=1}^{K}\omega_k\boldsymbol{a}_k\|^2 \leq \sum_{k=1}^{K}\omega_k\|\mathbf{a}_k\|^2$, s.t. $\sum_{k=1}^{K}\omega_k = 1$ and $\boldsymbol{\theta}_k^{(t)} = \boldsymbol{\theta}^{(t)}, \forall k$ due to local synchronization. Now $\mathbf{Z_3}$ can be bounded as follows:

$$
\mathbf{Z_3} = \mathbb{E}\left[\left\|\boldsymbol{d}_k^{(t)} - \widetilde{\boldsymbol{d}}_k^{(t)}\right\|^2\right] \overset{(i)}{=} \mathbb{E}\left[\left\|\boldsymbol{d}_k^{(t)} - \rho_k^{(t),\ell}\boldsymbol{d}_k^\ell\right\|^2\right] \overset{(ii)}{=} \mathbb{E}\left[\left\|\boldsymbol{d}_k^{(t)} - \frac{\frac{1}{\tau^2}\left\langle\boldsymbol{g}_k^{(t)},\boldsymbol{g}_k^\ell\right\rangle}{\frac{1}{\tau^2}\left\|\boldsymbol{g}_k^\ell\right\|^2}\boldsymbol{d}_k^\ell\right\|^2\right]
$$

$$
\overset{(iii)}{=} \mathbb{E}\left[\left\|\boldsymbol{d}_k^{(t)} - \frac{\left\langle\boldsymbol{d}_k^{(t)},\boldsymbol{d}_k^\ell\right\rangle}{\left\|\boldsymbol{d}_k^\ell\right\|^2}\boldsymbol{d}_k^\ell\right\|^2\right] = \mathbb{E}\left[\left\|\boldsymbol{d}_k^{(t)}\right\|^2 - \frac{\left\langle\boldsymbol{d}_k^{(t)},\boldsymbol{d}_k^\ell\right\rangle^2}{\left\|\boldsymbol{d}_k^\ell\right\|^2}\right]
$$

$$
\overset{(iv)}{=} \mathbb{E}\left[\left\|\boldsymbol{d}_k^{(t)}\right\|^2 - \left\|\boldsymbol{d}_k^{(t)}\right\|^2\cos^2(\boldsymbol{\alpha}_k^{(t),\ell})\right] = \mathbb{E}\left[\left\|\boldsymbol{d}_k^{(t)}\right\|^2\left(1 - \cos^2(\boldsymbol{\alpha}_k^{(t),\ell})\right)\right]
$$

$$
= 2\mathbb{E}\left[\left\|\boldsymbol{d}_k^{(t)}\right\|^2\sin^2(\boldsymbol{\alpha}_k^{(t),\ell})\right] \overset{(v)}{\leq} \mathbb{E}\left[\Delta^2\right] = \Delta^2, \tag{28}
$$

where $(i)$ uses (14), $(ii)$ uses LBGM definition from (D1), $(iii)$ uses the fact that $\boldsymbol{d}_k^{(t)} = \boldsymbol{g}_k^\ell/\tau$, $(iv)$ uses $\langle a, b\rangle = \|a\|\|b\|\cos(\alpha)$, and $(v)$ follows from the condition in the theorem. Substituting (28) back in (27), we get

$$
\frac{\mathbb{E}\left[F(\boldsymbol{\theta}^{(t+1)})\right] - F(\boldsymbol{\theta}^{(t)})}{\eta\tau}
$$

$$
\leq -\frac{1}{4}\left\|\nabla F(\boldsymbol{\theta}^{(t)})\right\|^2 + \frac{1}{2}\sum_{k=1}^{K}\omega_k\underbrace{\mathbb{E}\left[\left\|\nabla F_k(\boldsymbol{\theta}_k^{(t)}) - \boldsymbol{h}_k^{(t)}\right\|^2\right]}_{\mathbf{Z_4}} + 2\Delta^2 + \eta\beta\sigma^2, \tag{29}
$$

where $\mathbf{Z_4}$ is given by,

$$
\mathbf{Z_4} = \mathbb{E}\left[\left\|\nabla F_k(\boldsymbol{\theta}_k^{(t)}) - \boldsymbol{h}_k^{(t)}\right\|^2\right] = \frac{1}{\tau^2}\mathbb{E}\left[\left\|\sum_{b=0}^{\tau-1}\left(\nabla F_k(\boldsymbol{\theta}_k^{(t,0)}) - \nabla F_k(\boldsymbol{\theta}_k^{(t,b)})\right)\right\|^2\right]
$$

$$
\leq \frac{1}{\tau}\sum_{b=0}^{\tau-1}\mathbb{E}\left[\left\|\nabla F_k(\boldsymbol{\theta}_k^{(t,0)}) - \nabla F_k(\boldsymbol{\theta}_k^{(t,b)})\right\|^2\right] \leq \frac{\beta^2}{\tau}\sum_{b=0}^{\tau-1}\mathbb{E}\left[\left\|\boldsymbol{\theta}_k^{(t,0)} - \boldsymbol{\theta}_k^{(t,b)}\right\|^2\right]. \tag{30}
$$

Also, using the local update rule $\boldsymbol{\theta}_k^{(t,b)} \leftarrow \boldsymbol{\theta}_k^{(t,0)} - \eta \sum_{s=0}^{b-1} \mathbf{g}_k(\boldsymbol{\theta}_k^{(t,s)})$, where $\boldsymbol{\theta}_k^{(t,b)}$ is the model parameter obtained at the $b$-th local iteration of the global round $t$ at device $k$, we get:

$$
\begin{aligned}
\mathbf{Z_5} &= \mathbb{E}\left[\left\|\boldsymbol{\theta}_k^{(t,0)} - \boldsymbol{\theta}_k^{(t,b)}\right\|^2\right] \\
&= \eta^2 \mathbb{E}\left[\left\|\sum_{s=0}^{b-1} \mathbf{g}_k(\boldsymbol{\theta}_k^{(t,s)})\right\|^2\right] \overset{(i)}{\leq} 2\eta^2 \mathbb{E}\left[\left\|\sum_{s=0}^{b-1} \nabla F_k(\boldsymbol{\theta}_k^{(t,s)})\right\|^2\right] + 2\eta^2 \mathbb{E}\left[\left\|\sum_{s=0}^{b-1} \boldsymbol{\epsilon}_k^{(t,s)}\right\|^2\right] \\
&\overset{(ii)}{\leq} 2\eta^2 b \sum_{s=0}^{b-1} \mathbb{E}\left[\left\|\nabla F_k(\boldsymbol{\theta}_k^{(t,s)})\right\|^2\right] + 2\eta^2 \sum_{s=0}^{b-1} \mathbb{E}\left[\left\|\boldsymbol{\epsilon}_k^{(t,s)}\right\|^2\right] \\
&\leq 2\eta^2 b \sum_{s=0}^{\tau-1} \mathbb{E}\left[\left\|\nabla F_k(\boldsymbol{\theta}_k^{(t,s)})\right\|^2\right] + 2\eta^2 \sigma^2 b,
\end{aligned}
\tag{31}
$$

where $(i)$ uses Cauchy-Schwartz inequality and $(ii)$ uses Lemma 1 and Cauchy-Schwartz inequality. Also note that

$$
\sum_{b=0}^{\tau-1} b = \frac{\tau(\tau-1)}{2}.
\tag{32}
$$

Taking the cumulative sum of both hand sides of $\mathbf{Z_5}$ from (31) over all batches, i.e., $\frac{1}{\tau}\sum_{b=0}^{\tau-1}$, and using (32), we get:

$$
\frac{1}{\tau}\sum_{b=0}^{\tau-1} \mathbb{E}\left[\left\|\boldsymbol{\theta}_k^{(t,0)} - \boldsymbol{\theta}_k^{(t,b)}\right\|^2\right] \leq \sigma^2 \eta^2 (\tau-1) + \eta^2 (\tau-1) \sum_{b=0}^{\tau-1} \underbrace{\mathbb{E}\left[\left\|\nabla F_k(\boldsymbol{\theta}_k^{(t,b)})\right\|^2\right]}_{\mathbf{Z_6}}.
\tag{33}
$$

Furthermore, term $\mathbf{Z_6}$ can be bounded as follows:

$$
\begin{aligned}
\mathbf{Z_6} &= \mathbb{E}\left[\left\|\nabla F_k(\boldsymbol{\theta}_k^{(t,b)})\right\|^2\right] \overset{(i)}{\leq} 2\mathbb{E}\left[\left\|\nabla F_k(\boldsymbol{\theta}_k^{(t,b)}) - \nabla F_k(\boldsymbol{\theta}_k^{(t,0)})\right\|^2\right] + 2\mathbb{E}\left[\left\|\nabla F_k(\boldsymbol{\theta}_k^{(t,0)})\right\|^2\right] \\
&\overset{(ii)}{\leq} 2\beta^2 \mathbb{E}\left[\left\|\boldsymbol{\theta}_k^{(t,b)} - \boldsymbol{\theta}_k^{(t,0)}\right\|^2\right] + 2\mathbb{E}\left[\left\|\nabla F_k(\boldsymbol{\theta}_k^{(t,0)})\right\|^2\right],
\end{aligned}
\tag{34}
$$

where $(i)$ uses Cauchy-Schwartz inequality and $(ii)$ uses (**A1**). Replacing $\mathbf{Z_6}$ in (33) using (34), we get:

$$
\begin{aligned}
&\frac{1}{\tau}\sum_{b=0}^{\tau-1} \underbrace{\mathbb{E}\left[\left\|\boldsymbol{\theta}_k^{(t,0)} - \boldsymbol{\theta}_k^{(t,b)}\right\|^2\right]}_{\mathbf{Z_5}} \\
&\leq \eta^2 \sigma^2 (\tau-1) + 2\eta^2 \beta^2 (\tau-1) \sum_{b=0}^{\tau-1} \underbrace{\mathbb{E}\left[\left\|\boldsymbol{\theta}_k^{(t,b)} - \boldsymbol{\theta}_k^{(t,0)}\right\|^2\right]}_{\mathbf{Z_5}} + 2\eta^2 \tau(\tau-1) \mathbb{E}\left[\left\|\nabla F_k(\boldsymbol{\theta}_k^{(t,0)})\right\|^2\right].
\end{aligned}
\tag{35}
$$

Note that $\mathbf{Z_5}$, which is originally defined in (31), appears both in the left hand side (LHS) and right hand side (RHS) of the above expression. Rearranging the terms in the above inequality yields:

$$
\frac{1}{\tau}\sum_{b=0}^{\tau-1} \mathbb{E}\left[\left\|\boldsymbol{\theta}_k^{(t,0)} - \boldsymbol{\theta}_k^{(t,b)}\right\|^2\right] \leq \frac{\eta^2 \sigma^2 (\tau-1)}{1 - 2\eta^2 \beta^2 \tau(\tau-1)} + \frac{2\eta^2 \tau(\tau-1)}{1 - 2\eta^2 \beta^2 \tau(\tau-1)} \mathbb{E}\left[\left\|\nabla F_k(\boldsymbol{\theta}_k^{(t,0)})\right\|^2\right].
\tag{36}
$$

Defining $H = 2\eta^2\beta^2\tau(\tau - 1)$, the above inequality can be re-written to evaluate $\mathbf{Z_4}$,

$$
\begin{aligned}
\mathbf{Z_4} &= \frac{\beta^2}{\tau} \sum_{b=0}^{\tau-1} \mathbb{E}\left[\left\|\boldsymbol{\theta}^{(t,0)} - \boldsymbol{\theta}^{(t,b)}\right\|^2\right] \\
&\leq \frac{\eta^2\beta^2\sigma^2(\tau-1)}{1 - 2\eta^2\beta^2\tau(\tau-1)} + \frac{2\eta^2\beta^2\tau(\tau-1)}{1 - 2\eta^2\beta^2\tau(\tau-1)} \mathbb{E}\left[\left\|\nabla F_k(\boldsymbol{\theta}_k^{(t,0)})\right\|^2\right] \\
&= \frac{\eta^2\beta^2\sigma^2}{1-H}(\tau-1) + \frac{H}{1-H}\mathbb{E}\left[\left\|\nabla F_k(\boldsymbol{\theta}_k^{(t,0)})\right\|^2\right].
\end{aligned}
\tag{37}
$$

Taking a weighted sum from the both hand sides of the above inequality across all the workers and using (**A3**), we get:

$$
\begin{aligned}
\frac{1}{2}\sum_{k=1}^{K} \omega_k \mathbb{E}&\left[\left\|\nabla F_k(\boldsymbol{\theta}_k^{(t)}) - \boldsymbol{h}_k^{(t)}\right\|^2\right] \\
&\leq \frac{\eta^2\beta^2\sigma^2(\tau-1)}{2(1-H)}\sum_{k=1}^{K}\omega_k + \frac{H}{2(1-H)}\sum_{k=1}^{K}\omega_k\mathbb{E}\left[\left\|\nabla F_k(\boldsymbol{\theta}_k^{(t,0)})\right\|^2\right] \\
&= \frac{\eta^2\beta^2\sigma^2(\tau-1)}{2(1-H)} + \frac{H}{2(1-H)}\sum_{k=1}^{K}\omega_k\mathbb{E}\left[\left\|\nabla F_k(\boldsymbol{\theta}_k^{(t)})\right\|^2\right] \\
&\overset{(i)}{\leq} \frac{\eta^2\beta^2\sigma^2(\tau-1)}{2(1-H)} + \frac{H\Upsilon^2}{2(1-H)}\mathbb{E}\left[\left\|\nabla F(\boldsymbol{\theta}^{(t)})\right\|^2\right] + \frac{H\Gamma^2}{2(1-H)},
\end{aligned}
\tag{38}
$$

where $(i)$ follows from (**A3**) and $\boldsymbol{\theta}_k^{(t)} = \boldsymbol{\theta}^{(t)}$, $\forall k$ since computation occurs at the instance of global aggregation. Next, plugging (38) back in (29), we get:

$$
\begin{aligned}
\frac{\mathbb{E}\left[F(\boldsymbol{\theta}^{(t+1)})\right] - F(\boldsymbol{\theta}^{(t)})}{\eta\tau} \\
\leq -\frac{1}{4}\left\|\nabla F(\boldsymbol{\theta}^{(t)})\right\|^2 &+ \frac{1}{2}\sum_{k=1}^{K}\omega_k\mathbb{E}\left[\left\|\nabla F_k(\boldsymbol{\theta}_k^{(t)}) - \boldsymbol{h}_k^{(t)}\right\|^2\right] + 2\Delta^2 + \eta\beta\sigma^2 \\
\leq -\frac{1}{4}\left\|\nabla F(\boldsymbol{\theta}^{(t)})\right\|^2 &+ \frac{\eta^2\beta^2\sigma^2(\tau-1)}{2(1-H)} + \frac{H\Upsilon^2}{2(1-H)}\mathbb{E}\left[\left\|\nabla F(\boldsymbol{\theta}^{(t)})\right\|^2\right] + \frac{H\Gamma^2}{2(1-H)} \\
&+ 2\Delta^2 + \eta\beta\sigma^2 \\
\leq -\frac{1}{4}\left(1 - \frac{2H\Upsilon^2}{1-H}\right)&\left\|\nabla F(\boldsymbol{\theta}^{(t)})\right\|^2 + \frac{\eta^2\beta^2\sigma^2(\tau-1)}{2(1-H)} + \frac{H\Gamma^2}{2(1-H)} + 2\Delta^2 + \eta\beta\sigma^2.
\end{aligned}
\tag{39}
$$

If $H \leq \frac{1}{1+2\alpha\Upsilon^2}$ for some constant $\alpha > 1$, then it follows that $\frac{1}{1-H} \leq 1 + \frac{1}{2\alpha\Upsilon^2}$ and $\frac{2H\Upsilon^2}{1-H} \leq \frac{1}{\alpha}$. Choosing $\alpha = 2$ we can simplify the above expression as follows:

$$
\begin{aligned}
\frac{\mathbb{E}\left[F(\boldsymbol{\theta}^{(t+1)})\right] - F(\boldsymbol{\theta}^{(t)})}{\eta\tau} \\
\leq -\frac{1}{8}\left\|\nabla F(\boldsymbol{\theta}^{(t)})\right\|^2 &+ 2\Delta^2 + \eta\beta\sigma^2 + \eta^2\beta^2\sigma^2(\tau-1)\left(\frac{1}{2} + \frac{1}{8\Upsilon^2}\right) + 2\eta^2\beta^2\Gamma^2\tau(\tau-1)\left(1 + \frac{1}{4\Upsilon^2}\right) \\
\leq -\frac{1}{8}\left\|\nabla F(\boldsymbol{\theta}^{(t)})\right\|^2 &+ 2\Delta^2 + \eta\beta\sigma^2 + \frac{5}{8}\eta^2\beta^2\sigma^2(\tau-1) + \frac{5}{2}\eta^2\beta^2\Gamma^2\tau(\tau-1).
\end{aligned}
\tag{40}
$$

Rearranging the terms in the above inequality and taking the average across all aggregation rounds from the both hand sides, yields:

$$\frac{1}{T} \sum_{t=0}^{T-1} \mathbb{E}\left[\left\|\nabla F(\boldsymbol{\theta}^{(t,0)})\right\|^2\right]$$

$$\leq \frac{8\left[\sum_{t=0}^{T-1} \mathbb{E}\left[F(\boldsymbol{\theta}^{(t)})\right] - F(\boldsymbol{\theta}^{(t+1)})\right]}{\eta\tau T} + 16\Delta^2 + 8\eta\beta\sigma^2 + 5\eta^2\beta^2\sigma^2(\tau - 1) + 20\eta^2\beta^2\Gamma^2\tau(\tau - 1)$$

$$= \frac{8\left[F(\boldsymbol{\theta}^{(0)}) - F(\boldsymbol{\theta}^{(T)})\right]}{\eta\tau T} + 16\Delta^2 + 8\eta\beta\sigma^2 + 5\eta^2\beta^2\sigma^2(\tau - 1) + 20\eta^2\beta^2\Gamma^2\tau(\tau - 1)$$

$$\leq \frac{8\left[F(\boldsymbol{\theta}^{(0)}) - F^\star\right]}{\eta\tau T} + 16\Delta^2 + 8\eta\beta\sigma^2 + 5\eta^2\beta^2\sigma^2(\tau - 1) + 20\eta^2\beta^2\Gamma^2\tau(\tau - 1), \tag{41}$$

where we used the fact that $F$ is bounded below, since $F_k$-s are presumed to be bounded below, and $F^\star \leq F(\boldsymbol{\theta}), \forall \boldsymbol{\theta} \in \mathbb{R}^M$. This completes the proof of Theorem 1.

## A.1 CONDITION ON LEARNING RATE

From the two conditions on the learning rate used in the analysis above, we have

$$\eta\beta \leq \frac{1}{2\tau} \tag{42}$$

$$2\eta^2\beta^2\tau(\tau - 1) \leq \frac{1}{1 + 4\Upsilon^2} \tag{43}$$

We can further tighten the second constraint as,

$$2\eta^2\beta^2\tau(\tau - 1) \leq 2\eta^2\beta^2\tau^2 \leq \frac{1}{1 + 4\Upsilon^2} \tag{44}$$

Combining the two we have,

$$\eta\beta \leq \min\left\{\frac{1}{2\tau}, \frac{1}{\tau\sqrt{2(1 + 4\Upsilon^2)}}\right\}. \tag{45}$$

## B  PROOF OF COROLLARY 1

Using (41) we have:

$$\frac{1}{T}\sum_{t=0}^{T-1}\mathbb{E}\left[\left\|\nabla F(\boldsymbol{\theta}^{(t,0)})\right\|^2\right]$$

$$\leq \frac{8\left[F(\boldsymbol{\theta}^{(0)}) - F^\star\right]}{\eta\tau T} + 16\Delta^2 + 8\eta\beta\sigma^2 + 5\eta^2\beta^2\sigma^2(\tau-1) + 20\eta^2\beta^2\Gamma^2\tau(\tau-1). \qquad (46)$$

Next, using the assumption in the corollary statement, we have $\Delta^2 \leq \eta$. We then can upper bound the RHS of (46) to get:

$$\frac{1}{T}\sum_{t=0}^{T-1}\mathbb{E}\left[\left\|\nabla F(\boldsymbol{\theta}^{(t,0)})\right\|^2\right]$$

$$\leq \frac{8\left[F(\boldsymbol{\theta}^{(0)}) - F^\star\right]}{\eta\tau T} + 16\eta + 8\eta\beta\sigma^2 + 5\eta^2\beta^2\sigma^2(\tau-1) + 20\eta^2\beta^2\Gamma^2\tau(\tau-1). \qquad (47)$$

Choosing $\eta = \sqrt{\frac{1}{\tau T}}$, we get,

$$\frac{1}{T}\sum_{t=0}^{T-1}\mathbb{E}\left[\left\|\nabla F(\boldsymbol{\theta}^{(t,0)})\right\|^2\right]$$

$$\leq \frac{8\left[F(\boldsymbol{\theta}^{(0)}) - F^\star\right]}{\sqrt{\tau T}} + \frac{8\beta\sigma^2}{\sqrt{\tau T}} + \frac{16}{\sqrt{\tau T}} + \frac{5\beta^2\sigma^2(\tau-1)}{\tau T} + \frac{20\beta^2\Gamma^2\tau(\tau-1)}{\tau T}. \qquad (48)$$

We can write the above expression as,

$$\frac{1}{T}\sum_{t=0}^{T-1}\mathbb{E}\left[\left\|\nabla F(\boldsymbol{\theta}^{(t,0)})\right\|^2\right]$$

$$\leq \mathcal{O}\left(\frac{1}{\sqrt{\tau T}}\right) + \mathcal{O}\left(\frac{\sigma^2}{\sqrt{\tau T}}\right) + \mathcal{O}\left(\frac{1}{\sqrt{\tau T}}\right) + \mathcal{O}\left(\frac{\sigma^2(\tau-1)}{\tau T}\right) + \mathcal{O}\left(\frac{(\tau-1)\Gamma^2}{T}\right). \qquad (49)$$

This completes the proof for Corollary 1.

## C    ADDITIONAL DISCUSSIONS

### C.1    LBGM STORAGE CONSIDERATIONS

As discussed in Section 3, LBGM requires both server and workers to store the last copy of the workers' look-back gradient (LBG). While storing a single LBG (same size as the model) at each device might be trivial since the space complexity increases by a constant factor (i.e., the space complexity increases from $\mathcal{O}(M)$ to $\mathcal{O}(2M) = \mathcal{O}(M)$ where $M$ is the model size), storage of LBGs at the server might require more careful considerations since it scales linearly with the number of devices (i.e., space complexity increases from $\mathcal{O}(M)$ to $\mathcal{O}(KM)$ where $M$ is the model size and $K$ is the number of workers). Thus, the storage requirements can scale beyond memory capabilities of an aggregation server for a very large scale federated learning systems. We, therefore propose the following solutions for addressing the storage challenge:

- *Storage Offloading.* In a large scale federated learning system, it is realistic to assume network hierarchy Hosseinalipour et al. (2020), e.g., base stations, edge servers, cloud servers, etc. In such cases the storage burden can be offloaded and distributed across the network hierarchy where the LBGs of the devices are stored.

- *LBG Compression.* If the LBGM is applied on top of existing compression techniques such as Top-K, ATOMO, etc., the size of LBGs to be stored at the server also gets compressed which reduces the storage burden. Alternatively, we could use parameter clustering techniques (Son et al., 2018; Cho et al., 2021) to reduce LBG size at the server.

- *LBG Clustering.* In a very large scale federated learning system, say with a billion workers, it's unrealistic to assume that all the billion clients have very different LBGs given the low rank hypothesis (**H1**) and possible local data similarities across the workers. It should therefore be possible to cluster the LBGs at the server into a smaller number of centroids and only saving the centroids of the clusters instead of saving all the LBGs of the devices individually. The centroids can be broadcast across the devices to update the local version of the LBGs.

### C.2    HYPERPARAMETER TUNING

Most of the compression/communication savings baselines operate on a tradeoff between communication savings and accuracy. For hyperparameter selection, we first optimize the hyperparameters for the base algorithm such that we achieve the best possible communication saving subject to constraint that accuracy does not fall off much below the corresponding vanilla federated learning approach. For example, we optimize the value of $K$ in top-K sparsification by changing $K$ in orders of 10, i.e. $K = 10\%$, $K = 1\%$, $K = 0.1\%$, etc. and choose the value that gives the best tradeoff between the final model accuracy and communication savings (this value is generally around $K = 10\%$). Similarly, for ATOMO we consider rank-1, rank-2, and rank-3 approximations. While rank-1 approximation gives better communication savings, the corresponding accuracy falls off sharply. Rank-3 approximation gives only a marginal accuracy benefit over rank-2 approximation but adds considerably more communication burden. Thus we use rank-2 approximations in ATOMO. In the plug-and-play evaluations, the LBGM algorithm is applied on top of the base algorithms once their hyperparameters are tuned as a final step to show the additional benefits we can attain by exploiting the low-rank characteristic of the gradient subspace. Our chosen hyperparameters can be found in our code repository: https://github.com/shams-sam/FedOptim.

## D    ADDITIONAL PSEUDOCODES

### D.1    PSEUDOCODE FOR PRELIMINARY EXPERIMENTS

In this Appendix, we provide a psuedocode for generating the preliminary experimental results in Sec. 2. The actual implementation of the following function calls used in the pseudocode can be found in the listed files of our code repository: https://github.com/shams-sam/FedOptim:

- get_num_PCA_components: implemented in function estimate_optimal_ncomponents, file: src/common/nb_utils.py of the repository. In summary, we stack the accumulated gradients over the epochs an perform singular value decomposition, after which we do the standard analysis

for estimating the number of components explaining a given amount of variance in the datasets. Specifically, we count the number of singular values that account for the $99\%$ and $95\%$ of the aggregated singular values.

- get_PCA_components: implemented in function pca_transform, file: src/common/nb_utils.py of the repository. In summary, we stack the accumulated gradients over the epochs an perform singular value decomposition, after which we do the standard analysis for recovering the principal components explaining a given amount of variance in the datasets. Specifically, we recover the left singular vectors corresponding the singular values that account for the $99\%$ and $95\%$ of the aggregated singular values.

- cosine_similarity: implemented using functions sklearn.preprocessing.normalize and numpy.dot, such that $\text{cosine\_similarity}(a, b) = \text{normalize}(a).\text{dot}(\text{normalize}(b))$ where a and b are numpy.array, where normalize performs the vector normalization and dot is the standard vector dot product.

- plot_1, plot_2, and plot_3: implemented in files src/viz/prelim_1.py, src/viz/prelim_2.py, and src/viz/prelim_3.py respectively.

---

**Algorithm 2** Pseudocode for Preliminary Experiments in Section 2

---

1: Initialize model parameter $\boldsymbol{\theta}^{(0)}$.
2: Initialize actual_grads = {} ▷ store gradients for PCA
3: Initialize pca95_store = {} ▷ store #components accounting for 95% variance
4: Initialize pca99_store = {} ▷ store #components accounting for 99% variance

5: **for** $t = 0$ to $T - 1$ **do** ▷ training for $T$ epochs
6:     Initialize $\boldsymbol{g}^{(t)} \leftarrow \boldsymbol{0}$. ▷ initialize gradient accumulator
7:     Set $\boldsymbol{\theta}^{(t,0)} \leftarrow \boldsymbol{\theta}^{(t)}$.
8:     **for** $b = 0$ to $B - 1$ **do** ▷ $B$ minibatches per epoch
9:         Sample a minibatch of datapoints $\mathcal{B}$ from dataset $\mathcal{D}$.
10:         Compute $\boldsymbol{g}^{(t,b)} = \sum_{d \in \mathcal{B}} \nabla f(\boldsymbol{\theta}^{(t,b)}; d)/|\mathcal{B}|$.
11:         Update parameter: $\boldsymbol{\theta}^{(t,b+1)} \leftarrow \boldsymbol{\theta}^{(t,b)} - \eta \cdot \boldsymbol{g}^{(t,b)}$.
12:         Accumulate gradient: $\boldsymbol{g}^{(t)} \leftarrow \boldsymbol{g}^{(t)} + \boldsymbol{g}^{(t,b)}$.
13:     **end for**
14:     Set $\boldsymbol{\theta}^{(t+1)} \leftarrow \boldsymbol{\theta}^{(t,B)}$.
15:     actual_grads.append($\boldsymbol{g}^{(t)}$) ▷ append accumulated gradient to store
16:     pca95_store.append(get_num_PCA_components(actual_grads, variance = 0.95))
17:     pca99_store.append(get_num_PCA_components(actual_grads, variance = 0.99))

18: **end for**

19: plot_1(pca95_store, pca99_store) ▷ plot of PCA component progression

20: principal_grads = get_PCA_components(actual_grads, variance = 0.99)
21: heatmap = zeros(len(actual_grads), len(principal_grads))
22: **for** $i = 0$ to len(actual_grads) $- 1$ **do**
23:     **for** $j = 0$ to len(principal_grads) $- 1$ **do**
24:         heatmap[i, j] = cosine_similarity(actual_grads[i], principal_grads[j])
25:     **end for**
26: **end for**

27: plot_2(heatmap) ▷ plot of overlap of actual and principal gradients

28: heatmap = zeros(len(actual_grads), len(actual_grads))
29: **for** $i = 0$ to len(actual_grads) $- 1$ **do**
30:     **for** $j = i$ to len(actual_grads) $- 1$ **do**
31:         heatmap[i, j] = heatmap[j, i] = cosine_similarity(actual_grads[i], actual_grads[j])
32:     **end for**
33: **end for**

34: plot_3(heatmap) ▷ plot of similarity among actual gradients

---

## D.2   Pseudocode for Device Sampling Experiments

The pseudocode for LBGM with device sampling is given below. The process is largely similar to Algorithm 1, except modifications in the global aggregation strategy. During global aggregation the server samples a subset $K'$ of all the available clients and receives updates from only those clients for aggregation as shown in line 15-17 of the Algorithm 3.

In terms of the effect of sampling on the unsampled devices, as long as an unsampled device that joins at a later step of the training have a fairly good look-back gradient (i.e., its newly generated gradient are close to its look-back gradient (LBG)), there is no need for transmission of the entire parameter vector. This would often happen in practice unless the device engages in model training after a very long period of inactivity, in which case it would need to transmit its updated LBG before engaging in LBGM communication savings.

---

**Algorithm 3** LBGM with Device Sampling

---

**Notation:**

$\boldsymbol{\theta}^{(t)}$: global model parameter at global aggregation round $t$.

$\boldsymbol{\theta}_k^{(t,b)}$: model parameter at worker $k$, at global aggregation round $t$ and local update $b$.

$\boldsymbol{g}_k^{(t)}$: accumulated gradient at worker $k$ at global aggregation round $t$.

$\boldsymbol{g}_k^{\ell}$: last full gradient transmitted to server, termed look-back gradient (LBG).

$\alpha_k^{(t),\ell}$: phase between the accumulated gradient $\boldsymbol{g}_k^{(t)}$ and LBG $\boldsymbol{g}_k^{\ell}$, termed look-back phase (LBP).

**Training at worker $k$:**

1: Update local parameters: $\boldsymbol{\theta}_k^{(t,0)} \leftarrow \boldsymbol{\theta}^{(t)}$, and initialize gradient accumulator: $\boldsymbol{g}_k^{(t)} \leftarrow \boldsymbol{0}$.
2: **for** $b = 0$ to $(\tau\text{-}1)$ **do**
3:     Sample a minibatch of datapoints $\mathcal{B}_k$ from $\mathcal{D}_k$ and compute $\boldsymbol{g}_k^{(t,b)} = \sum_{d \in \mathcal{B}_k} \nabla f_k(\boldsymbol{\theta}_k^{(t,b)}; d)/|\mathcal{B}_k|$.
4:     Update local parameters: $\boldsymbol{\theta}_k^{(t,b+1)} \leftarrow \boldsymbol{\theta}_k^{(t,b)} - \eta \cdot \boldsymbol{g}_k^{(t,b)}$, and accumulate gradient: $\boldsymbol{g}_k^{(t)} \leftarrow \boldsymbol{g}_k^{(t)} + \boldsymbol{g}_k^{(t,b)}$.
5: **end for**
6: Calculate the LBP error: $\sin^2(\alpha_k^{(t),\ell}) = 1 - \left( \langle \boldsymbol{g}_k^{(t)}, \boldsymbol{g}_k^{\ell} \rangle / \left( \|\boldsymbol{g}_k^{(t)}\| \times \|\boldsymbol{g}_k^{\ell}\| \right) \right)^2$
7: **if** $\sin^2(\alpha_k^{(t),\ell}) \leq \delta_k^{\text{threshold}}$ **then**                                        ▷ checking the LBP error
8:     Send scalar LBC to the server: $\boldsymbol{\mu}_k^{(t)} \leftarrow \rho_k^{(t),\ell} = \langle \boldsymbol{g}_k^{(t)}, \boldsymbol{g}_k^{\ell} \rangle / \|\boldsymbol{g}_k^{\ell}\|^2$.
9: **else**                                                                                          ▷ updating the LBG
10:     Send actual gradient to the server: $\boldsymbol{\mu}_k^{(t)} \leftarrow \boldsymbol{g}_k^{(t)}$.
11:     Update worker-copy of LBG: $\boldsymbol{g}_k^{\ell} \leftarrow \boldsymbol{g}_k^{(t)}$.
12: **end if**

**Global update at the aggregation server:**

13: Initialize global parameter $\boldsymbol{\theta}^{(0)}$ and broadcast it across workers.
14: **for** $t = 0$ to $(T-1)$ **do**
15:     sample set of indices $K'$, a random subset from the pool of devices $\{1, 2, ..., K\}$.
16:     Receive updates from workers $\{\boldsymbol{\mu}_k^{(t)}\}_{k \in K'}$.
17:     Update global parameters: $\boldsymbol{\theta}^{(t+1)} \leftarrow \boldsymbol{\theta}^{(t)} - \frac{\eta}{|K'|} \sum_{k \in K'} \omega_k \left[ s_k \cdot \boldsymbol{\mu}_k^{(t)} \cdot \boldsymbol{g}_k^{\ell} + (1 - s_k) \cdot \boldsymbol{\mu}_k^{(t)} \right]$,

   where $s_k$ is an indicator function given by, $s_k = \begin{cases} 1, & \text{if } \boldsymbol{\mu}_k^{(t)} \text{ is a scalar} \\ 0, & \text{otherwise, i.e., if } \boldsymbol{\mu}_k^{(t)} \text{ is a vector} \end{cases}$.

18:     Update server-copy of look-back gradients (LBGs): $\boldsymbol{g}_k^{\ell} \leftarrow (1 - s_k)\boldsymbol{\mu}_k^{(t)} + (s_k)\boldsymbol{g}_k^{\ell}, \ \ \forall k \in K'$.
19: **end for**

---

## E  ADDITIONAL PRELIMINARY EXPERIMENTS

We next present the additional experiments performed to test hypotheses, (**H1**) & (**H2**). As mentioned in Section 2, we study the rank-characteristics of centralized training using SGD on multiple datasets: FMNIST, MNIST, CIFAR-10, CelebA, COCO, and PascalVOC, and model architectures: CNN, FCN, Resnet18, VGG19, and U-Net. Section E.1 presents experiments complementary to the one presented in Fig. 1, while Section E.2 & E.3 present experiments complementary to those presented in Fig. 2, & 3, respectively. *Please follow the hyperlinks for the ease of navigating through the figures.*

### E.1  PCA COMPONENT PROGRESSION

Together with Fig. 1, Figs. 10-13 show that both N99-PCA and N95-PCA are significantly lower than that the total number of gradients calculated during model training irrespective of model/dataset/learning task for multiple datasets and models which all agree with (**H1**). Specifically, the principal gradients (i.e., red and blue lines in the top row of the plots) are substantially lower (often as low as 10% of number of epochs, i.e., gradients generated) in these experiments. Refer below for the details of the figures.

1. Fig. 9 repeats the experiment conducted in Fig. 1 on CIFAR-100 using FCN, CNN, Resnet18, & VGG19.

2. Fig. 10 repeats the experiment conducted in Fig. 1 on MNIST using FCN, CNN, Resnet18, & VGG19.

3. Fig. 11 repeats the experiment conducted in Fig. 1 on FMNIST using FCN, CNN, Resnet18, & VGG19.

4. Fig. 12 repeats the experiment conducted in Fig. 1 on CIFAR-10, FMNIST, and MNIST using SVM, suggesting that we can use `LBGM` for classic classifiers that are not necessary neural networks.

5. Fig. 13 repeats the experiment conducted in Fig. 1 on COCO, and PascalVOC using U-Net.

### E.2  OVERLAP OF ACTUAL AND PRINCIPAL GRADIENT

Next, we perform experiments summarized in Fig. 14-35 to further validate our observation in Fig. 2: (i) cosine similarity of actual gradients with principal gradients varies gradually over time, and (ii) actual gradients have a high cosine similarity with one or more of the principal gradients. Refer below for the details of the figures. *Each subplot is marked with #L, the layer number of the neural network, and #elem, the number of elements in each layer.*

In the plots with dense heatmaps (a large number of gradients in along x and y axis) for larger models such as Fig. 14, it is harder to observe the overlap among the actual gradient and the PCA gradients. However, we can still notice the number of prinicipal components (along y axis) is substantially lower than the total number of epochs gradients (along y axis). A better picture of gradient overlap with other gradients can still be seen in corresponding inter-gradient overlap plot in Fig. 36, which is consistent with the corresponding PCA progression shown in Fig. 1. Note that the lesser number of prinicipal gradient directions (e.g., CNN in Fig. 1) implies a higher overlap among the generated gradients (e.g., CNN in Fig. 3), while a larger number of PGDs (e.g., VGG19 in Fig. 1) implies a lower overlap among generated gradients (e.g., VGG19 in Fig. 40).

1. Fig. 14 repeats the experiment conducted in Fig. 2 on CelebA using VGG19.

2. Fig. 15 repeats the experiment conducted in Fig. 2 on CelebA using Resnet18.

3. Fig. 16 repeats the experiment conducted in Fig. 2 on CelebA using FCN.

4. Fig. 17 repeats the experiment conducted in Fig. 2 on CelebA using CNN.

5. Fig. 18 repeats the experiment conducted in Fig. 2 on CIFAR-10 using VGG19.

6. Fig. 19 repeats the experiment conducted in Fig. 2 on CIFAR-10 using Resnet18.

7. Fig. 20 repeats the experiment conducted in Fig. 2 on CIFAR-10 using FCN.

8. Fig. 21 repeats the experiment conducted in Fig. 2 on CIFAR-10 using CNN.

9. Fig. 22 repeats the experiment conducted in Fig. 2 on CIFAR-100 using VGG19.

10. Fig. 23 repeats the experiment conducted in Fig. 2 on CIFAR-100 using Resnet18.

11. Fig. 24 repeats the experiment conducted in Fig. 2 on CIFAR-100 using FCN.

12. Fig. 25 repeats the experiment conducted in Fig. 2 on CIFAR-100 using CNN.

13. Fig. 26 repeats the experiment conducted in Fig. 2 on FMNIST using VGG19.

14. Fig. 27 repeats the experiment conducted in Fig. 2 on FMNIST using Resnet18.

15. Fig. 28 repeats the experiment conducted in Fig. 2 on FMNIST using FCN.

16. Fig. 29 repeats the experiment conducted in Fig. 2 on FMNIST using CNN.

17. Fig. 30 repeats the experiment conducted in Fig. 2 on MNIST using VGG19.

18. Fig. 31 repeats the experiment conducted in Fig. 2 on MNIST using Resnet18.

19. Fig. 32 repeats the experiment conducted in Fig. 2 on MNIST using FCN.

20. Fig. 33 repeats the experiment conducted in Fig. 2 on MNIST using CNN.

21. Fig. 34 repeats the experiment conducted in Fig. 2 on PascalVOC using U-Net.

22. Fig. 35 repeats the experiment conducted in Fig. 2 on COCO using U-Net.

### E.3 SIMILARITY AMONG CONSECUTIVE GENERATED GRADIENTS

Furthermore, we perform experiments summarized in Fig. 36-57. Together with Fig. 3, these experiments show that there is a significant overlap of consecutive gradients generated during SGD iterations, which further substantiates (**H2**) and bolsters our main idea that *"gradients transmitted in FL can be recycled/reused to represent the gradients generated in the subsequent iterations"*. Refer below for the details of the figures.

1. Fig. 36 repeats the experiment conducted in Fig. 3 on CelebA using VGG19.

2. Fig. 37 repeats the experiment conducted in Fig. 3 on CelebA using Resnet18.

3. Fig. 38 repeats the experiment conducted in Fig. 3 on CelebA using FCN.

4. Fig. 39 repeats the experiment conducted in Fig. 3 on CelebA using CNN.

5. Fig. 40 repeats the experiment conducted in Fig. 3 on CIFAR-10 using VGG19.

6. Fig. 41 repeats the experiment conducted in Fig. 3 on CIFAR-10 using Resnet18.

7. Fig. 42 repeats the experiment conducted in Fig. 3 on CIFAR-10 using FCN.

8. Fig. 43 repeats the experiment conducted in Fig. 3 on CIFAR-10 using CNN.

9. Fig. 44 repeats the experiment conducted in Fig. 3 on CIFAR-100 using VGG19.

10. Fig. 45 repeats the experiment conducted in Fig. 3 on CIFAR-100 using Resnet18.

11. Fig. 46 repeats the experiment conducted in Fig. 3 on CIFAR-100 using FCN.

12. Fig. 47 repeats the experiment conducted in Fig. 3 on CIFAR-100 using CNN.

13. Fig. 48 repeats the experiment conducted in Fig. 3 on FMNIST using VGG19.

14. Fig. 49 repeats the experiment conducted in Fig. 3 on FMNIST using Resnet18.

15. Fig. 50 repeats the experiment conducted in Fig. 3 on FMNIST using FCN.

16. Fig. 51 repeats the experiment conducted in Fig. 3 on FMNIST using CNN.

17. Fig. 52 repeats the experiment conducted in Fig. 3 on MNIST using VGG19.

18. Fig. 53 repeats the experiment conducted in Fig. 3 on MNIST using Resnet18.

19. Fig. 54 repeats the experiment conducted in Fig. 3 on MNIST using FCN.

20. Fig. 55 repeats the experiment conducted in Fig. 3 on MNIST using CNN.

21. Fig. 56 repeats the experiment conducted in Fig. 3 on PascalVOC using U-Net.

22. Fig. 57 repeats the experiment conducted in Fig. 3 on COCO using U-Net.

## F    ADDITIONAL LBGM EXPERIMENTS

In this section, we present complimentary experiments to the properties studies in Section 4 of the main text. In particular, we show that our observations hold for datasets: CIFAR-10, CIFAR-100, CelebA, FMNIST, and MNIST. We also present results when using shallower models FCN or deeper models Resnet18 different from CNNs. Section F.1 gives further evidence of utility of LBGM as a standalone solution. Section F.2 lists figures that summarize the effect of changing $\delta_k^{\text{threshold}}$ on model performance for mentioned datasets and models. In Section F.3 we list figures that summarize additional experiments to support the observations made in Fig. 7 and Section F.4 lists figures for additional experiments corresponding to the observations made in Fig. 8. Finally, Section F.5 presents results on LBGM algorithm corresponding to the case wherein $50\%$ of clients are randomly sampled during global aggregation. *Since the two layer FCN considered is a simple classifier, it does not perform well on complex datasets such as CIFAR-10, CIFAR-100, and CelebA. Thus, the respective results are omitted for FCN on these complicated datasets and the performance of FCN is only studied for MNIST and FMNIST datasets. Similarly, the 4-layer CNN architecture does not perform well on CIFAR-100 dataset and hence the corresponding results are ommited. We also present results using U-Net architecture for semantic segmentation on PascalVOC dataset.*

### F.1    LBGM AS A STANDALONE ALGORITHM.

1. Fig. 58 shows the result of repeating the experiment conducted in Fig. 5 (CNNs on non-iid data distribution) on CNNs for iid data distribution for datasets CIFAR-10, FMNIST, and MNIST, and U-Net for segmentation for dataset PascalVOC.

2. Fig. 59 shows the result of repeating the experiment conducted in Fig. 5 (CNNs on non-iid data distribution) on FCNs for both non-iid and iid data distributions on datasets FMNIST and MNIST.

3. Fig. 60 shows the result of repeating the experiment conducted in Fig. 5 (CNNs on non-iid data distribution) on Resnet18s for non-iid data distribution on datasets CelebA, CIFAR-10, CIFAR-100, FMNIST, and MNIST using a setup similar to that of Wang et al. (2018).

### F.2    EFFECT OF $\delta_k^{\text{threshold}}$ ON LBGM.

1. Fig. 61 shows the result of repeating the experiment conducted in Fig. 6 (CNNs on non-iid data distribution) on CNNs for iid data distribution for datasets CIFAR-10, FMNIST, and MNIST, and U-Net for segmentation for dataset PascalVOC.

2. Fig. 62 shows the result of repeating the experiment conducted in Fig. 6 (CNNs on non-iid data distribution) on FCNs for both non-iid and iid data distributions on datasets FMNIST and MNIST.

3. Fig. 63 shows the result of repeating the experiment conducted in Fig. 6 (CNNs on non-iid data distribution) on Resnet18s for non-iid data distribution on datasets CelebA, CIFAR-10 and CIFAR-100 using a setup similar to that of Wang et al. (2018).

### F.3    LBGM AS A PLUG-AND-PLAY ALGORITHM.

1. Fig. 64 shows the result of repeating the experiment conducted in Fig. 7 (CNNs on non-iid data distribution) on CNNs for iid data distribution for datasets CIFAR-10, FMNIST, and MNIST.

2. Fig. 65 shows the result of repeating the experiment conducted in Fig. 7 (CNNs on non-iid data distribution) on FCNs for both non-iid and iid data distributions on datasets FMNIST and MNIST.

3. Fig. 66 shows the result of repeating the experiment conducted in Fig. 7 (CNNs on non-iid data distribution) on Resnet18s for non-iid data distribution on datasets CelebA, CIFAR-10 and CIFAR-100 using a setup similar to that of Wang et al. (2018).

### F.4    GENERALIZABILITY OF LBGM TO DISTRIBUTED TRAINING.

1. Fig. 67 shows the result of repeating the experiment conducted in Fig. 8 (CNNs on non-iid data distribution) on CNNs for iid data distribution for datasets CIFAR-10, FMNIST, and MNIST.

2. Fig. 68 shows the result of repeating the experiment conducted in Fig. 8 (CNNs on non-iid data distribution) on FCNs for both non-iid and iid data distributions on datasets FMNIST and MNIST.

3. Fig. 69 shows the result of repeating the experiment conducted in Fig. 8 (CNNs on non-iid data distribution) on Resnet18s for non-iid data distribution on datasets CelebA, CIFAR-10 and CIFAR-100 using a setup similar to that of Wang et al. (2018).

## F.5 LBGM UNDER CLIENT SAMPLING.

We present results with LBGM under client sampling in this subsection. The results are qualitatively similar to those presented in Sec. 4 under "LBGM as Standalone Algorithm". For example, our results on the MNIST dataset for $50\%$ client participation shows a $35\%$ and $55\%$ improvement in communication efficiency for only $0.2\%$ and $4\%$ drop in accuracy for the corresponding i.i.d and non-i.i.d cases (see column 1 of Fig. 71&70 respectively).

1. Fig. 70 shows the result of repeating the experiment conducted in Fig. 5 (CNNs on non-iid data distribution) under $50\%$ client sampling using CNNs for non-iid data distribution for datasets CIFAR-10, FMNIST, and MNIST, and regression for dataset CelebA.

2. Fig. 71 shows the result of repeating the experiment conducted in Fig. 5 (CNNs on non-iid data distribution) under $50\%$ client sampling using CNNs for iid data distribution for datasets CIFAR-10, FMNIST, and MNIST.

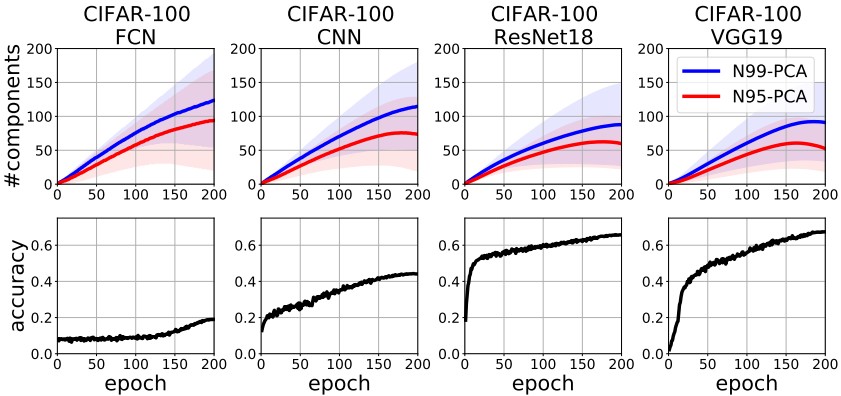

Figure 9: *PCA Components Progression*. Repeat of Fig. 1 on CIFAR-100 dataset.

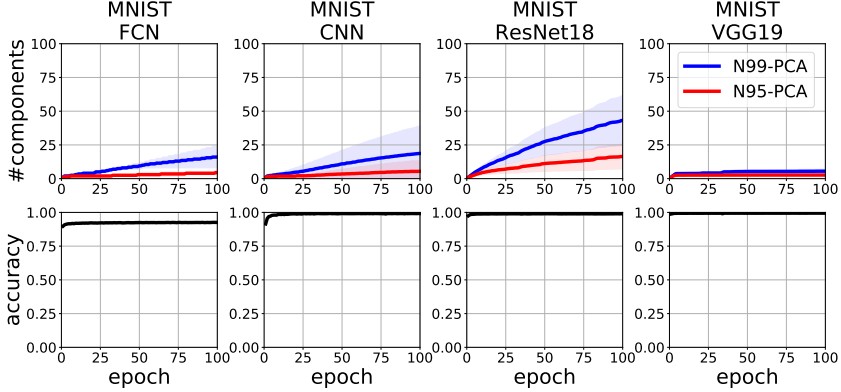

Figure 10: *PCA Components Progression*. Repeat of Fig. 1 on MNIST dataset.

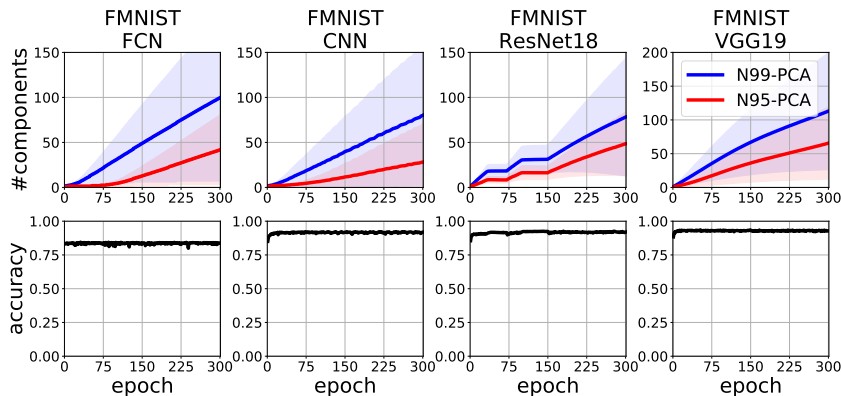

Figure 11: *PCA Components Progression*. Repeat of Fig. 1 on FMNIST dataset.

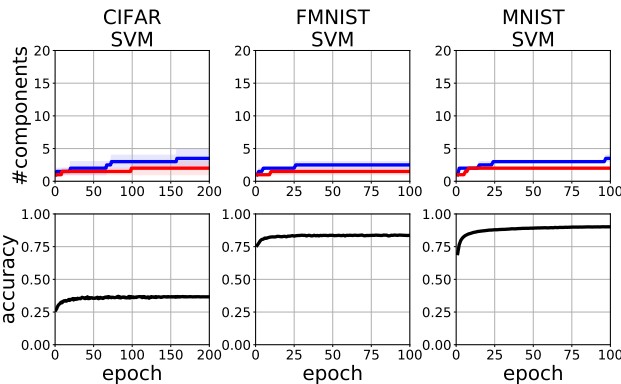

Figure 12: *PCA Components Progression*. Repeat of Fig. 1 on CIFAR-10, F-MNIST, and MNIST datasets but using squared SVM classifier.

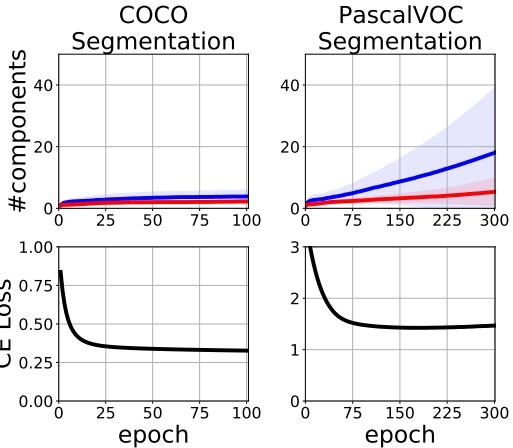

Figure 13: *PCA Components Progression*. Repeat of Fig. 1 on COCO, and PascalVOC datasets but using U-Net classifier.

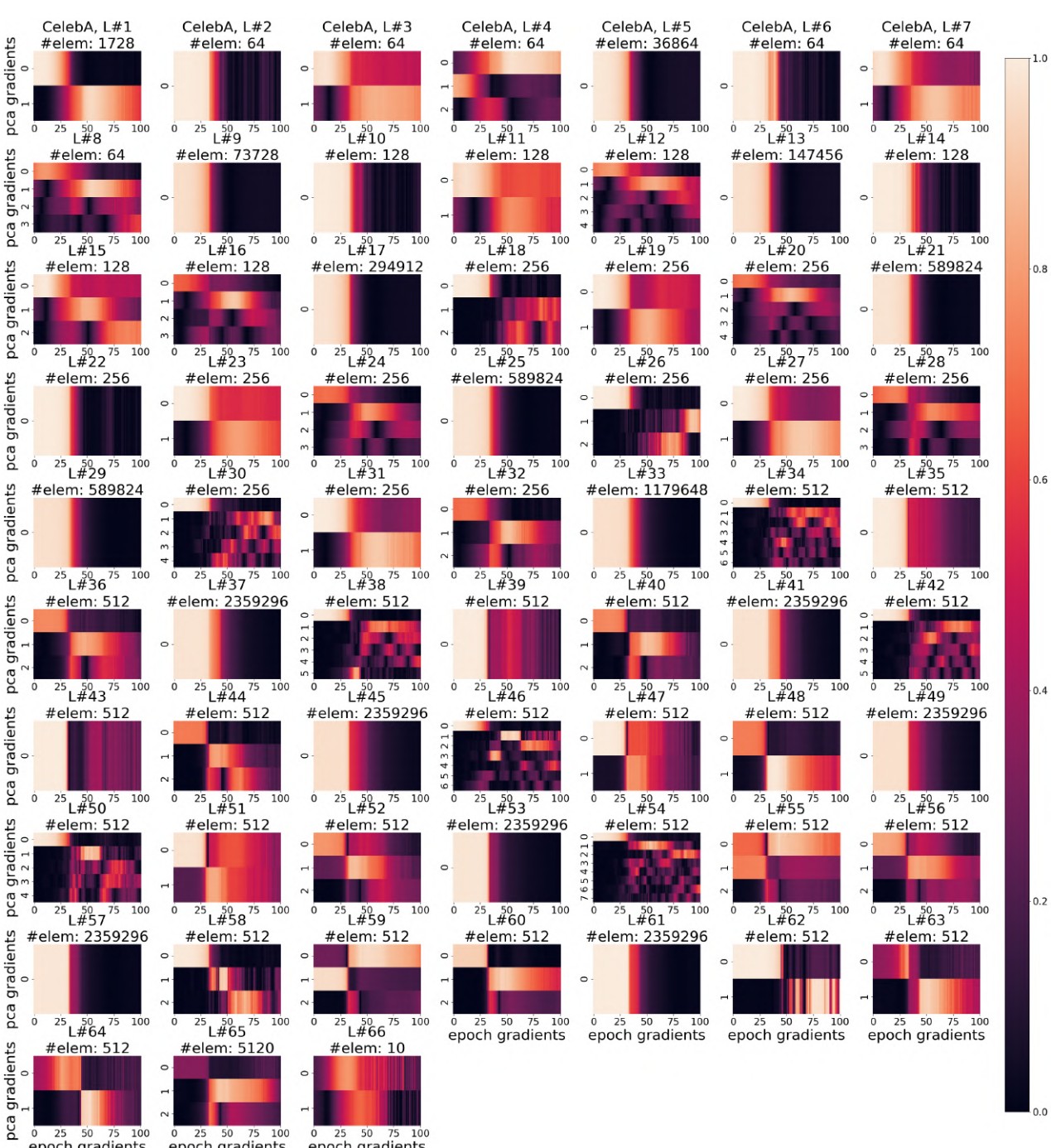

Figure 14: *PCA Components Overlap with Gradient.* Repeat of Fig. 2 on **VGG19** trained on **CelebA** dataset.

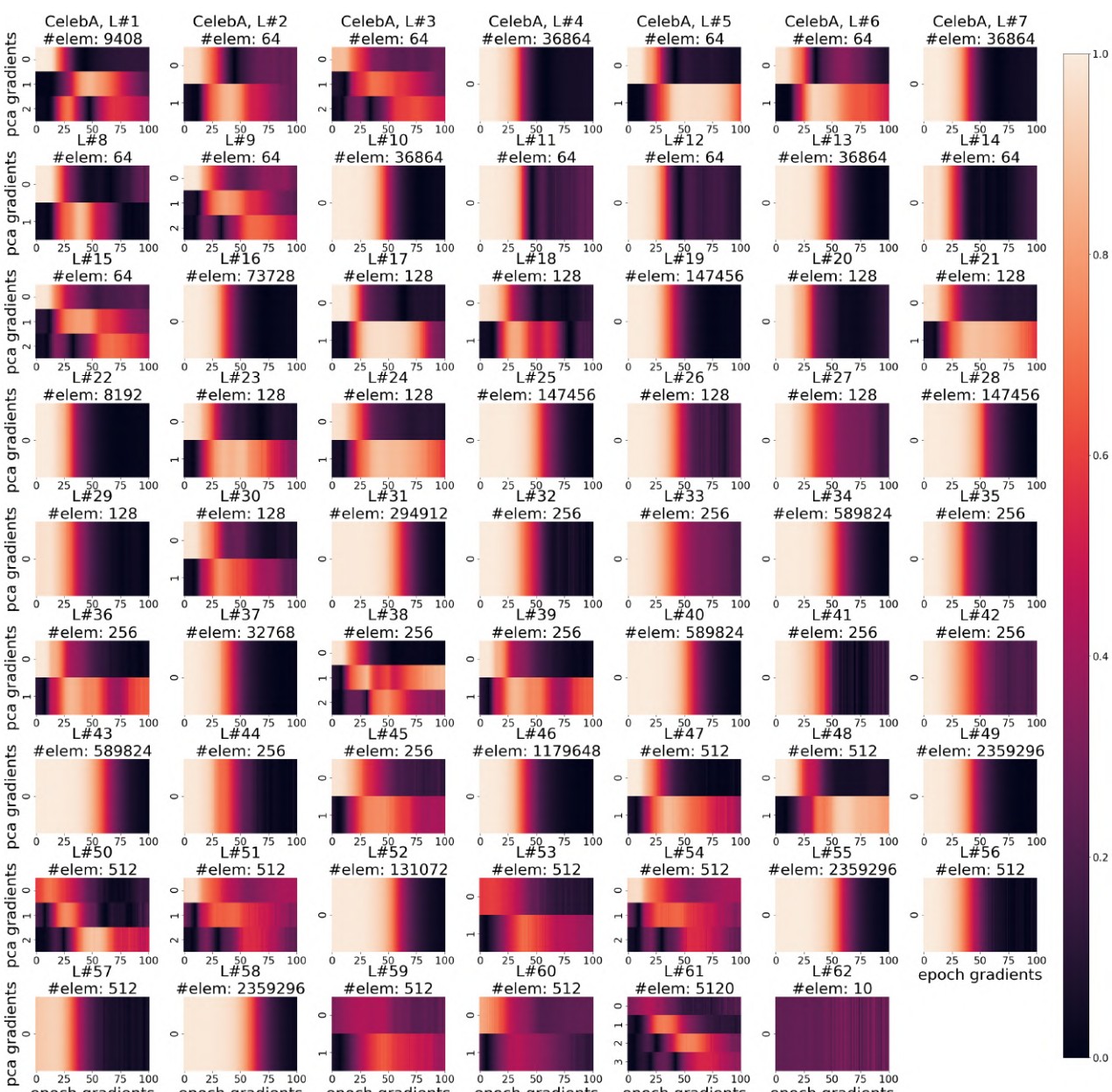

Figure 15: *PCA Components Overlap with Gradient.* Repeat of Fig. 2 on **ResNet18** trained on **CelebA** dataset.

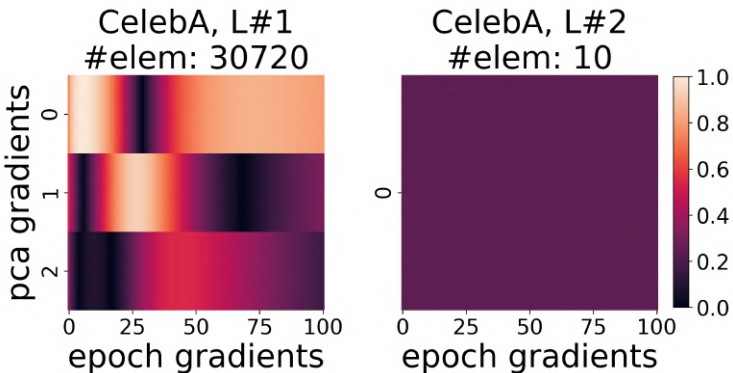

Figure 16: *PCA Components Overlap with Gradient.* Repeat of Fig. 2 on **FCN** trained on **CelebA** dataset.

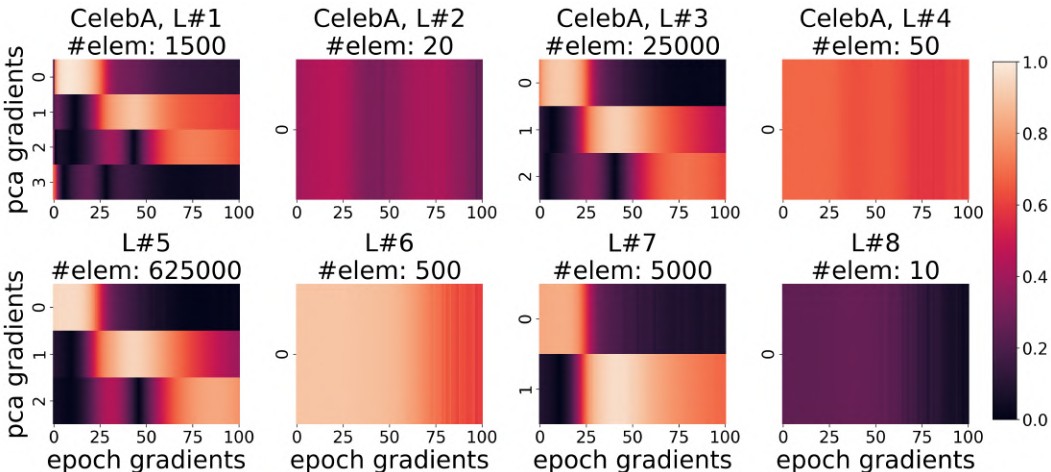

Figure 17: *PCA Components Overlap with Gradient.* Fig. 2 on **CNN** trained on **CelebA** dataset.

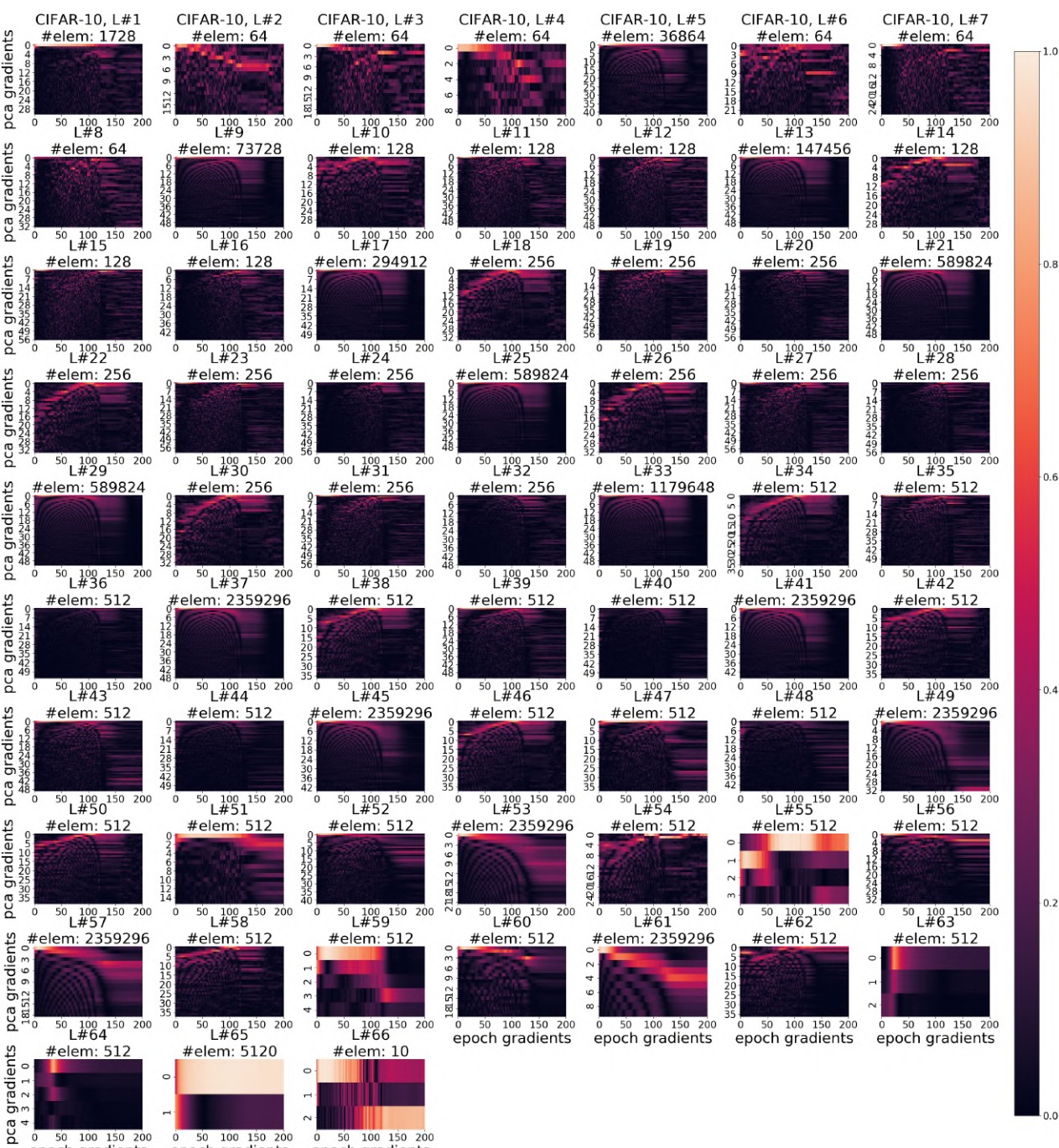

Figure 18: *PCA Components Overlap with Gradient*. Repeat of Fig. 2 on **VGG19** trained on **CIFAR-10** dataset.

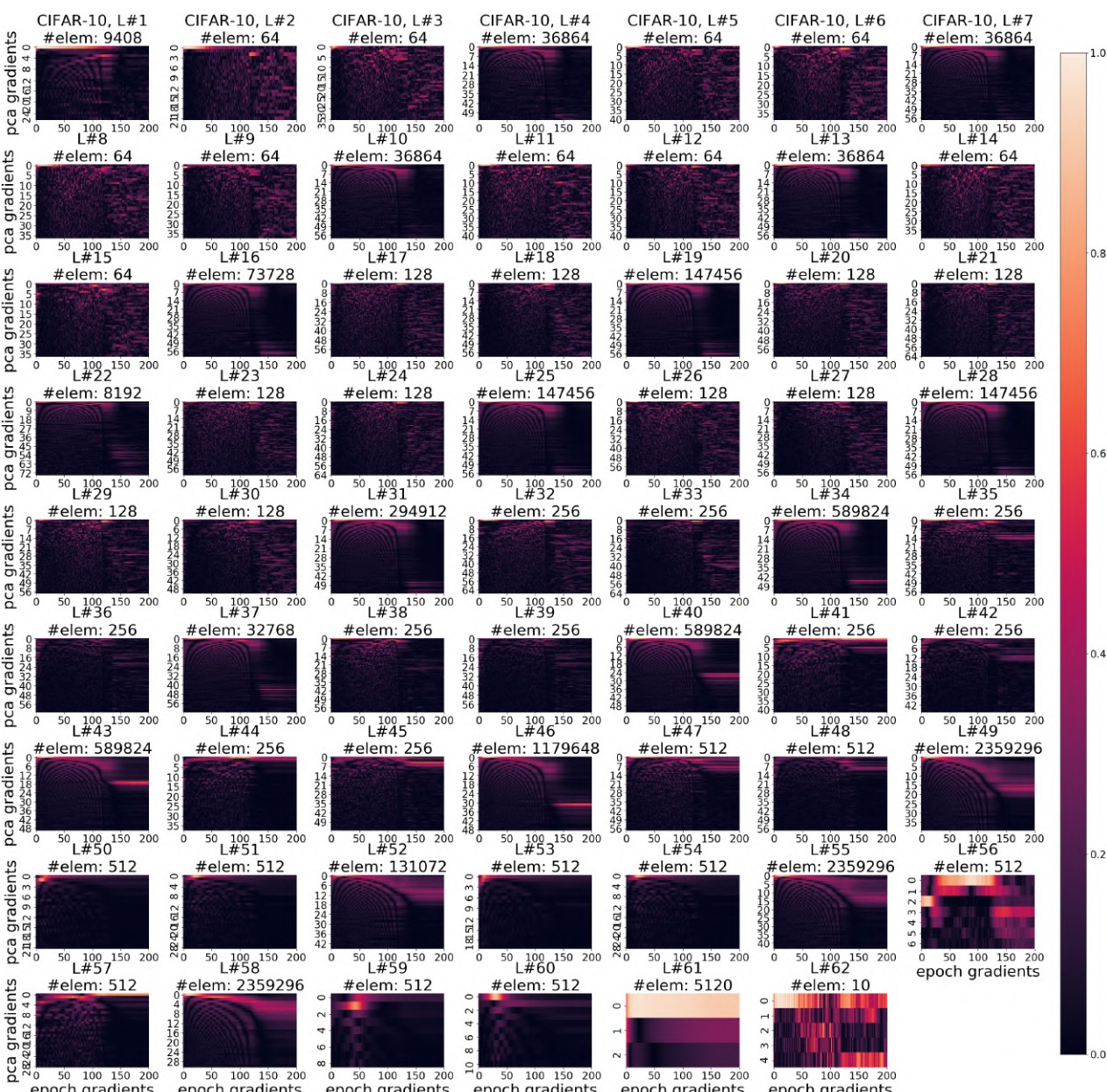

Figure 19: *PCA Components Overlap with Gradient.* Repeat of Fig. 2 on **ResNet18** trained on **CIFAR-10** dataset.

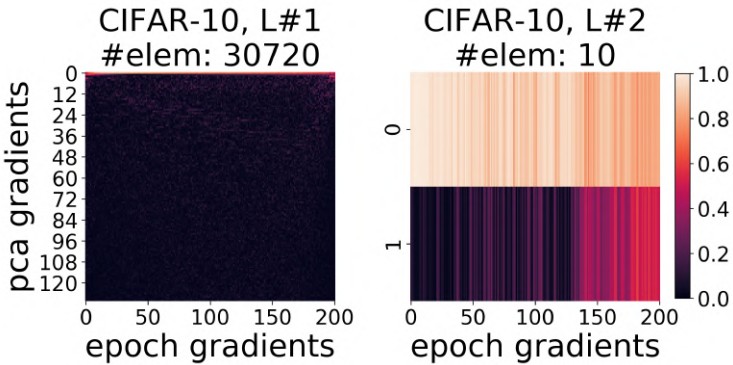

Figure 20: *PCA Components Overlap with Gradient*. Repeat of Fig. 2 on **FCN** trained on **CIFAR-10** dataset.

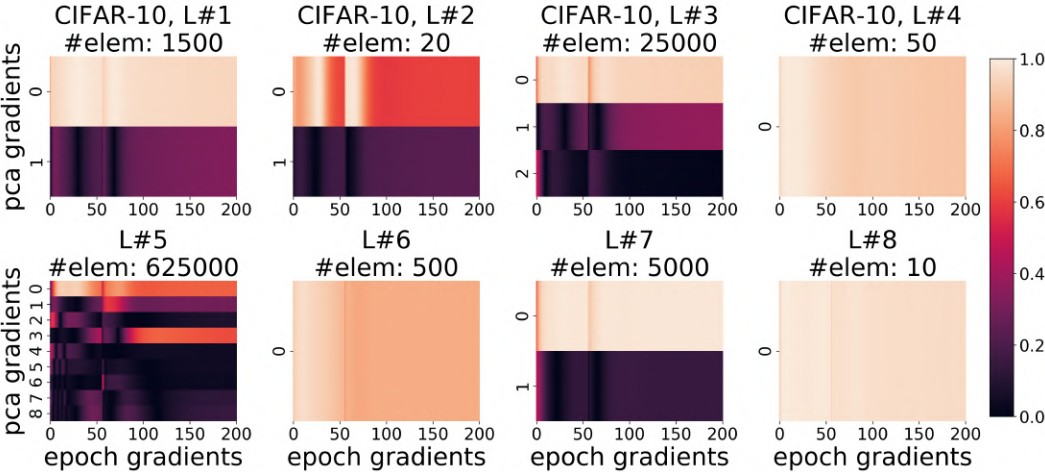

Figure 21: *PCA Components Overlap with Gradient*. Fig. 2 on **CNN** trained on **CIFAR-10** dataset.

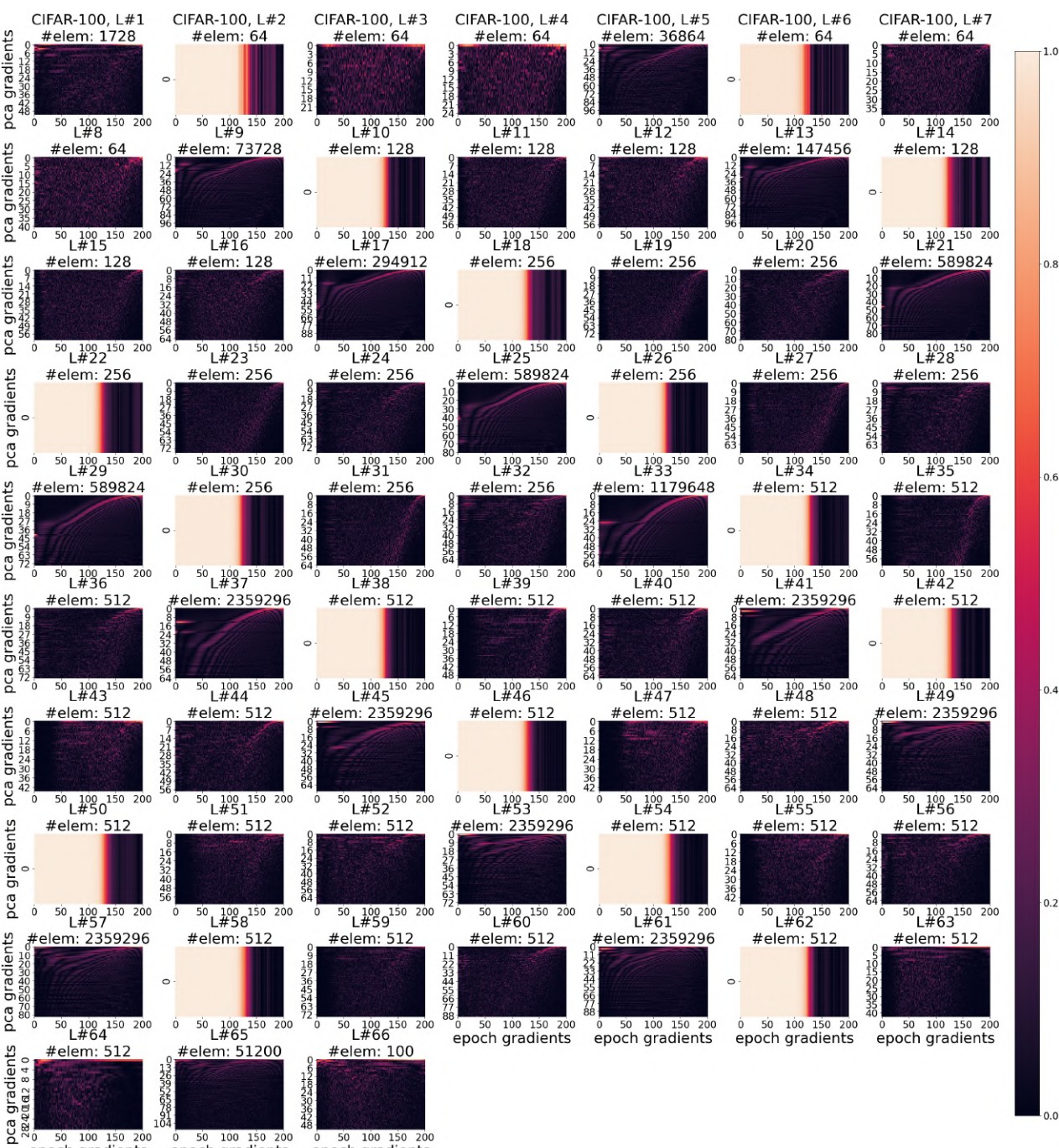

Figure 22: *PCA Components Overlap with Gradient.* Repeat of Fig. 2 on **VGG19** trained on **CIFAR-100** dataset.

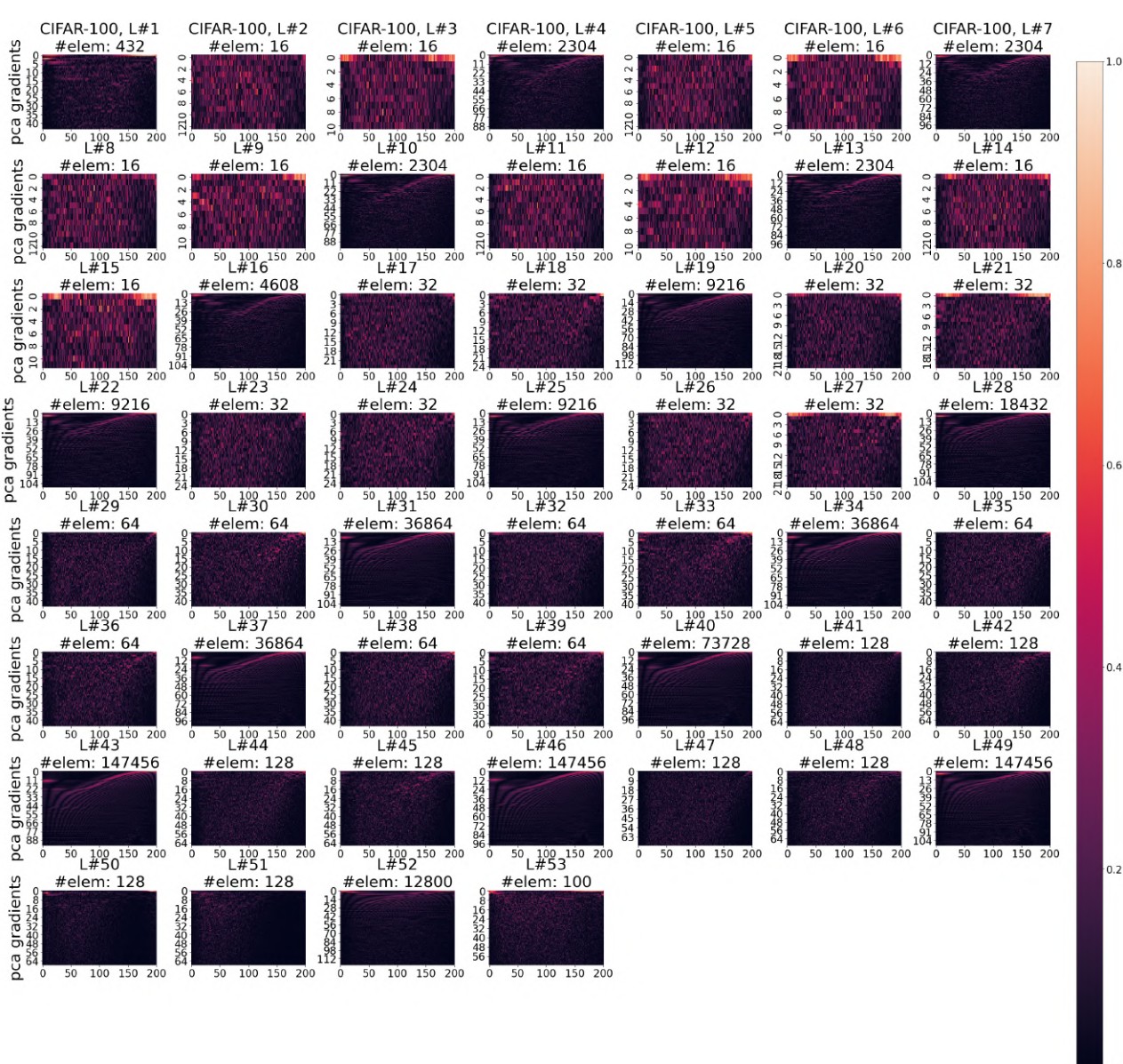

Figure 23: *PCA Components Overlap with Gradient.* Repeat of Fig. 2 on **ResNet18** trained on **CIFAR-100** dataset.

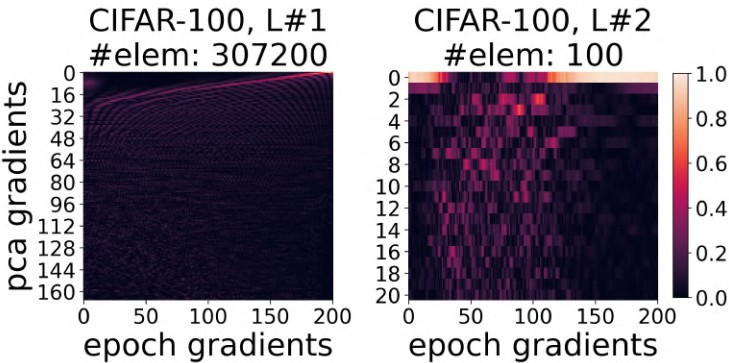

Figure 24: *PCA Components Overlap with Gradient*. Repeat of Fig. 2 on **FCN** trained on **CIFAR-100** dataset.

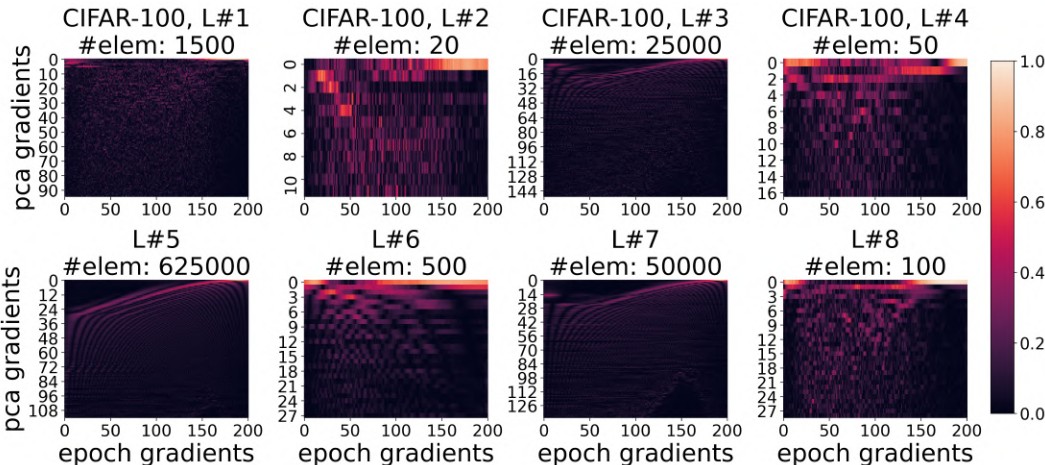

Figure 25: *PCA Components Overlap with Gradient*. Fig. 2 on **CNN** trained on **CIFAR-100** dataset.

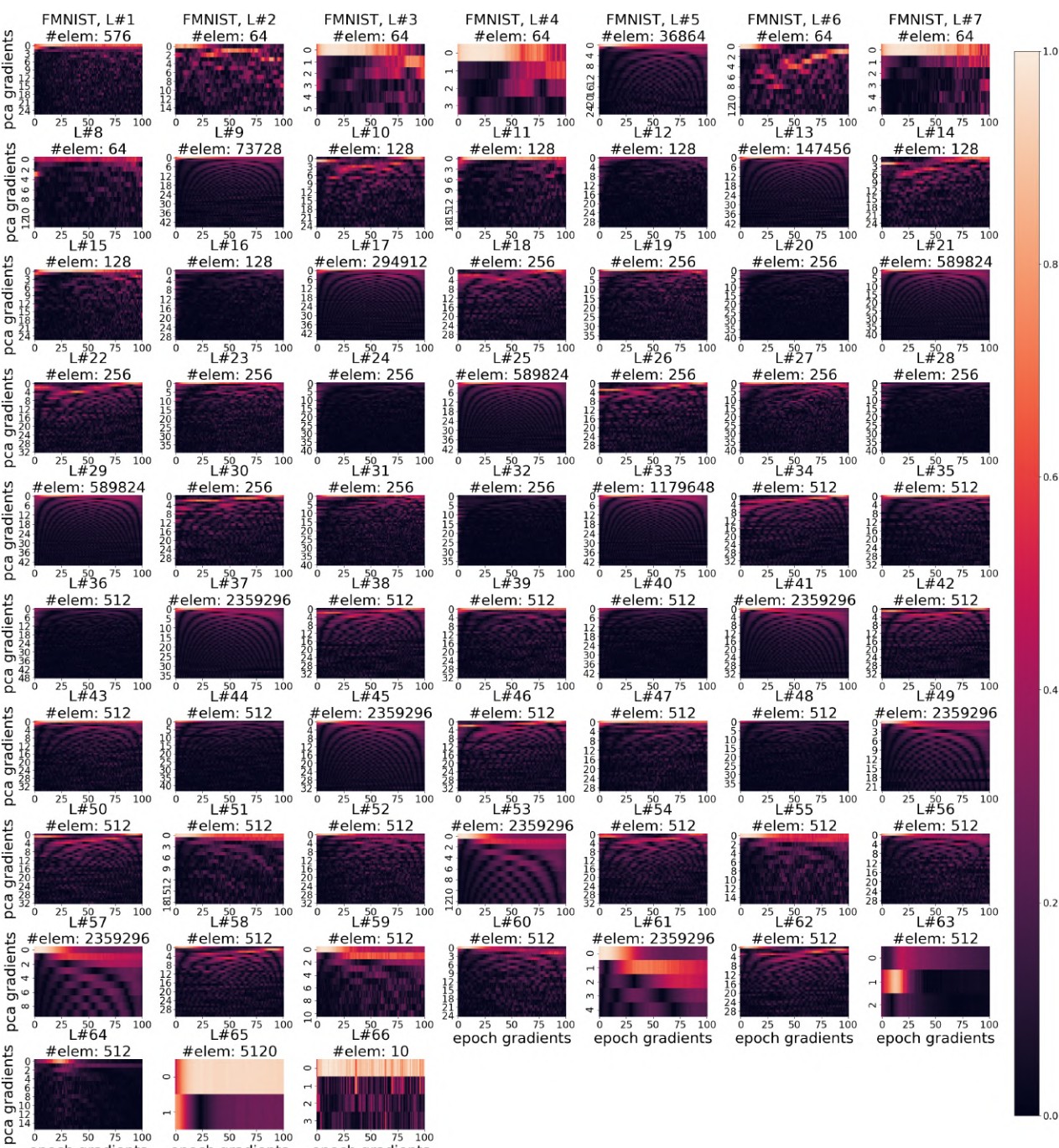

Figure 26: *PCA Components Overlap with Gradient.* Repeat of Fig. 2 on **VGG19** trained on **FMNIST** dataset.

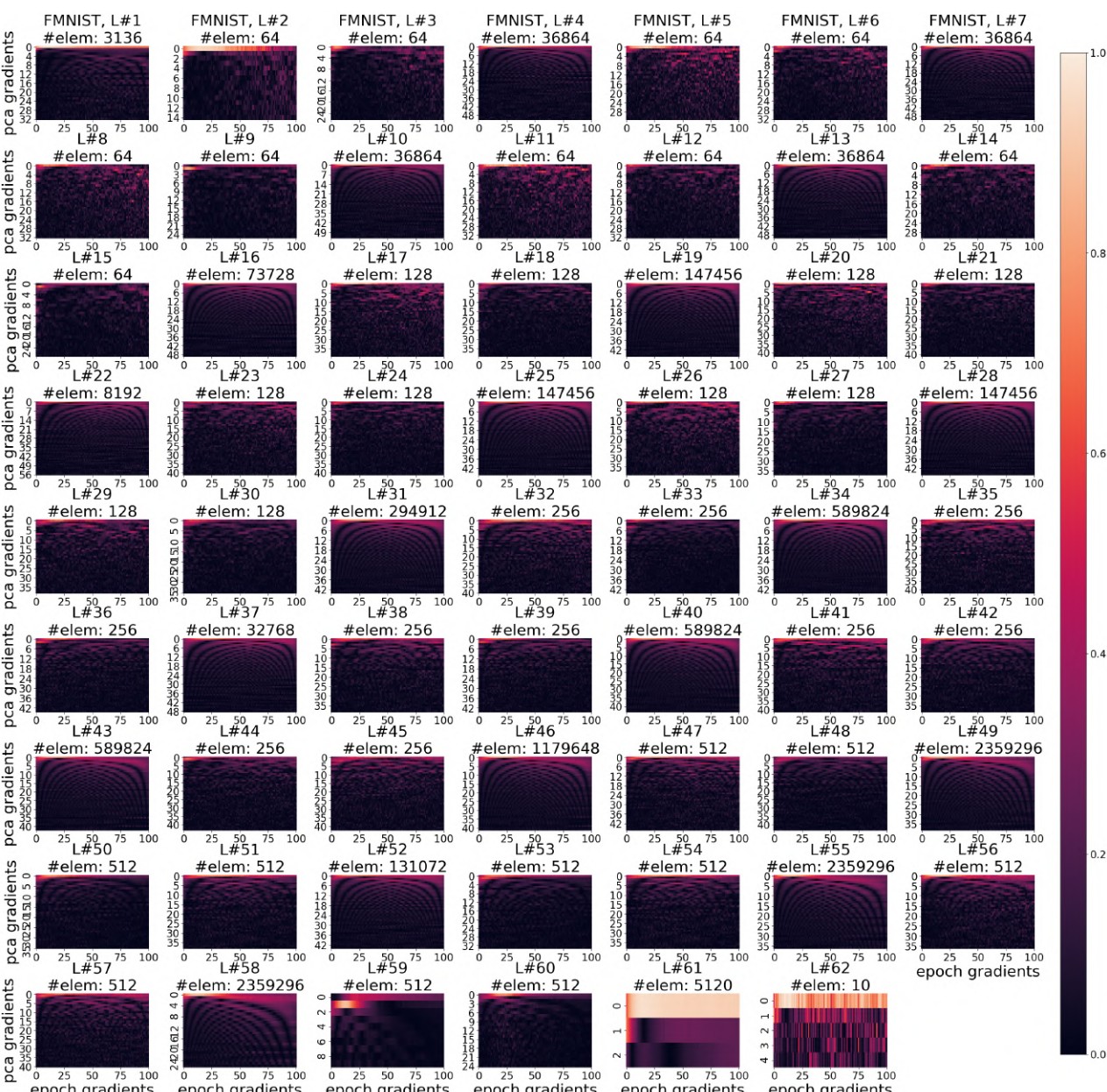

Figure 27: *PCA Components Overlap with Gradient.* Repeat of Fig. 2 on **ResNet18** trained on **FMNIST** dataset.

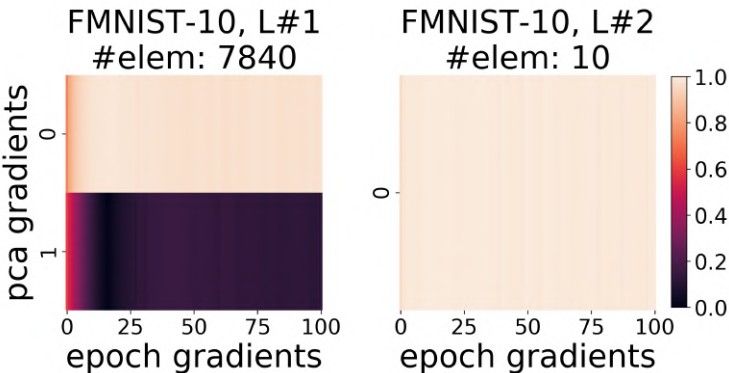

Figure 28: *PCA Components Overlap with Gradient.* Repeat of Fig. 2 on **FCN** trained on **FMNIST** dataset.

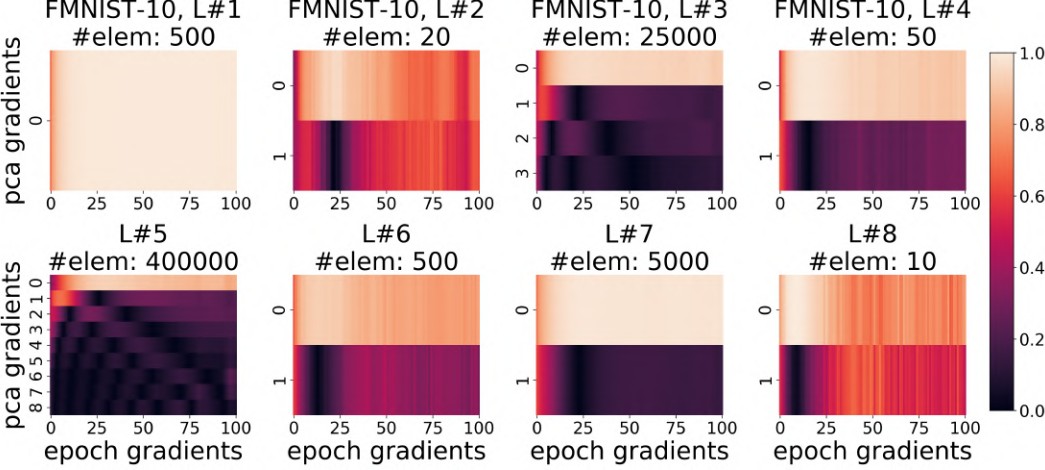

Figure 29: *PCA Components Overlap with Gradient.* Fig. 2 on **CNN** trained on **FMNIST** dataset.

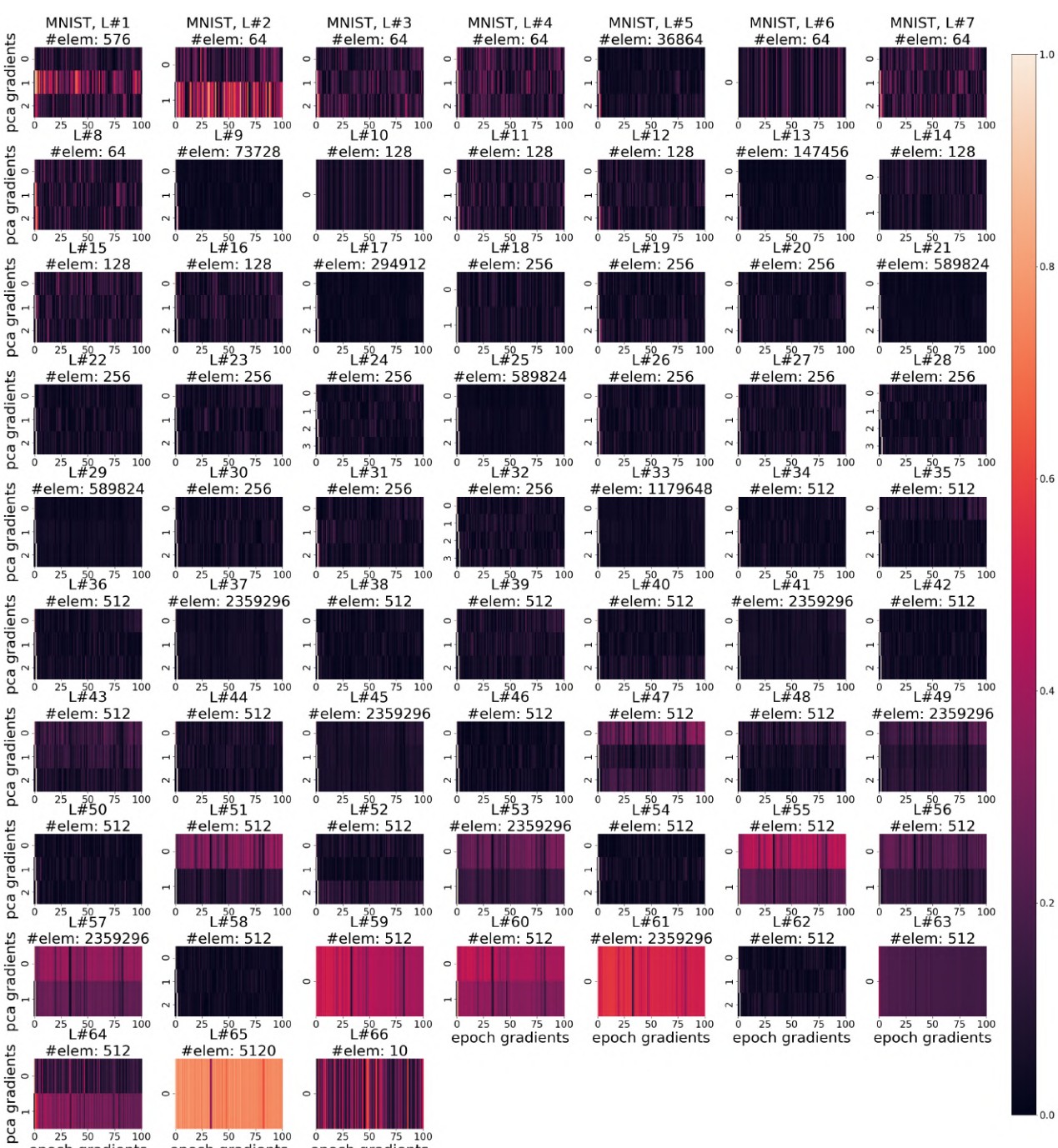

Figure 30: *PCA Components Overlap with Gradient.* Repeat of Fig. 2 on **VGG19** trained on **MNIST** dataset.

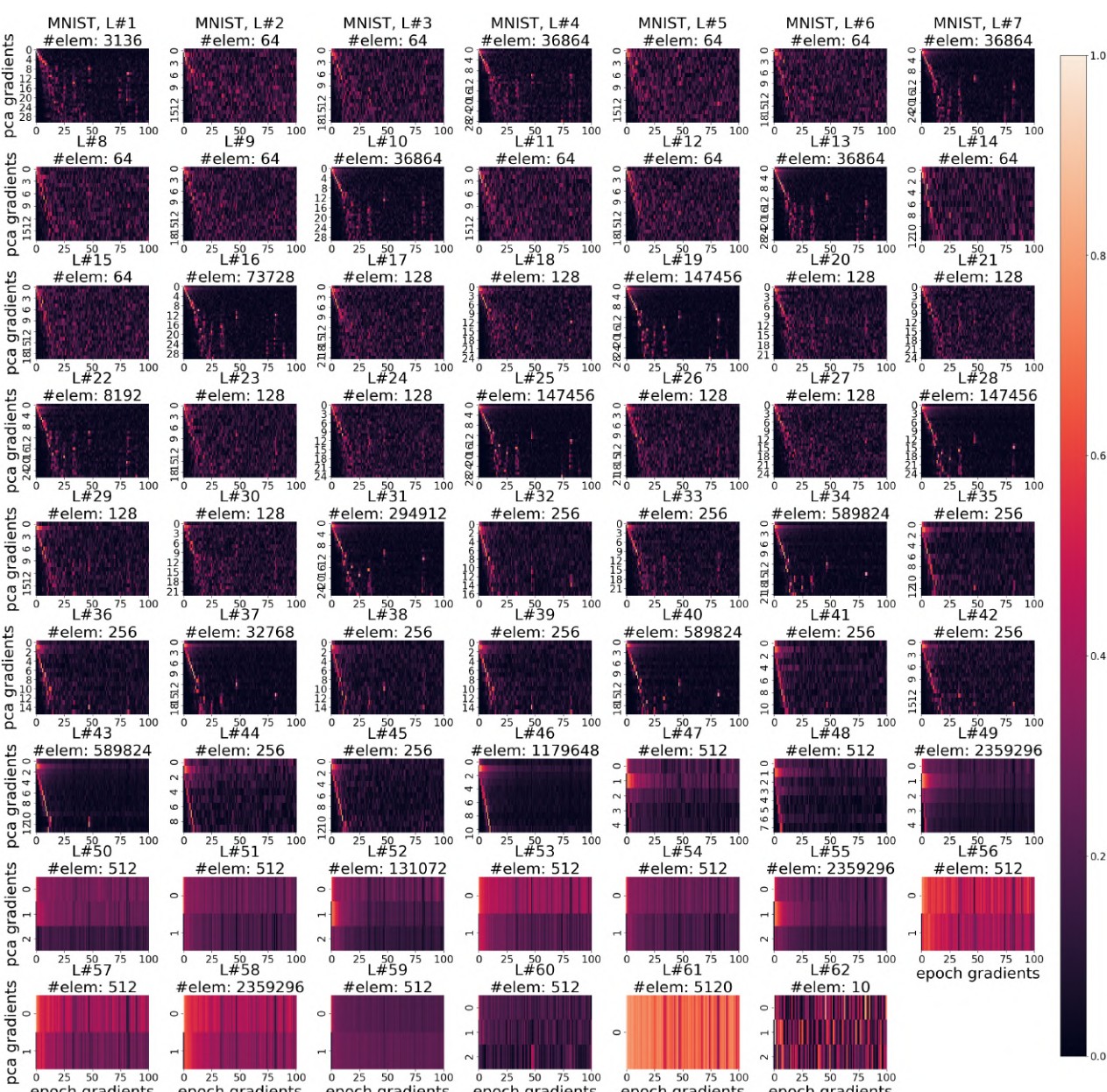

Figure 31: *PCA Components Overlap with Gradient*. Repeat of Fig. 2 on **ResNet18** trained on **MNIST** dataset.

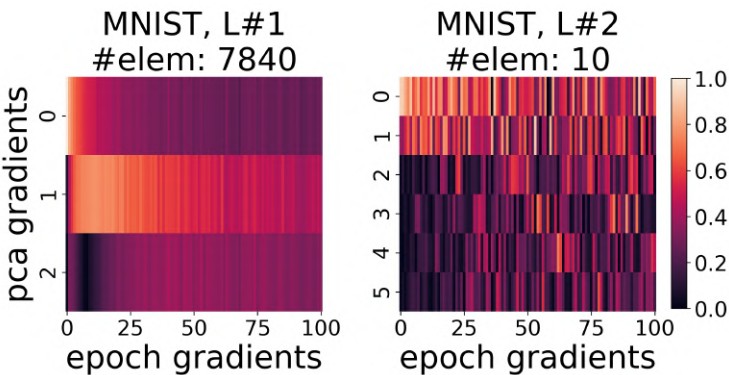

Figure 32: *PCA Components Overlap with Gradient*. Repeat of Fig. 2 on **FCN** trained on **MNIST** dataset.

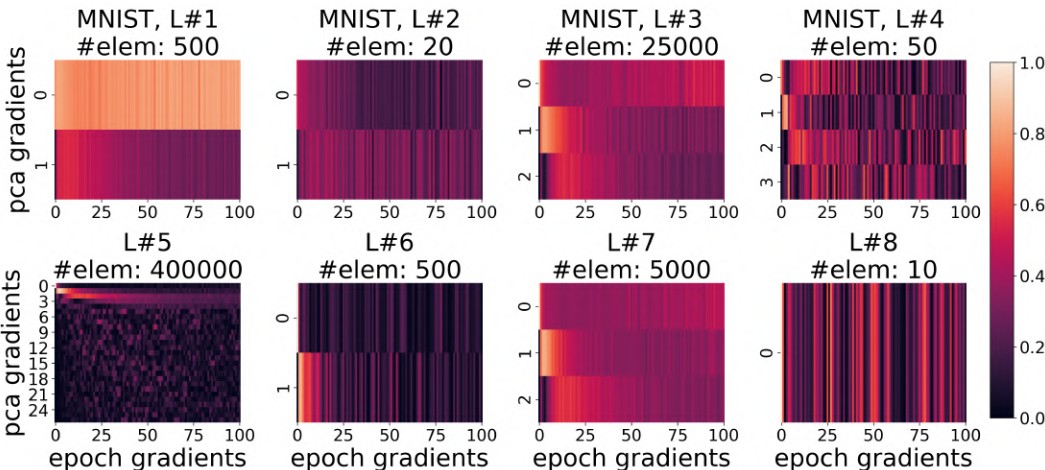

Figure 33: *PCA Components Overlap with Gradient*. Fig. 2 on **CNN** trained on **MNIST** dataset.

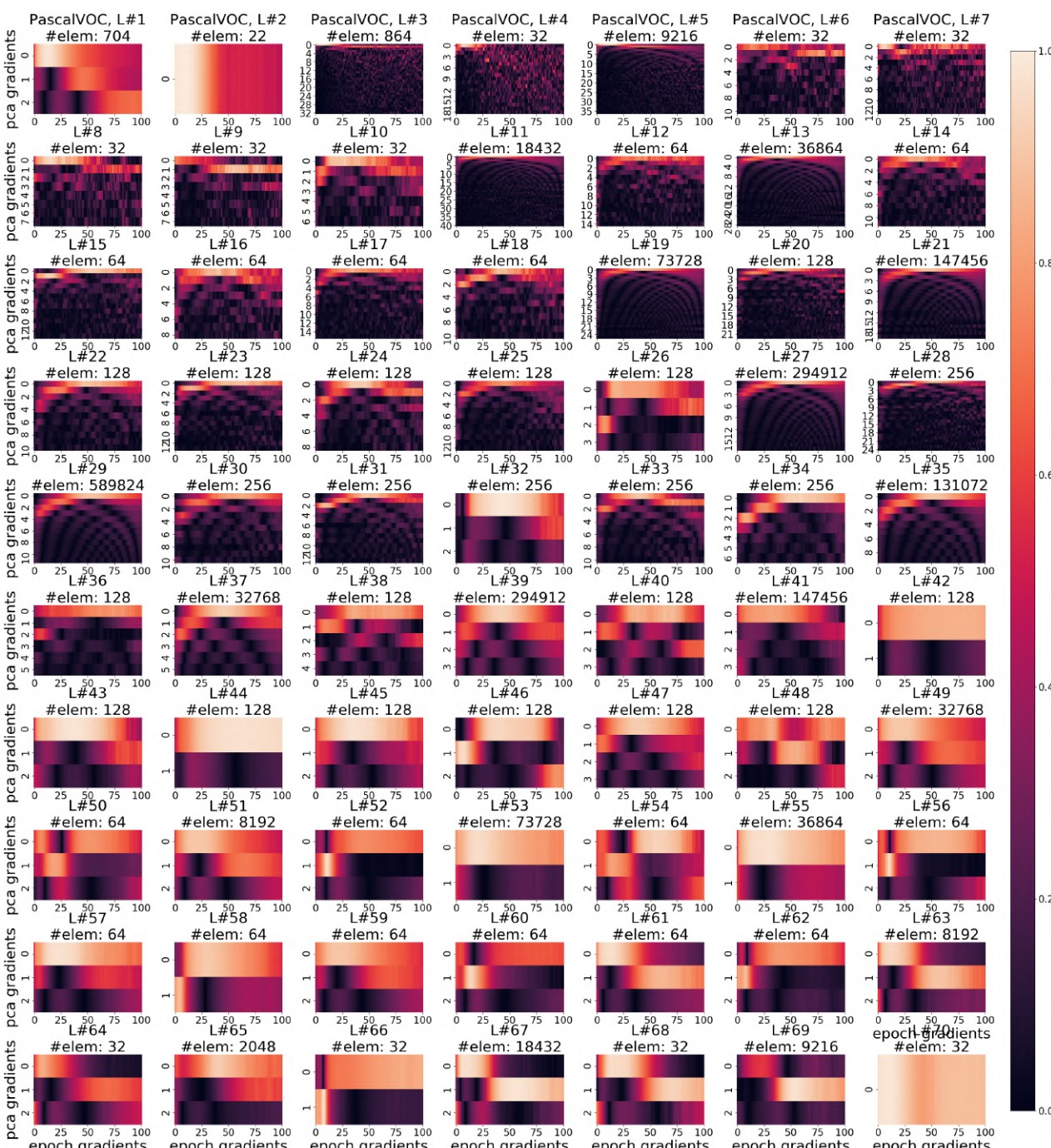

Figure 34: *PCA Components Overlap with Gradient*. Repeat of Fig. 2 on **U-Net** trained on **PascalVOC** dataset.

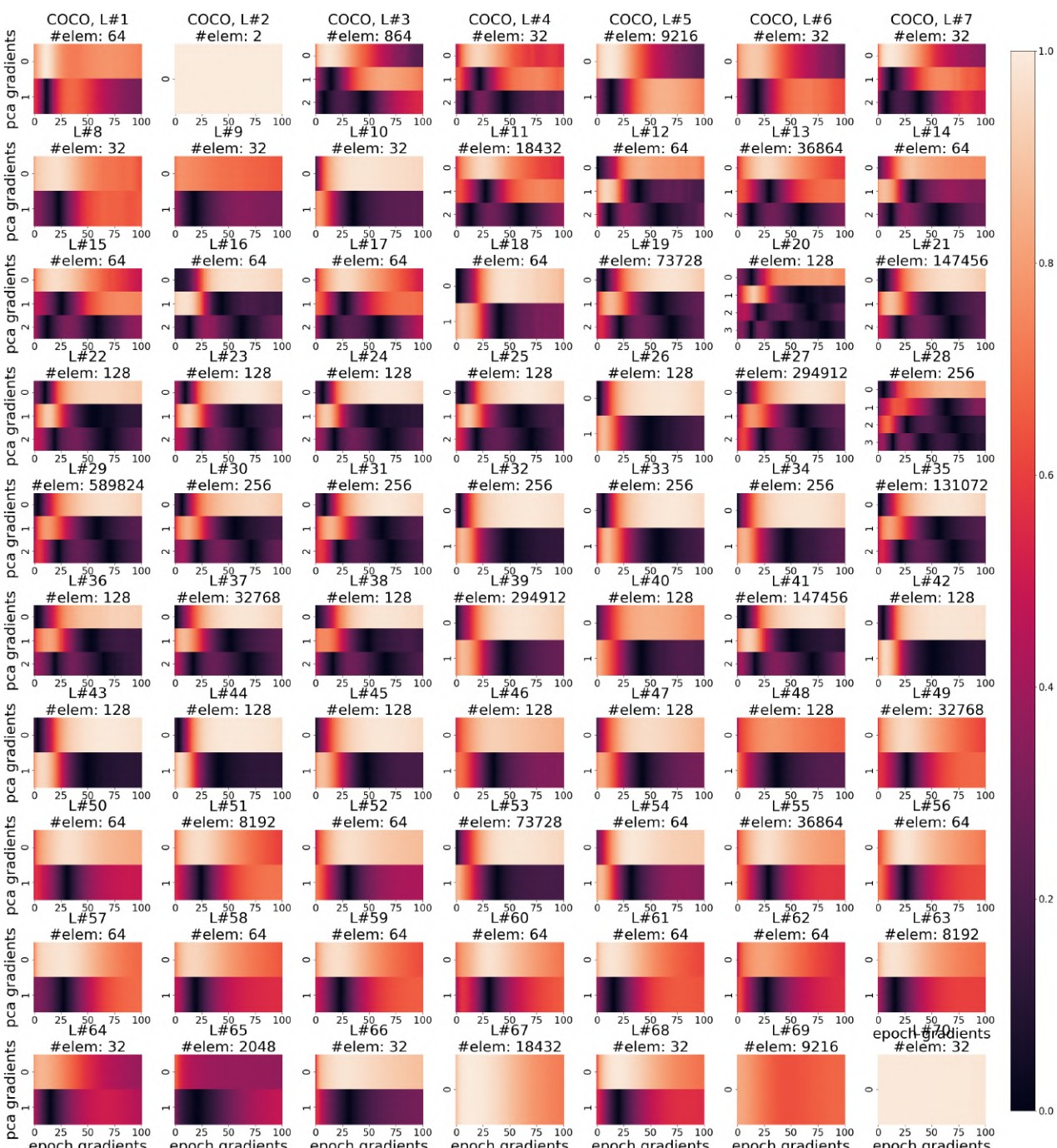

Figure 35: *PCA Components Overlap with Gradient*. Repeat of Fig. 2 on **U-Net** trained on **COCO** dataset.

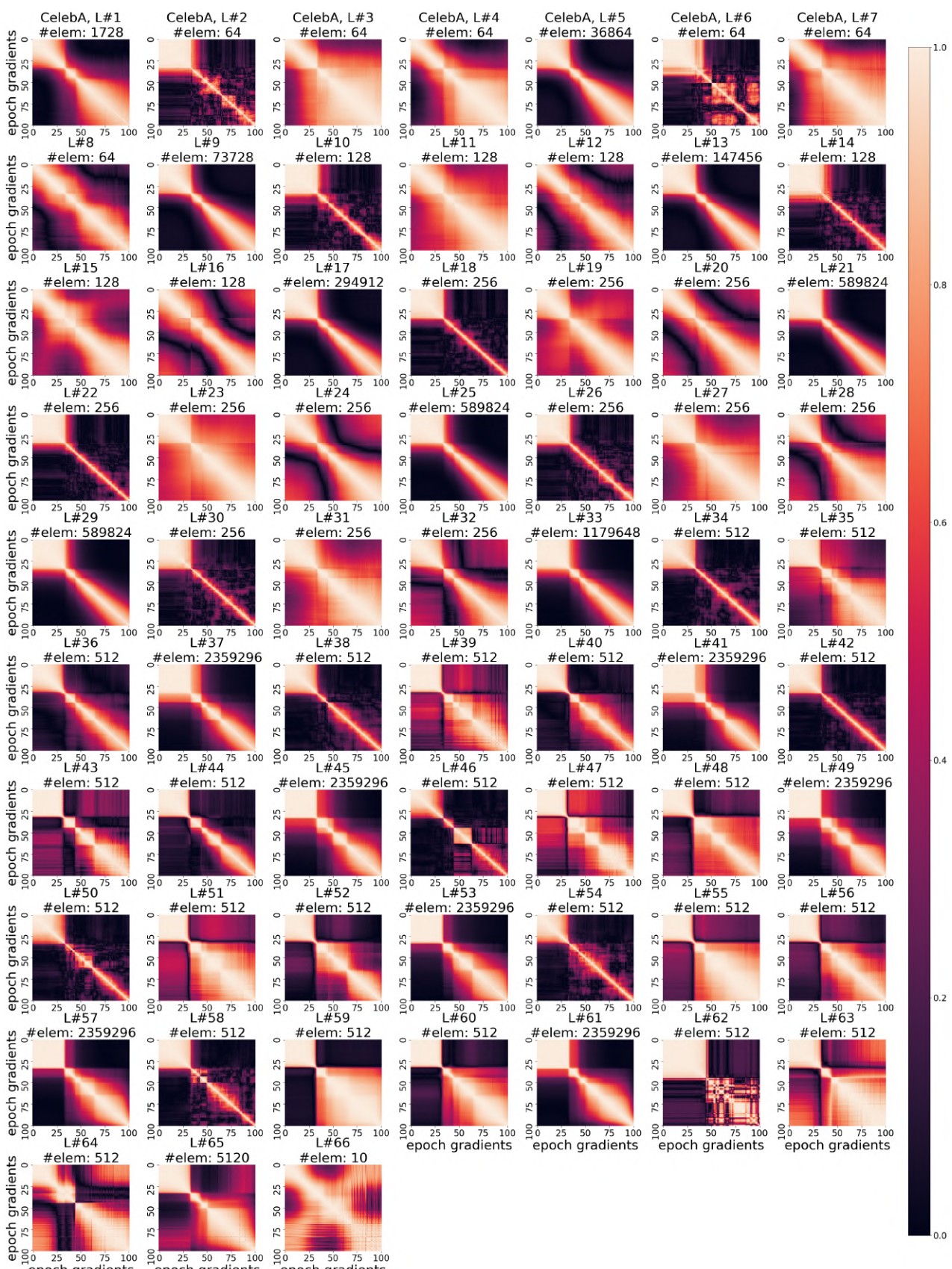

Figure 36: *PCA Components Overlap with Gradient*. Repeat of Fig. 3 on **VGG19** trained on **CelebA** dataset.

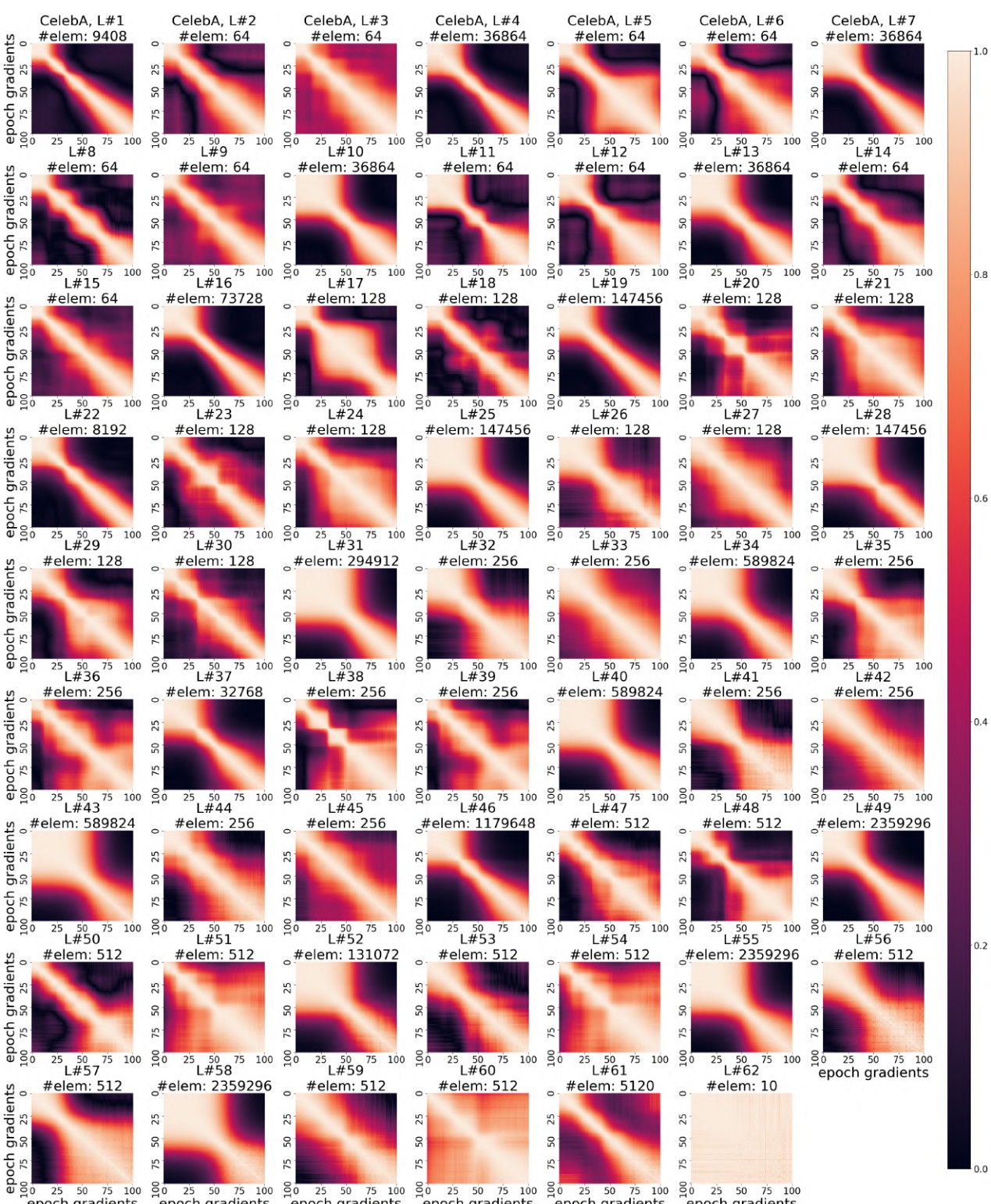

Figure 37: *PCA Components Overlap with Gradient.* Repeat of Fig. 3 on **ResNet18** trained on **CelebA** dataset.

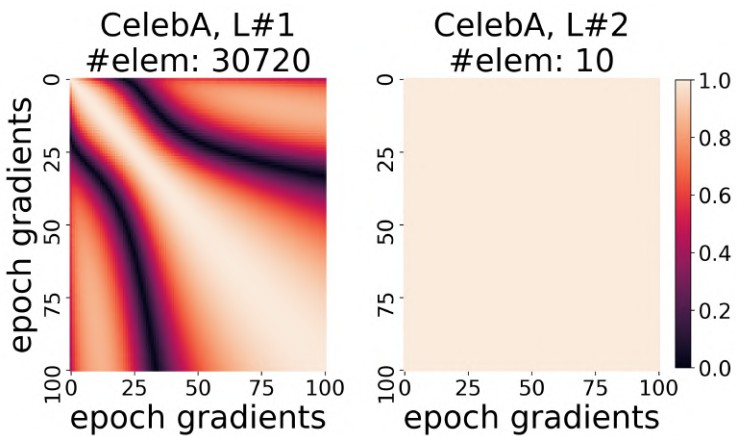

Figure 38: *PCA Components Overlap with Gradient*. Repeat of Fig. 3 on **FCN** trained on **CelebA** dataset.

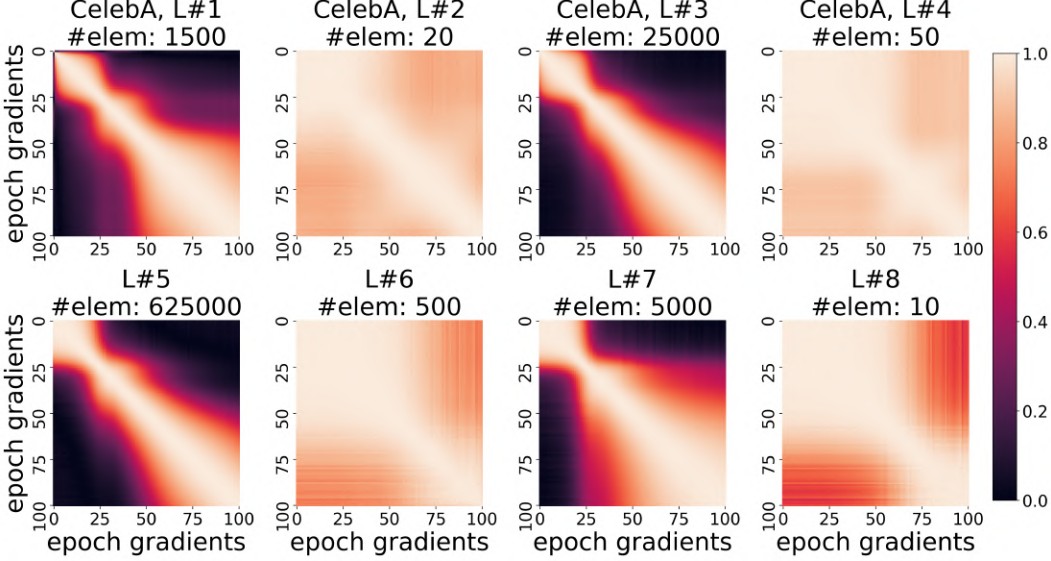

Figure 39: *PCA Components Overlap with Gradient*. Fig. 3 on **CNN** trained on **CelebA** dataset.

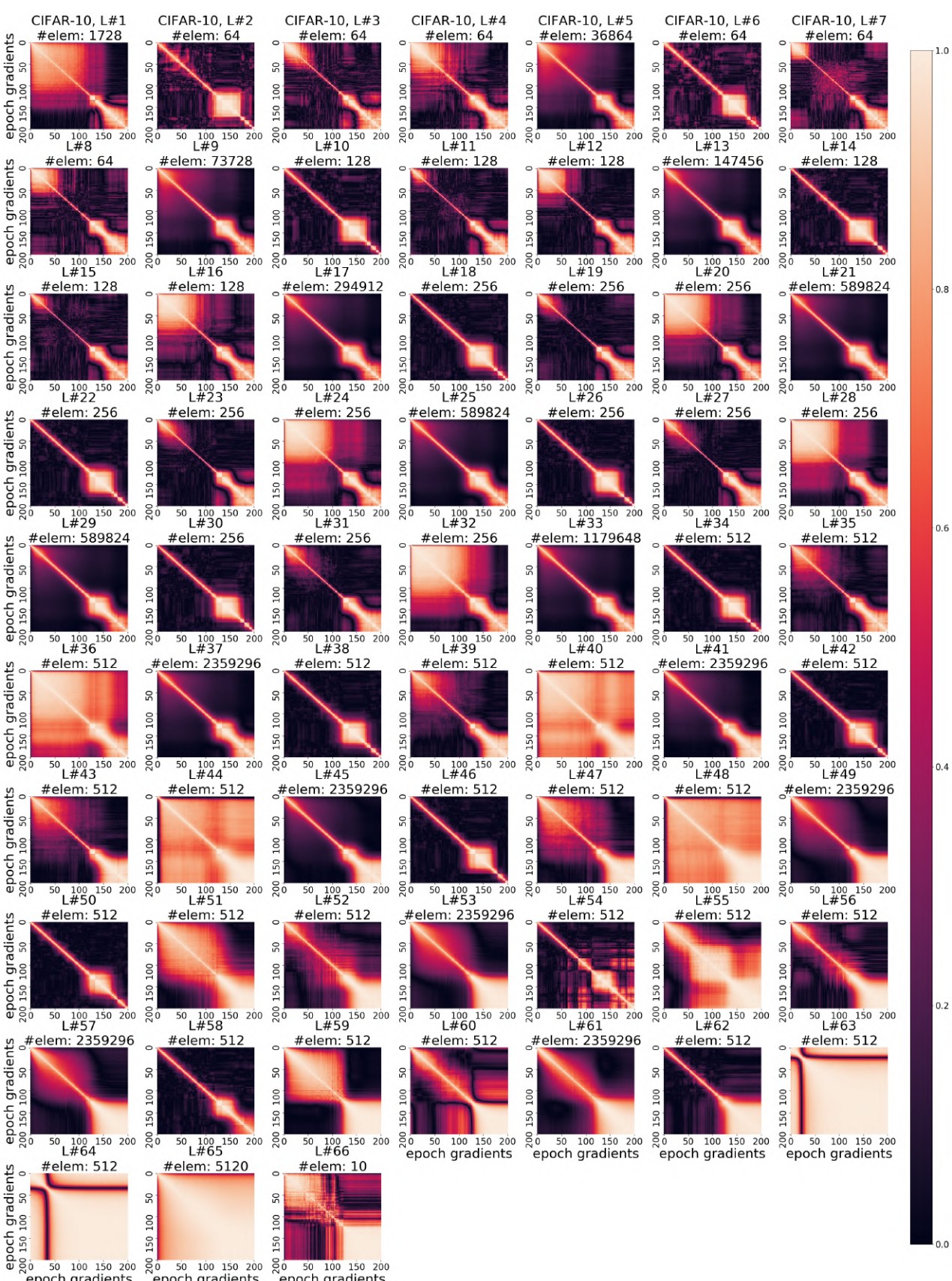

Figure 40: *PCA Components Overlap with Gradient*. Repeat of Fig. 3 on **VGG19** trained on **CIFAR-10** dataset.

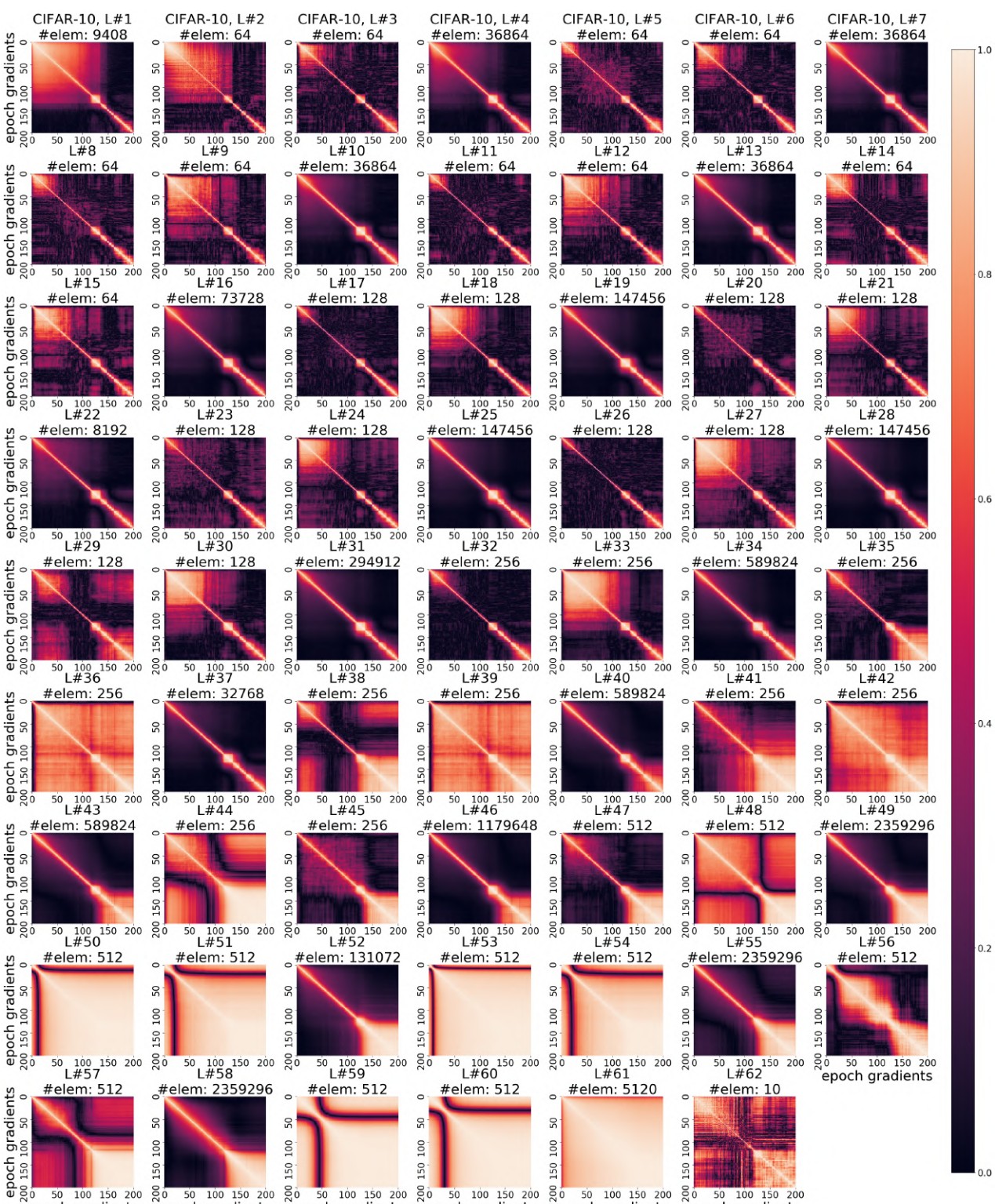

Figure 41: *PCA Components Overlap with Gradient.* Repeat of Fig. 3 on **ResNet18** trained on **CIFAR-10** dataset.

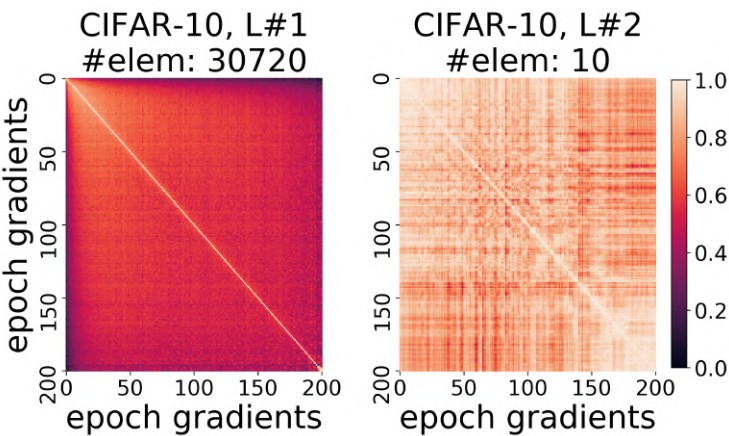

Figure 42: *PCA Components Overlap with Gradient*. Repeat of Fig. 3 on **FCN** trained on **CIFAR-10** dataset.

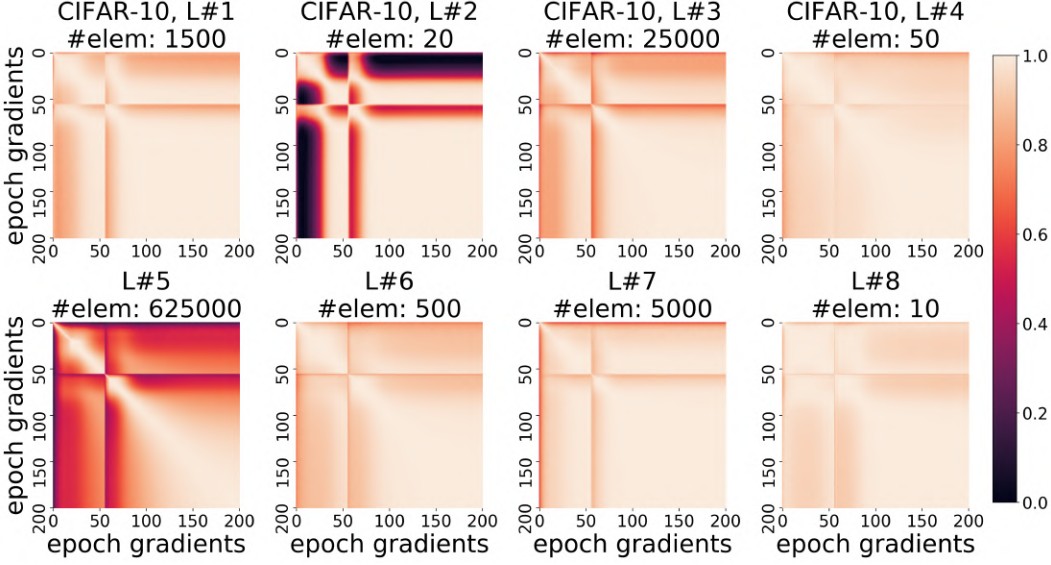

Figure 43: *PCA Components Overlap with Gradient*. Fig. 3 on **CNN** trained on **CIFAR-10** dataset.

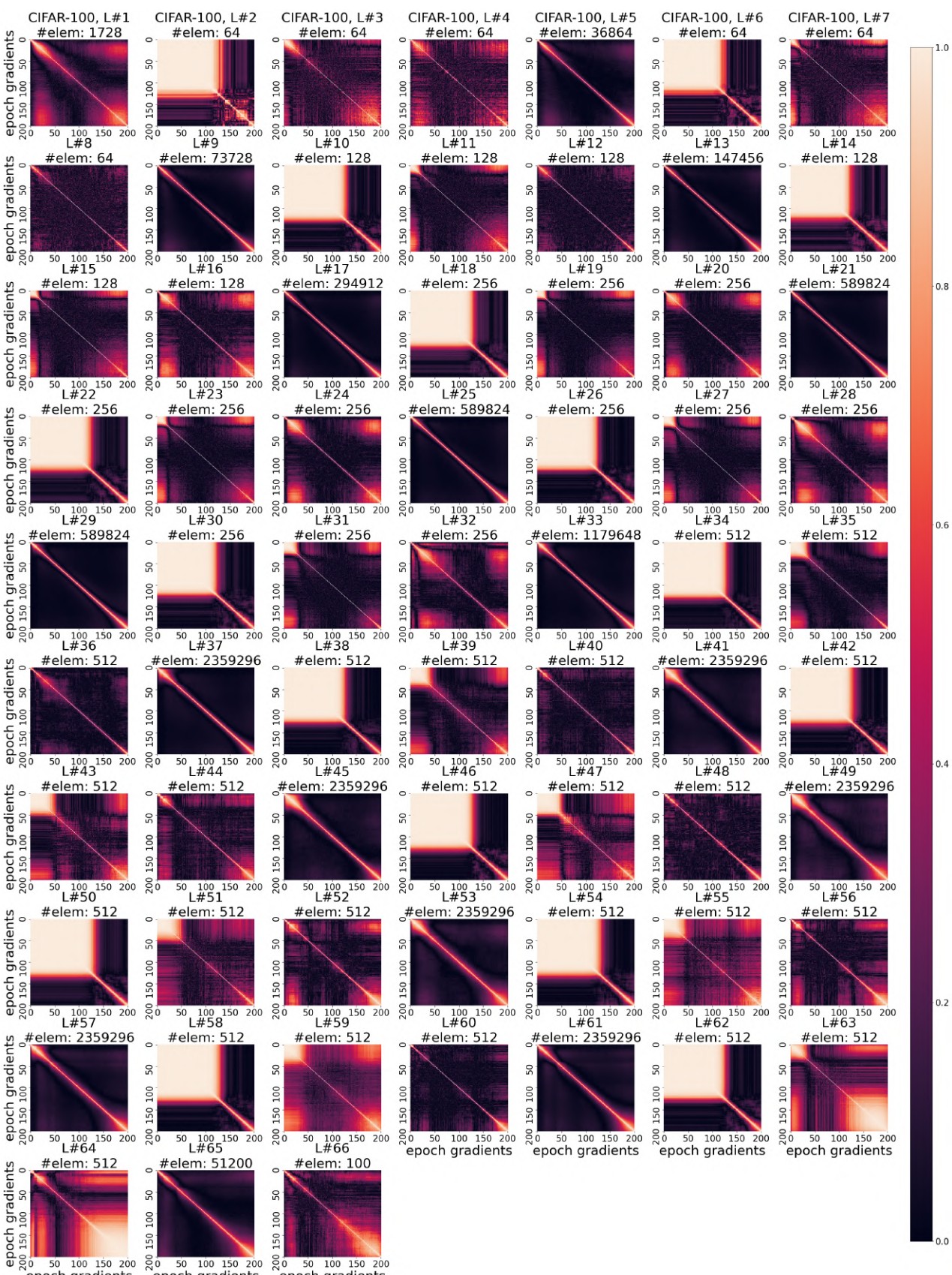

Figure 44: *PCA Components Overlap with Gradient.* Repeat of Fig. 3 on **VGG19** trained on **CIFAR-100** dataset.

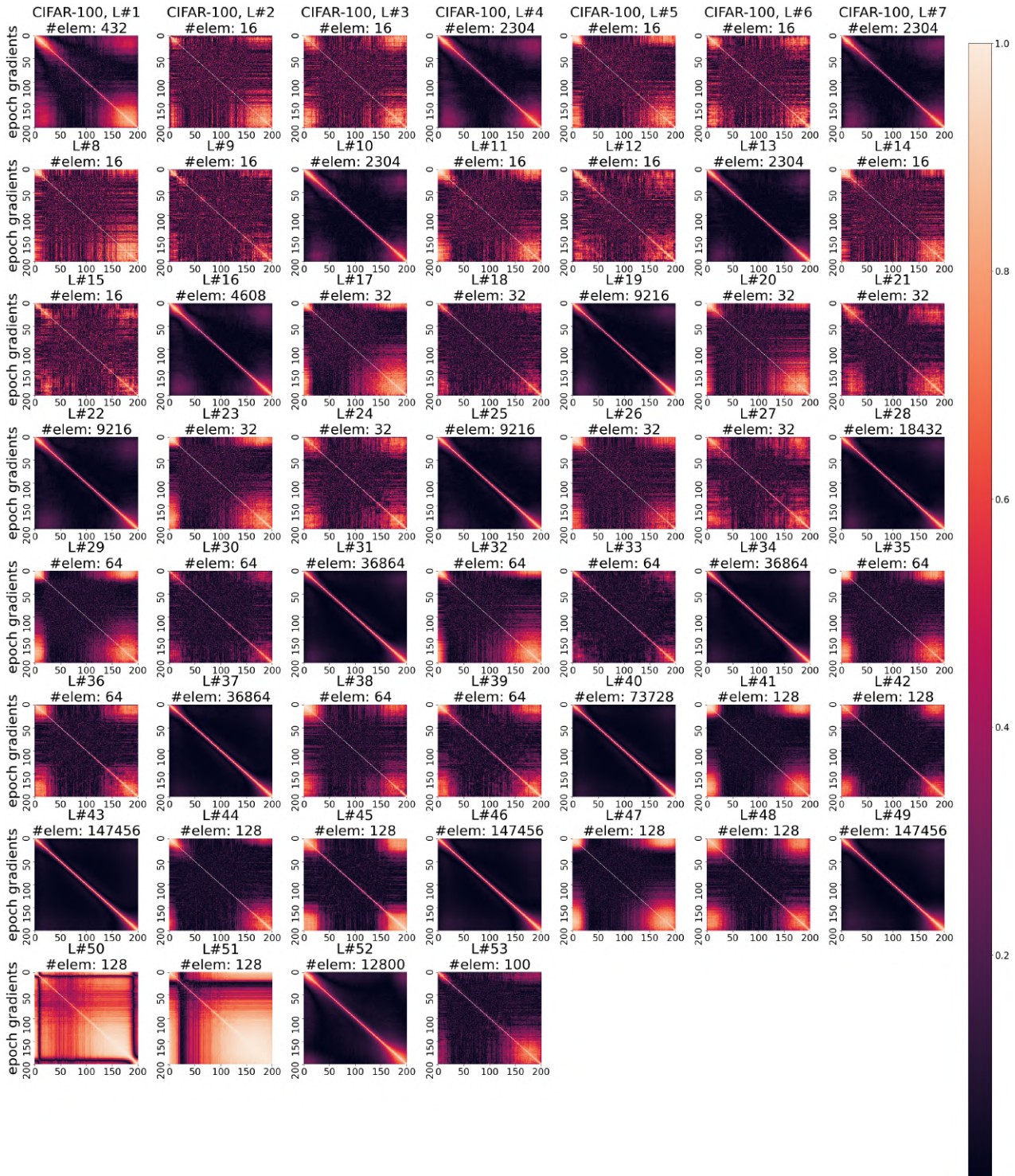

Figure 45: *PCA Components Overlap with Gradient.* Repeat of Fig. 3 on **ResNet18** trained on **CIFAR-100** dataset.

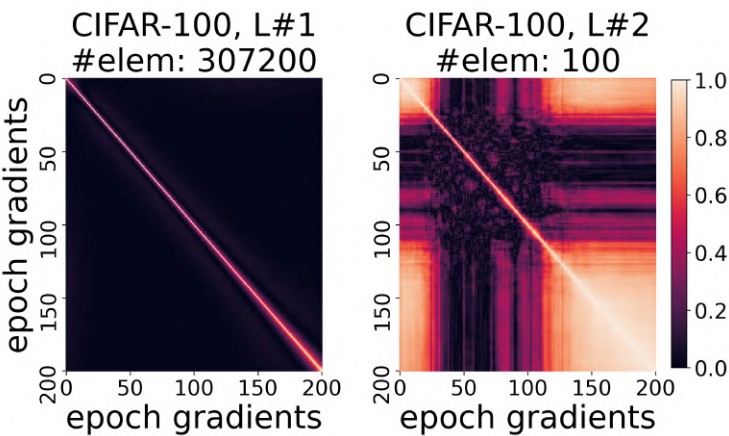

Figure 46: *PCA Components Overlap with Gradient*. Repeat of Fig. 3 on **FCN** trained on **CIFAR-100** dataset.

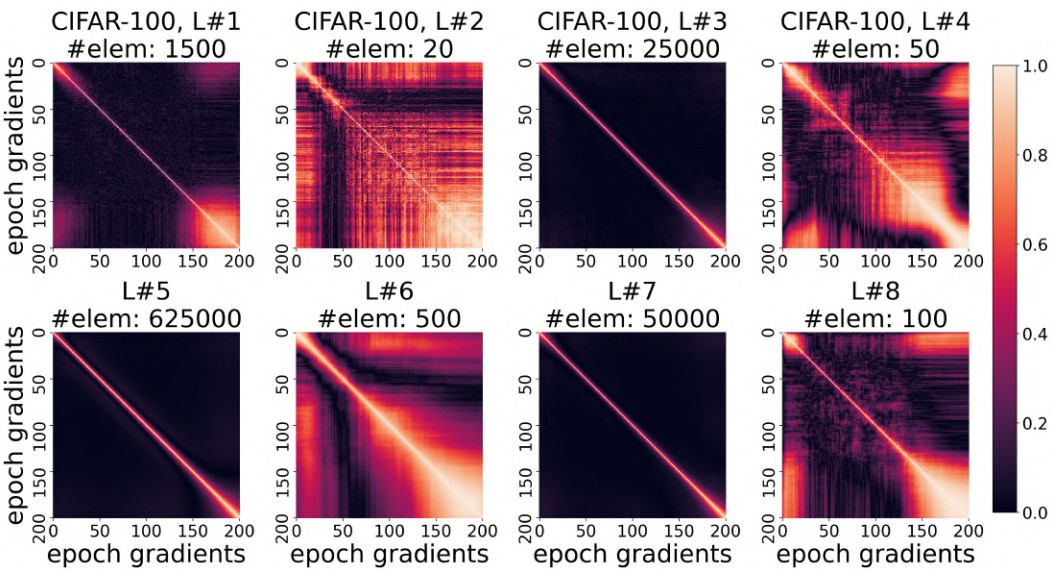

Figure 47: *PCA Components Overlap with Gradient*. Fig. 3 on **CNN** trained on **CIFAR-100** dataset.

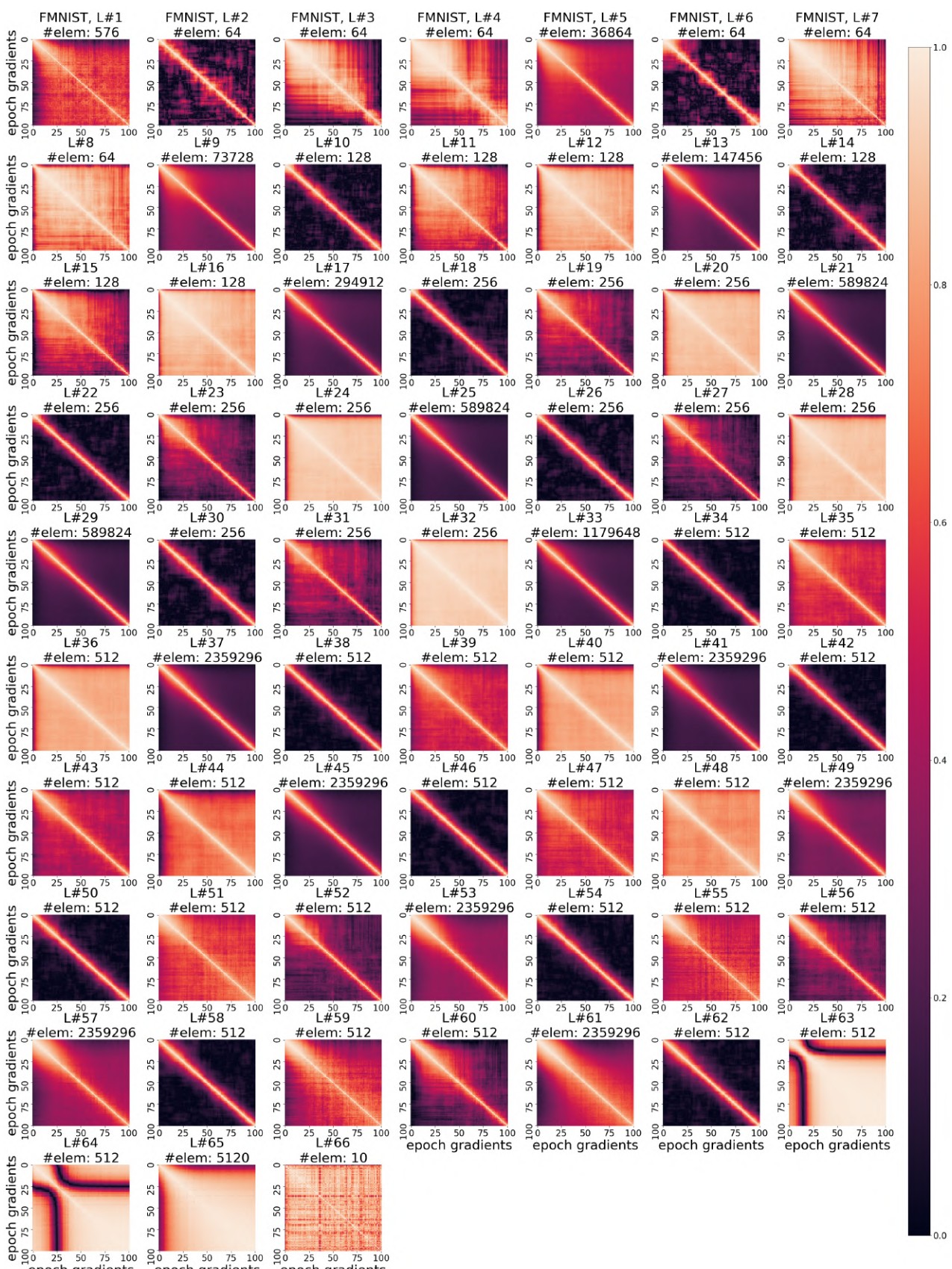

Figure 48: *PCA Components Overlap with Gradient*. Repeat of Fig. 3 on **VGG19** trained on **FMNIST** dataset.

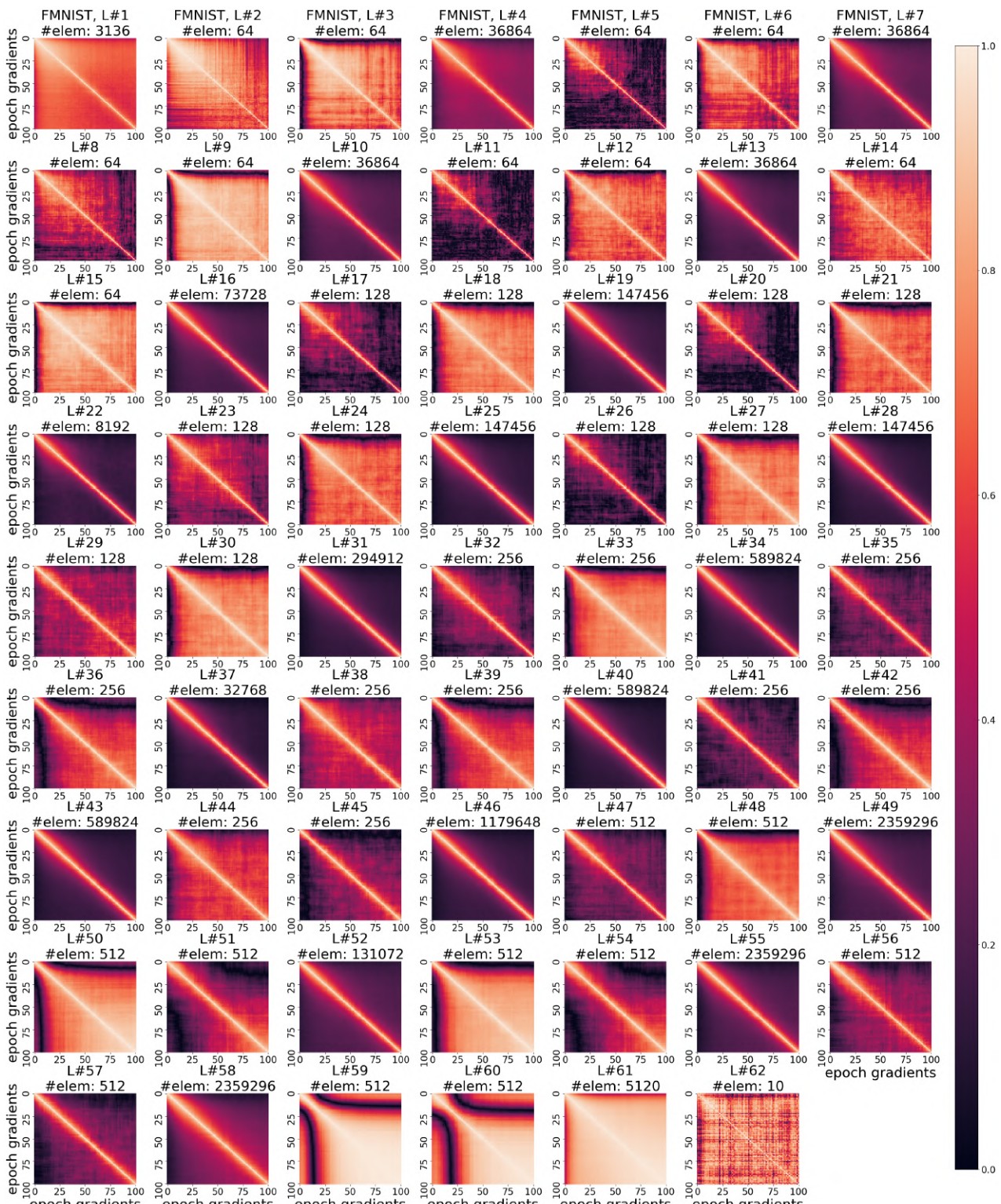

Figure 49: *PCA Components Overlap with Gradient.* Repeat of Fig. 3 on **ResNet18** trained on **FMNIST** dataset.

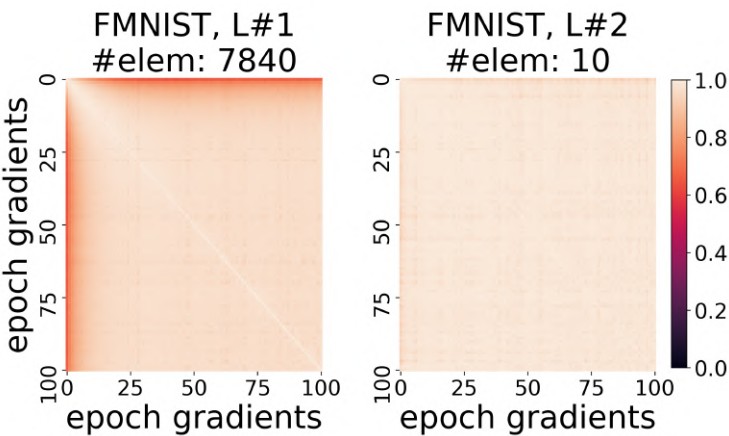

Figure 50: *PCA Components Overlap with Gradient.* Repeat of Fig. 3 on **FCN** trained on **FMNIST** dataset.

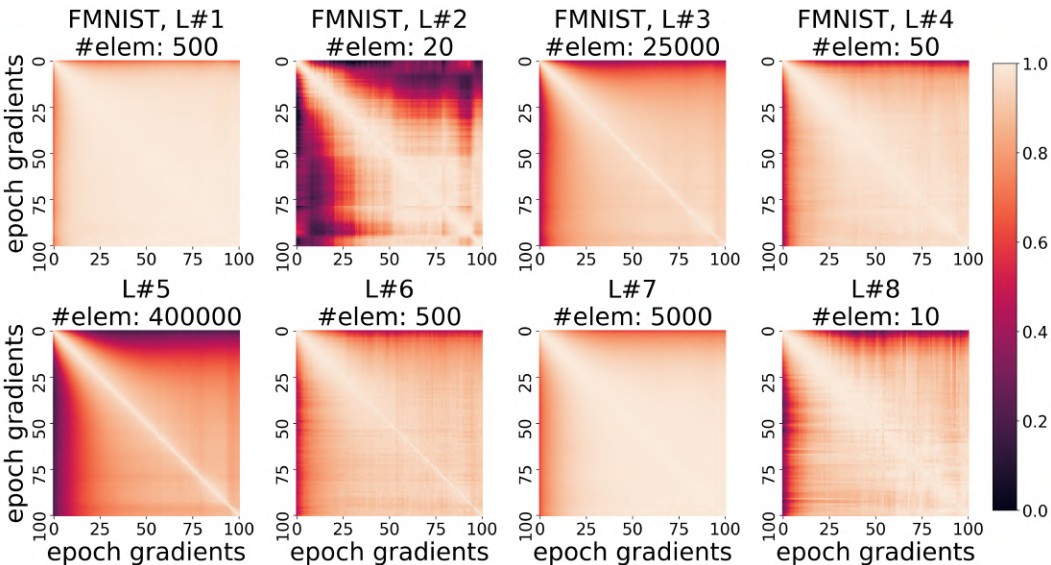

Figure 51: *PCA Components Overlap with Gradient.* Fig. 3 on **CNN** trained on **FMNIST** dataset.

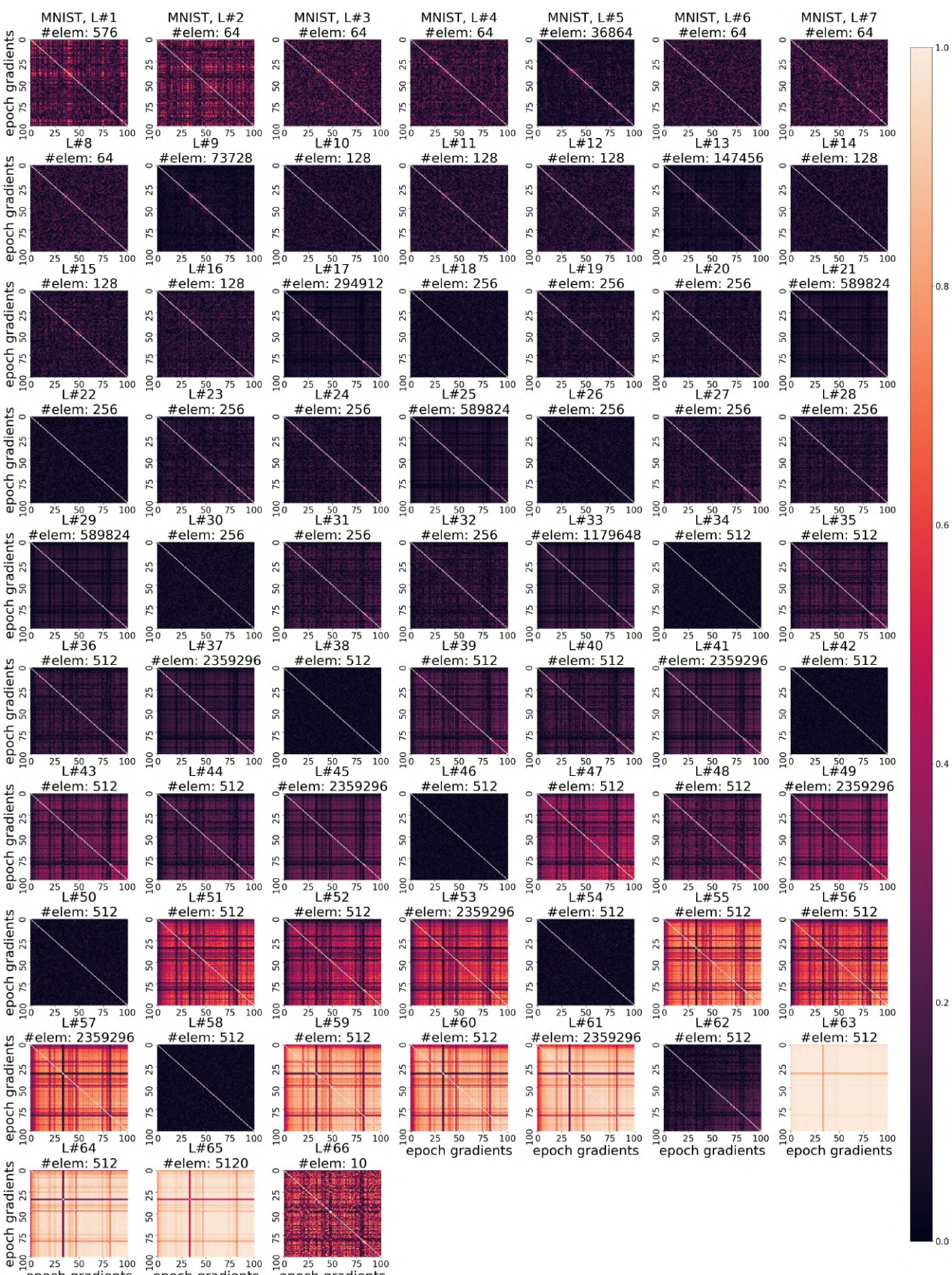

Figure 52: *PCA Components Overlap with Gradient*. Repeat of Fig. 3 on **VGG19** trained on **MNIST** dataset.

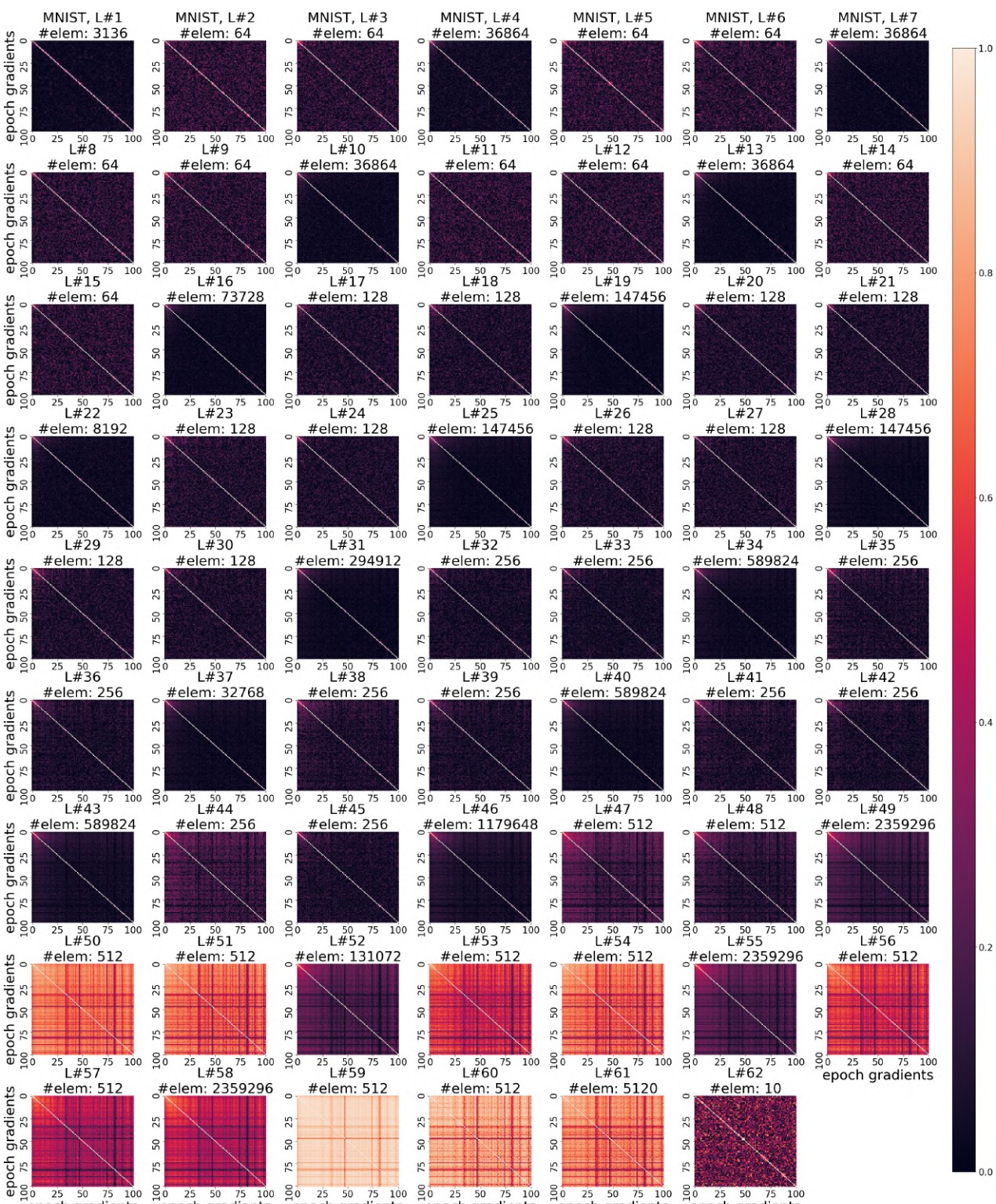

Figure 53: *PCA Components Overlap with Gradient*. Repeat of Fig. 3 on **ResNet18** trained on **MNIST** dataset.

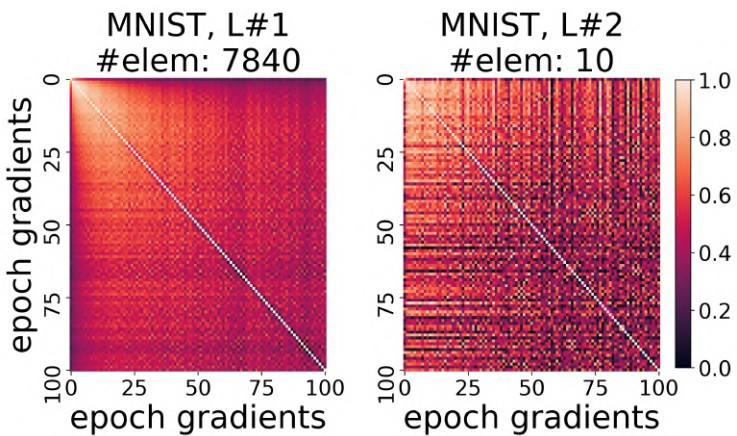

Figure 54: *PCA Components Overlap with Gradient*. Repeat of Fig. 3 on **FCN** trained on **MNIST** dataset.

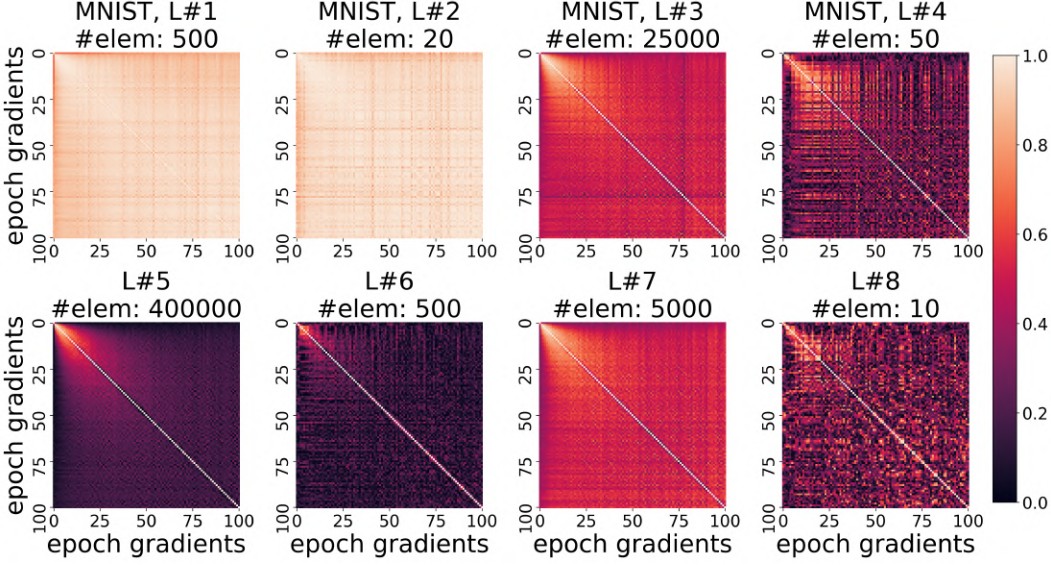

Figure 55: *PCA Components Overlap with Gradient*. Fig. 3 on **CNN** trained on **MNIST** dataset.

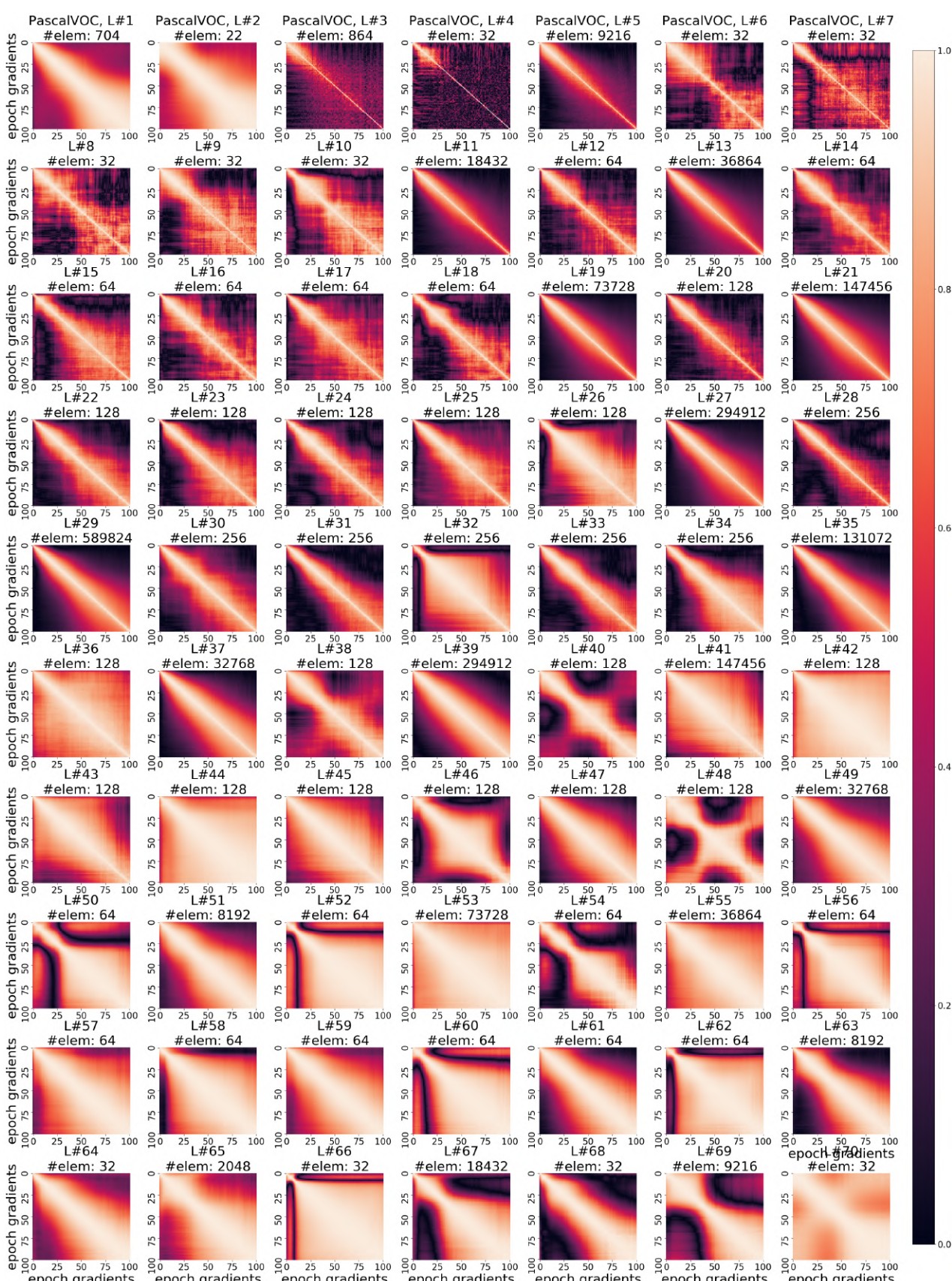

Figure 56: *PCA Components Overlap with Gradient*. Repeat of Fig. 3 on **U-Net** trained on **PascalVOC** dataset.

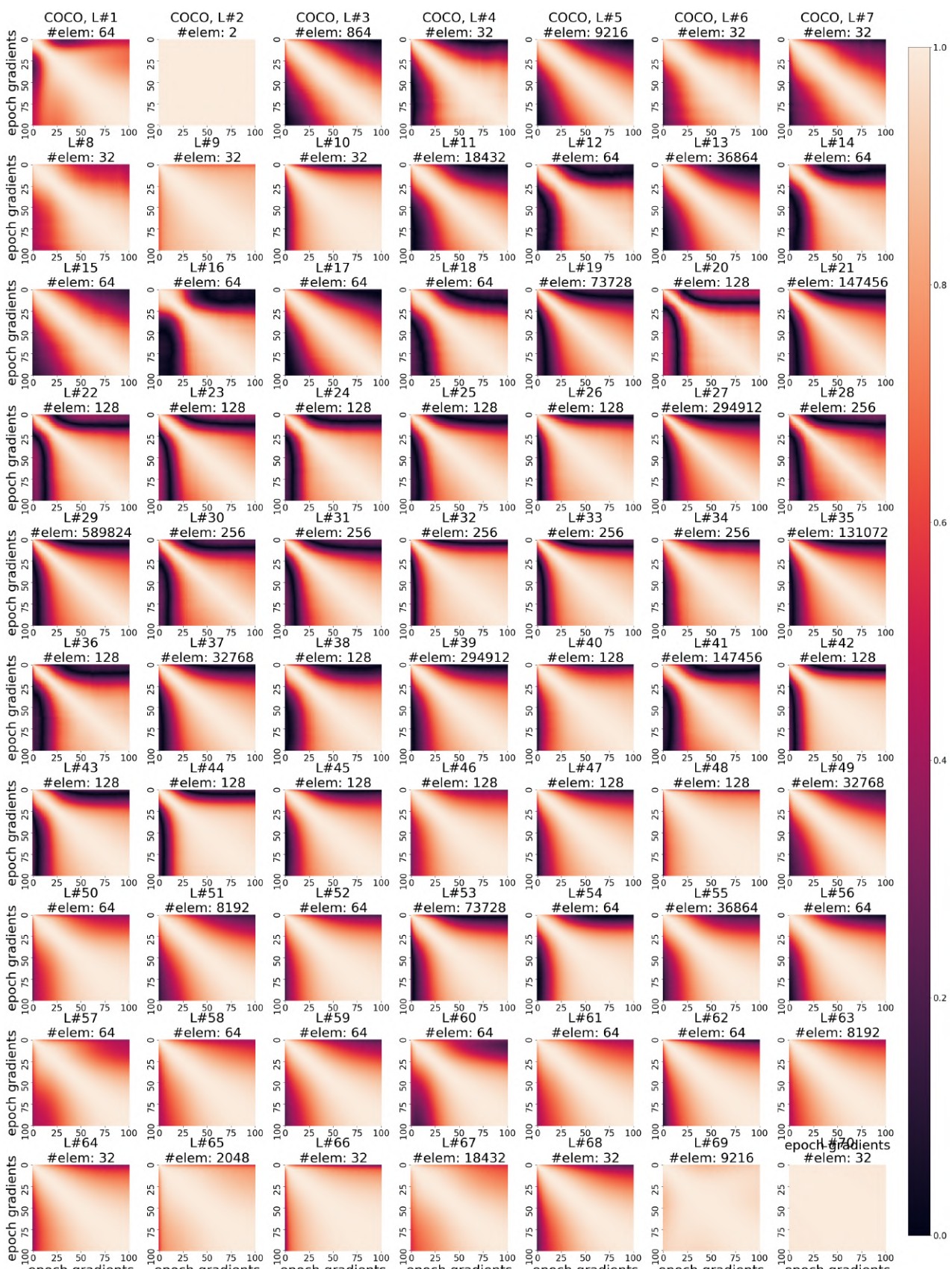

Figure 57: *PCA Components Overlap with Gradient.* Repeat of Fig. 3 on **U-Net** trained on **COCO** dataset.

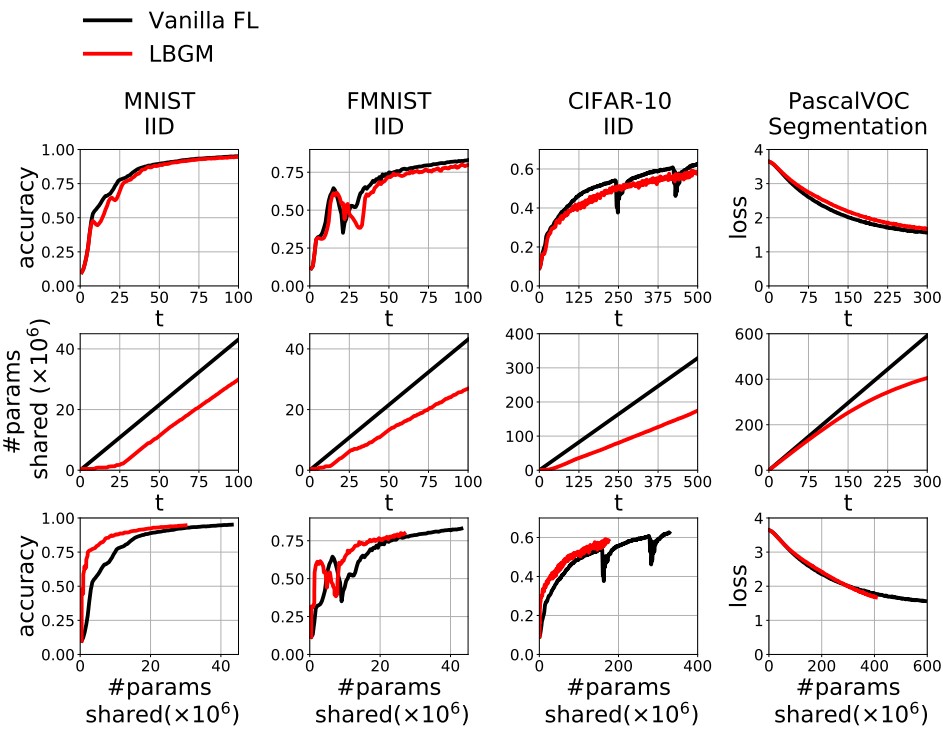

Figure 58: LBGM *as a Standalone Algorithm*. Experimental results in Fig. 5 repeated on datasets: **CIFAR-10**, **FMNIST**, and **MNIST** (iid data distribution) using classifier: **CNN**, and dataset: **PascalVOC** using **U-Net** architecture.

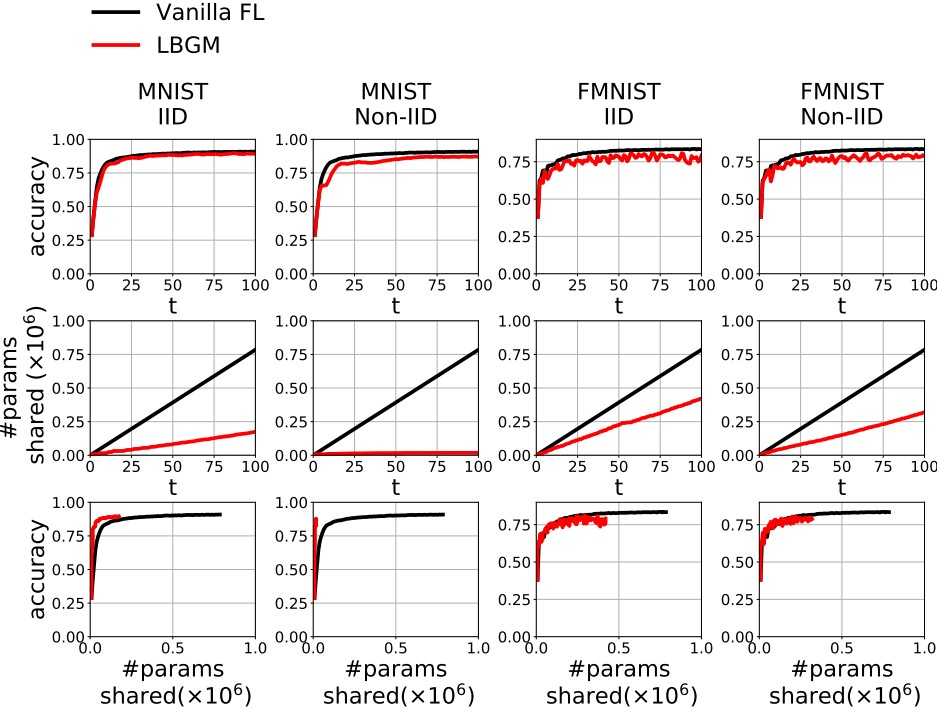

Figure 59: LBGM *as a Standalone Algorithm*. Experimental results in Fig. 5 repeated for datasets: **FMNIST** and **MNIST** (both iid and non-iid data distribution) using classifier: **FCN**.

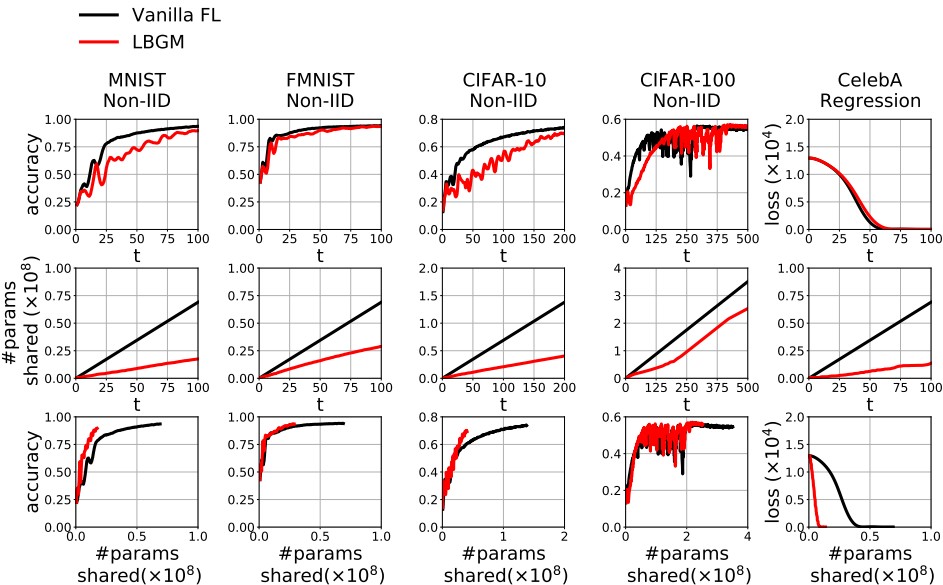

Figure 60: LBGM *as a Standalone Algorithm*. Experimental results in Fig. 5 repeated for datasets: **CIFAR-10**, **CIFAR-100**, **FMNIST**, **MNIST** (non-iid data distribution), and **CelebA** (face landmark regression task) using classifier: **Resnet18**.

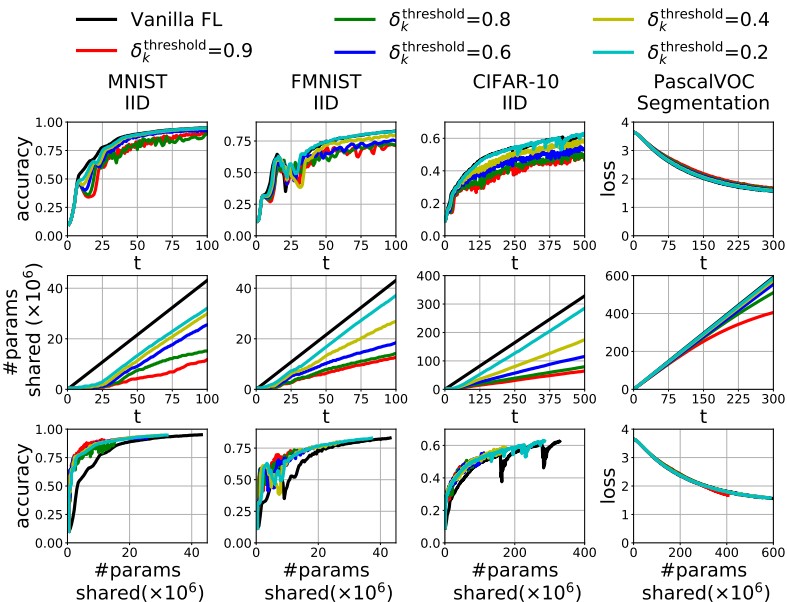

Figure 61: *Effect of $\delta_k^{\text{threshold}}$ on* LBGM. Experimental results in Fig. 6 repeated for datasets: **CIFAR-10**, **FMNIST**, and **MNIST** (iid data distribution) using classifier: **CNN**, and dataset: **PascalVOC** using **U-Net** architecture.

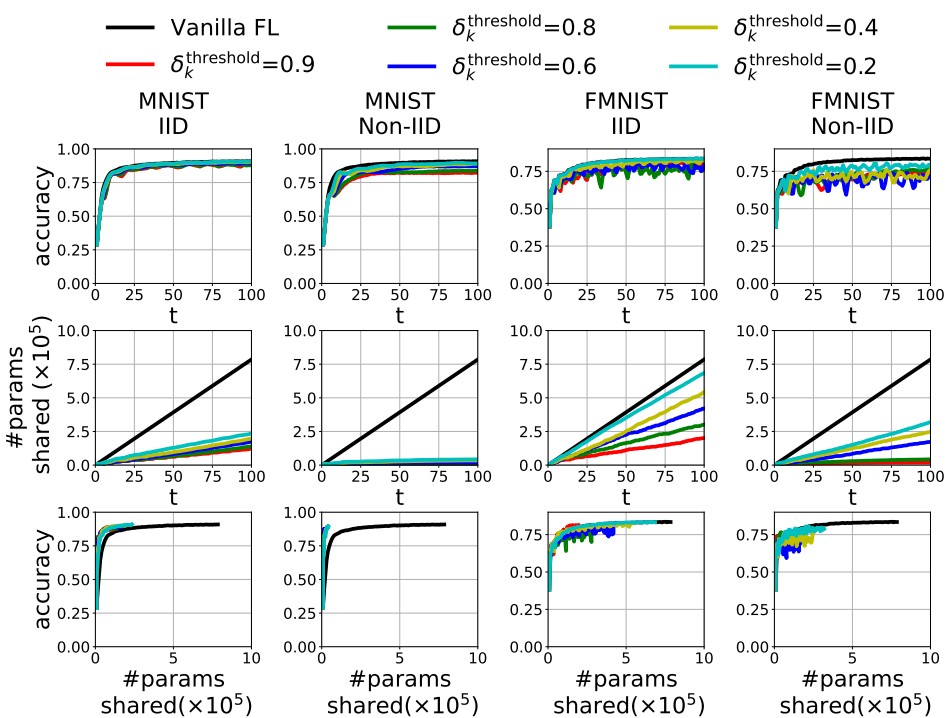

Figure 62: *Effect of $\delta_k^{\text{threshold}}$ on* LBGM. Experimental results in Fig. 6 repeated for datasets: **FMNIST** and **MNIST** (both iid and non-iid data distribution) using classifier: **FCN**.

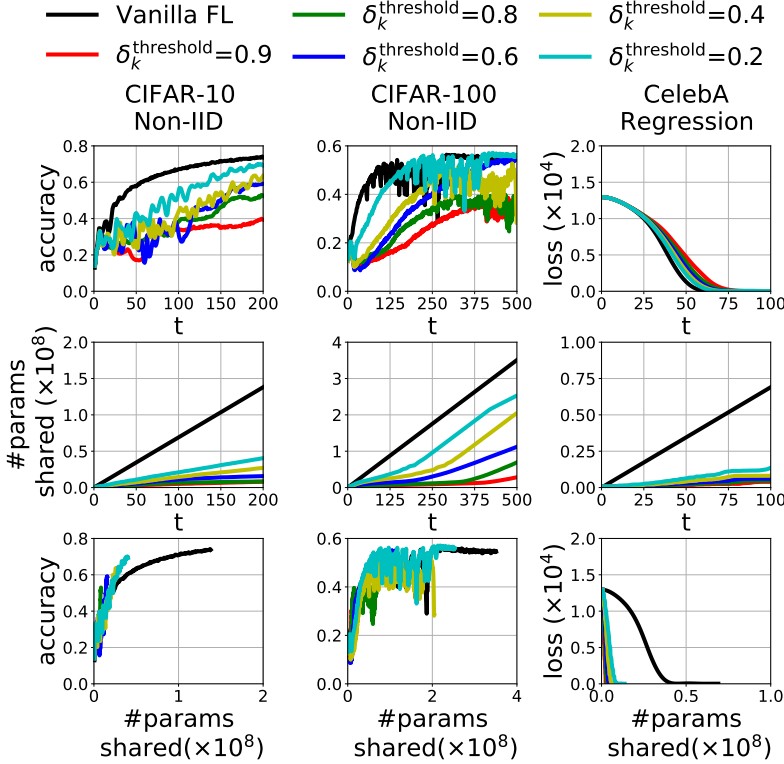

Figure 63: *Effect of $\delta_k^{\text{threshold}}$ on* LBGM. Experimental results in Fig. 6 repeated for datasets: **CIFAR-10**, **CIFAR-100** (non-iid data distribution), and **CelebA** (face landmark regression task) using classifier: **Resnet18**.

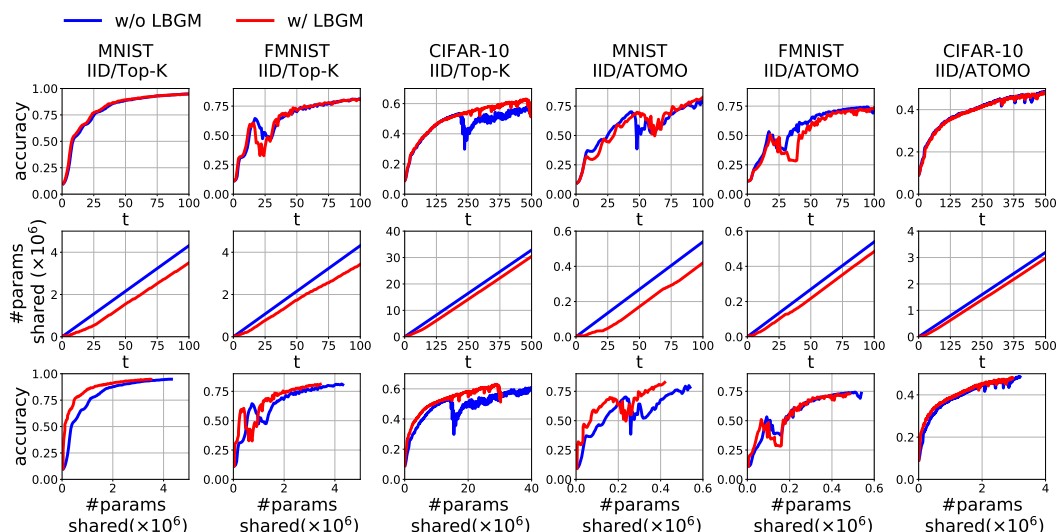

Figure 64: LBGM *as a Plug-and-Play Algorithm.*. Experimental results in Fig. 7 repeated for dataset: **CIFAR-10**, **FMNIST**, and **MNIST** (iid data distribution) using classifier: **CNN**.

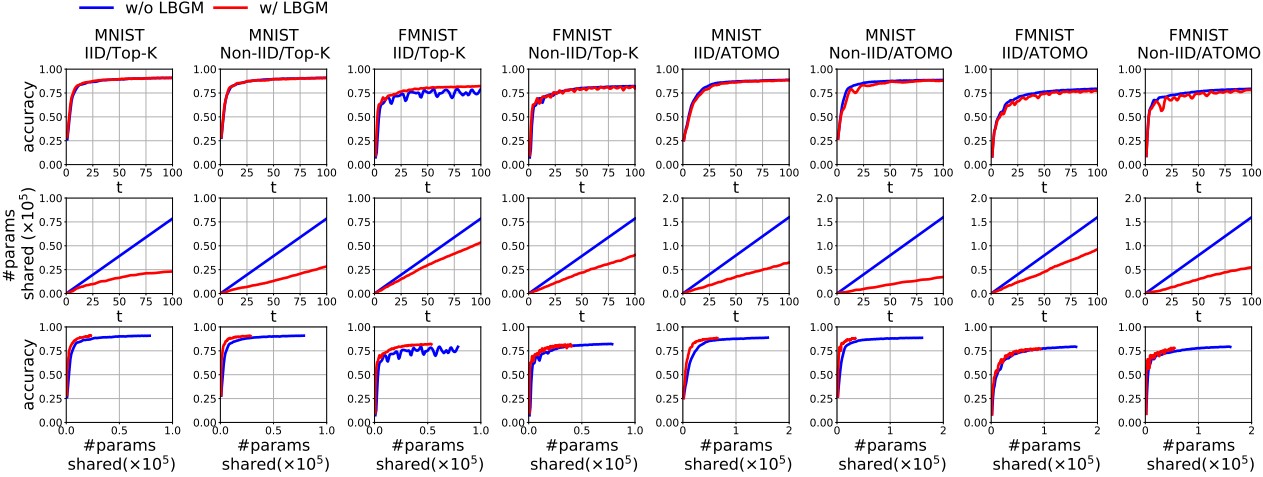

Figure 65: LBGM *as a Plug-and-Play Algorithm.*. Experimental results in Fig. 7 repeated for datasets: **FMNIST** and **MNIST** (both iid and non-iid data distribution) using classifier: **FCN**.

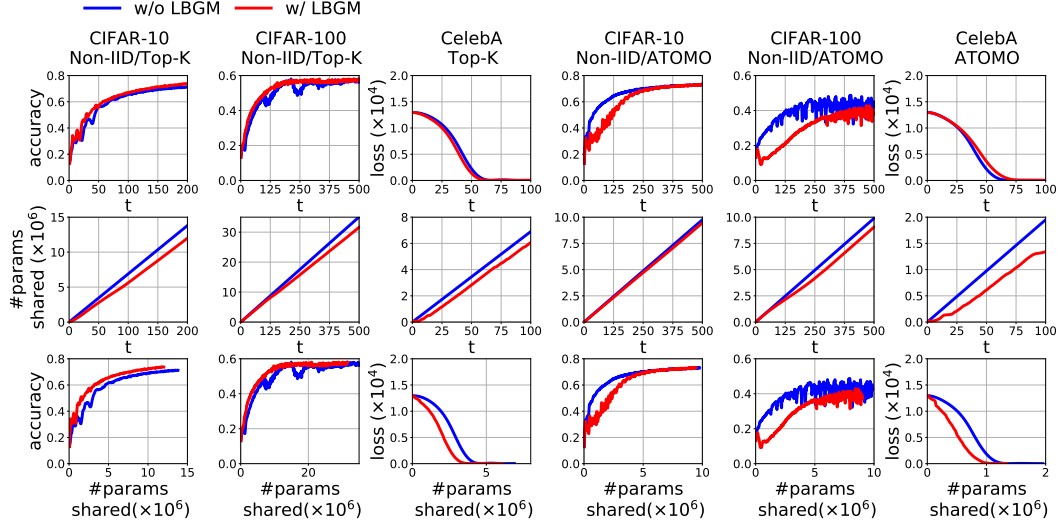

Figure 66: *Effect of* $\delta_k^{\text{threshold}}$ *on* LBGM. Experimental results in Fig. 7 repeated for datasets: **CIFAR-10**, **CIFAR-100** (non-iid data distribution), and **CelebA** (face landmark regression task) using classifier: **Resnet18**.

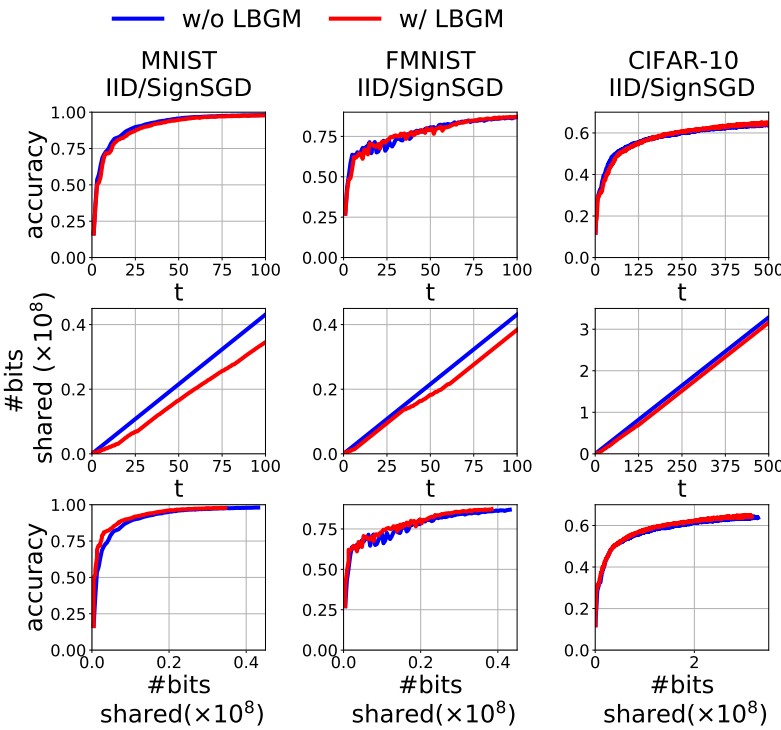

Figure 67: Experimental results in Fig. 8 repeated for dataset: **CIFAR-10**, **FMNIST**, **FMNIST** (iid data distribution) using classifier: **CNN**.

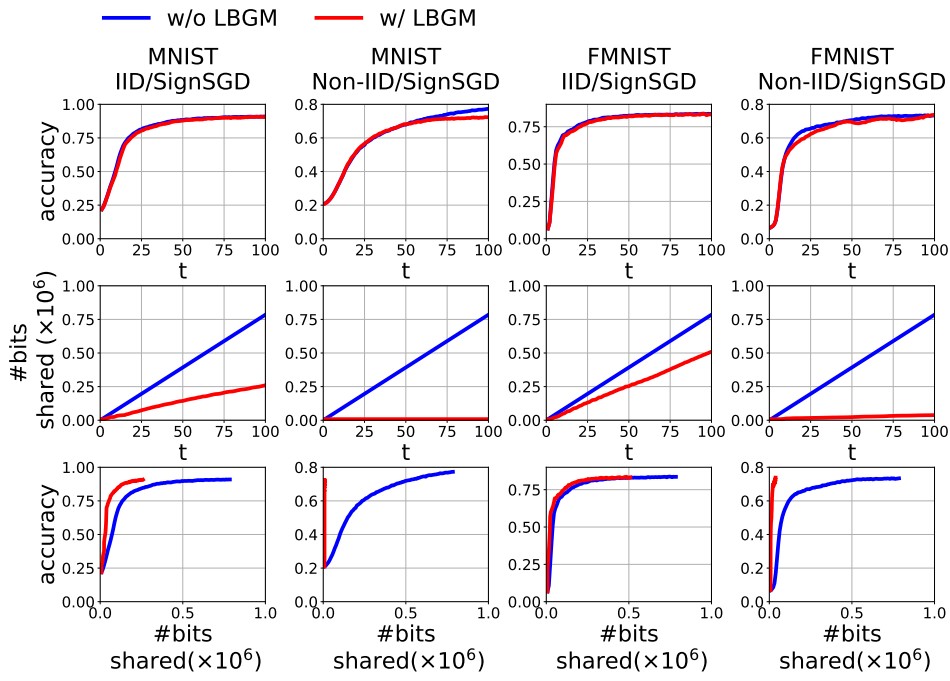

Figure 68: Experimental results in Fig. 8 repeated for dataset: **FMNIST** and **MNIST** (both iid and non-iid data distribution) using classifier: **FCN**.

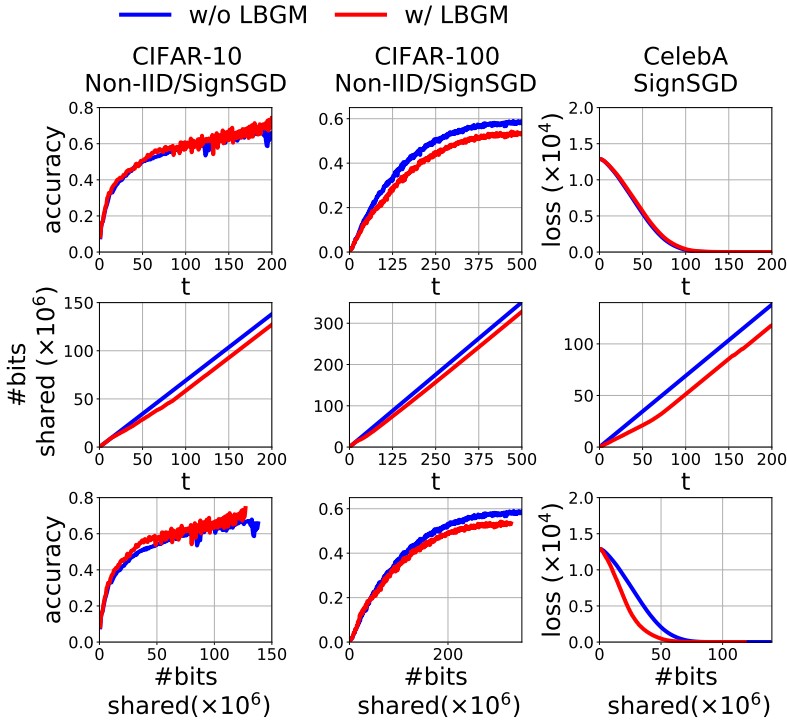

Figure 69: *Effect of $\delta_k^{\text{threshold}}$ on* LBGM. Experimental results in Fig. 8 repeated for datasets: **CIFAR-10**, **CIFAR-100** (non-iid data distribution), and **CelebA** (face landmark regression task) using classifier: **Resnet18**.

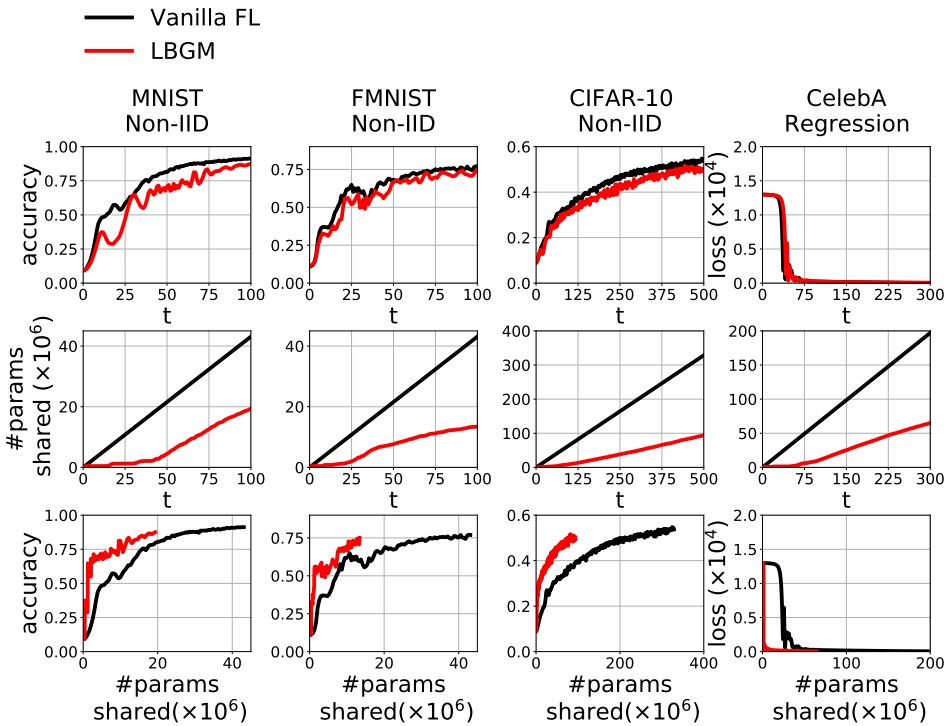

Figure 70: Experimental results in Fig. 5 repeated for dataset: **CIFAR-10**, **FMNIST**, **FMNIST** (non-iid data distribution) and **CelebA** (face landmark regression taks) using classifier: **CNN** under 50% **client sampling** for both Vanilla FL and LBGM.

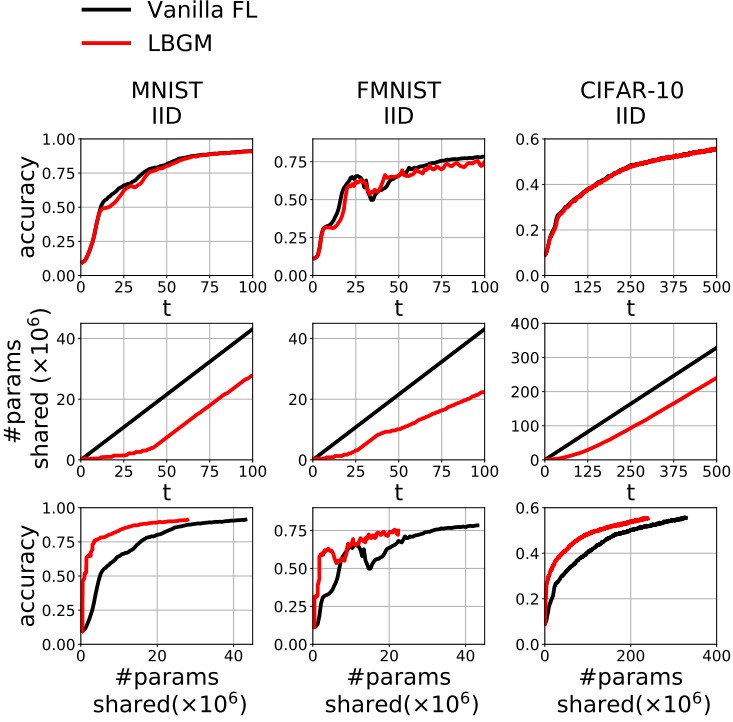

Figure 71: Experimental results in Fig. 5 repeated for dataset: **CIFAR-10**, **FMNIST**, **FMNIST** (iid data distribution) using classifier: **CNN** under 50% **client sampling** for both Vanilla FL and LBGM.

