# OpenReview forum: "Recycling Model Updates in Federated Learning: Are Gradient Subspaces Low-Rank?"
_ICLR.cc/2022/Conference — ICLR 2022 Poster_

### Official Review · Reviewer_FQxe · 2021-10-28

**Correctness:** 3
**Technical Novelty And Significance:** 2
**Empirical Novelty And Significance:** 3
**Recommendation:** 8
**Confidence:** 3

**Main Review:**

From my perspective, the main novelty is two folds.
One is the investigation of the gradient subspace, while the other is the proposed method which reuses similar past gradients to save communication.
The proof technique for the theorem is standard. However, it seems to have some mistakes, which though can be fixed with additional efforts and some quantities redefined.
The paper provides abundant experiments which I appreciate a lot.
Experiments illustrate communication efficiency and the plug-and-play property of their method.
Most of the writing is clear but some parts are hard to understand.

I have the following concerns:
1. The readability for Figure 1-3 should be improved. Without the help of Algorithm 2, I can’t figure out the meaning of each coordinate axis.
2. In the first paragraph of Section 3, it is better to add $\theta^{(t, 0)}_k$ is initialized as $\theta_k$ explicitly.
3. The description of LBGM Algorithm is hard to understand at the first glance. Some undefined terms are used, like look-back coefficients. I have to look up their meanings and exact mathematical forms in Lemma 1 and Algorithm 1, which increases the reading burden. I suggest the author could give a specific definition of frequently used terminology at the beginning. Besides, to better deliver the main idea of Algorithm 1, certain properties could be provided. For example, each device uploads just one LBG to the server, and the server tracks and maintains all updated LBGs.
4. Lemma 1 seems to have a mistake. Since both $\rho_k^{(t), \ell}$ and $\cos(\alpha_k^{(t), \ell})$ are scalers, the equality (L1) implies that the vectors $g_k^{\ell}$ and $g_k^{(t)}$ have the same direction (and thus are parallel), which is contradictory to (a) in Figure 4, where $g_k^{\ell}$ and $g_k^{(t)}$ has an angle.
5. The equality of the projections of a vector into two different vectors, though being wrong, is frequently used in the proof of Theorem 1. As a result, (14) and the first line of $Z_5$ are incorrect. However, these mistakes can be avoided. Letting $\tilde{d}_k^{(t)} = d_k^{(t)} \cos(\alpha_k^{(t), \ell})$ and abandon $\tilde{d}_k^{(t)} =  \rho_k^{(t), \ell}d_k^{\ell}$ solve the first problem. To bound $Z_5$, we can directly use the fact: $(1-\cos \theta)^2 \le |\sin \theta|$ for all $- \pi/2 \le \theta \le \pi/2$. In the way, though Theorem 1 can be rescued to be correct, the current proof is definitely not ready for publication.
6. It is unclear the frequency of LBC being transmitted. Note that communication is only saved when we communicate LBC rather than the whole vector. In the paper, related results include Figure 5 and 6 which investigates the variation of the total number of parameters shared. Besides, to compute the LBPs, we should first compute $\nabla F(\theta_k^{(t)})$ which incurs additional communication. I don’t know whether Figure 5 and 6 have taken this factor into account or not.

Some discussion:
1. I think there are more choices to set LBGs. In the paper, the LBG are just one of the history gradient. Why not define LBG as a moving average? This means we update the new LBG as the average of all LBGs. When the algorithm starts to converges, the average could be more stable.
2. The paper spends a lot of time studying the low-rank property of subsequent gradients. Why not let each device maintain more than one LBGs? In this way, each device has an LBG set, which is updated when the space it spans can’t explain the variance of the new coming LBG. Maybe we need PCA to determine whether the LBG set meets the requirement. With more LBGs, I guess the empirical performance would be improved further. After all, current experiments show with LBGM the test accuracy might slightly decay.

**Summary Of The Paper:**

The paper studies the gradient subspace and finds it low-rank propery.
This observation motivates them to propose a new algorithm that reuses similar past gradients to save communication.
They provide a theoretical analysis (with some mistakes) and conduct experiments to validate their method.

**Summary Of The Review:**

I appreciate the empirical investigation of low-rank gradient space and the idea behind the proposed method.
However, the current version of this paper is not proper for publication due to the mentioned concerns.

---

> ### Author Response · Authors · 2021-11-18
> **Response to Reviewer FQxe (1 of 1)**
>
> Thank you for taking the time to provide your feedback on our paper. Below are our point-by-point responses.
>
> > **Comment:   The readability for Figure 1-3 ... coordinate axis."**
>
> **Response:** To address this, we have taken another pass through Figs. 1-3, adding additional clarifications of the terminology and key takeaways in the captions.
>
> --------
>
> >**Comment:  In the first paragraph ... initialized as $\theta_k$ explicitly."**
>
> **Response:** We agree that this is an important point to clarify. We have added it to the algorithm description in Section 3.
>
> --------
>
> > **Comment:   The description of LBGM ... maintains all updated LBGs."**
>
> **Response:** We have added a definitions/terminology section at the top of Algorithm 1 to make it more self-contained.
>
> --------
>
> > **Comment:   Lemma 1 seems to have a mistake. ... not ready for publication."**
>
> **Response:** We apologize for this mistake. In particular, the second equality in (L1) was missing  a "unit vector direction", and the equivalency (i.e., $\equiv$) was meant to be imposed on the norm of the gradients. *In the revised manuscript, this has been corrected throughout our proofs and analysis (Fig. 4(a) has also been updated)*. Since the analysis is dependent on the L2 norm, the flow of the derivations remains intact, and the results indeed still hold as anticipated by the reviewer.
>
> ---------
>
> > **Comment:   It is unclear the frequency ... factor into account or not"**
>
> **Response:** The calculation of LBPs is conducted completely locally at each device (see line 6 of Alg. 1), since each device also has the memory of its look-back gradient. Thus, we do not require computing the global gradient (i.e., $\nabla F(\theta_k^{(t)})$) to estimate the local LBPs. We are happy to add additional clarification if needed.
>
> ---------
>
> > **Comment:  I think there are more choices ... average could be more stable."**
>
> **Response:** This is an interesting idea worth further investigation in future work. The length of the moving average window may play a key role in the performance of LBG in this scenario, which requires careful consideration. Note that since some of our experiments use momentum in SGD, this moving average effect with window size $1$ is already considered in the LBG calculated by a device. As our paper provides a foundational technique for recycling gradients in federated learning, we indeed expect multiple enhancements that can be applied on our LBGM algorithm in future works.
>
> --------
>
> > **Comment:  The paper spends a lot of ... test accuracy might slightly decay."**
>
> **Response:**
>
> - Storing more than one LBG might indeed be helpful since it reduces the error of gradient reconstruction at the main server. However it increases the storage burden on the server and the device. If test accuracy is a concern we can also tune the $\delta^{\mathsf{threshold}}$ accordingly.
>
> - Taking the example of Figure 5, CIFAR-10 uses $\delta^{\mathsf{threshold}} = 0.2$ and realizes a $14\\%$ drop in performance for a $79\\%$ gain in communication savings. If we reduce it to $\delta^{\mathsf{threshold}} = 0.05$, we obtain a performance drop of only $4\\%$ while still retaining a communication savings of $55\\%$. Similarly, for $\delta^{\mathsf{threshold}} = 0.01$, we get $22\\%$ communication savings for a negligible drop in accuracy (by only $0.1\\%$). *We have included this discussion in Section 4 of the main paper under "LBGM as a Standalone Algorithm."*

---

### Official Review · Reviewer_LE1r · 2021-11-02

**Correctness:** 3
**Technical Novelty And Significance:** 3
**Empirical Novelty And Significance:** 2
**Recommendation:** 8
**Confidence:** 4

**Main Review:**

## Important existing work

I'd prelude everything by first mentioning **[this paper](https://arxiv.org/pdf/2103.11154.pdf)** from earlier this year, which is closely related to the paper but hasn't been cited. It approximates the subspace $S$ using only the (first-order) information from the optimization trajectory and then uses it to build a quasi-newton method. In light of this, it is not true that the paper is the first work that studies,
>the effect of overparameterization on optimization of NN via the rank-characteristics of the first order optimization (gradient) landscape.

Furthermore, an **[older paper](https://arxiv.org/pdf/1812.04754.pdf)** (which is cited) has already made the case that the stochastic gradients lie in a tiny sub-space, which is **(H1)** in this paper. I don't understand why this is underplayed while discussing this paper in the related works section, where this paper is clubbed with other papers that study the hessian?

Having said that, I am a bit apprehensive about both **(H1)** and **(H2)** (more below). But if we believe these hypotheses, it is indeed novel to use them in the context of federated learning. I'd encourage the authors to revise their writing and their discussion of novelty in the light of these papers.

## Results on other data-sets

It is commendable that the authors do experiments with many different data sets and architectures and include them in the appendix. However, I am not convinced that the two hypotheses **(H1)** and **(H2)** are well corroborated in all the experiments. For instance, it is not clear if **(H1)** holds true in figures 11 and 13. Why were these experiments not run long enough to actually let the PCA components stabilize and the curve flatten out? Similarly in figures 18, 19, 23, 26, 27, 31, i.e., experiments with bigger models it is not clear if either hypothesis is true, and the figures look uninformative when compared to figure 2 which is presented in the main paper. Similarly, the consecutive gradient alignment curves in figures 52, 53 look uninformative.

I am not saying that this phenomenon can be refuted entirely, but it seems that it is at least not robust to the choice of the model and dataset. And this has not been discussed sufficiently in the paper.

Finally, why were figures 5-8 not presented for instance for U-Net for Pascal-VOC? Were the improvements worse? If they were it is important to include those for a fair presentation of the work. In fact, for this technique to be relevant to federated learning, it is important to know how it scales to bigger models and datasets. I am specifically mentioning this instance, based on the gradient alignment curves and model size.

### Learning rate scheduling
What is happening at the 25th iteration in figure 3 for CelebA? Is there a learning rate decay there? These kinds of shifts in principal components happen throughout all the curves in the appendix as well. Looking at the provided code it is not clear what learning rate schedules were used for respective figures, and it would be good to highlight this in the paper. More importantly, does the schedule impact the principal components? It seems to be the case looking at the curves. It would be good to investigate that.

## Convergence Rate

I am mostly happy with the convergence analysis and the following discussion. The analysis is not particularly novel, but it is good to see how $\Delta$ appears in the convergence guarantee.

***

## Minor comments

>Can we observe the effect of overparameterization in gradient descent-based optimization of NNs

1) This comes off as an ill-posed question, given the plethora of empirical and theoretical work that has come up in this area.

>We further reveal that LBGM can be extended to distributed training

2) The distinction that the authors make between FL and distributed training is slightly confusing (even later in the paper). Latter subsumes the former, and it seems that the authors want to demarcate between homogeneous v/s heterogeneous data-distribution settings or cross-silo v/s cross-device FL. This is especially confusing when the authors choose to demarcate between the baselines for **P2** and **P3**. I think the point there is that just communicating gradient signs can be detrimental for the heterogeneous setting?

3) I don't like the term *principal gradients* for principal components of $S$, as they are not really *gradients*. Moreover, I couldn't find anywhere a mathematical description of how these components are calculated, perhaps using SVD of the concatenation of all the gradients? The pseudocode in the appendix just uses a black-box function. It would be useful to include this in the appendix and specify how different levels of variance are specified.

>SGD is resilient to noisy updates and often benefits from them in terms of generalization error

4) This is a vague statement, first of all, what does it mean for an optimization algorithm to be resilient to noisy updates? Do you mean that the analysis of SGD can be done for any unbiased stochastic gradients, with bounded variance? That is true, but it is unclear (at least apriori) if the noise introduced by the LBGM procedure is like that. This is a slippery slope argument to make. Or do you mean that while training with SGD, additive noise doesn't hurt the performance? If you mean the latter, I am not convinced that there is conclusive evidence to show this, across a range of models, data sets, and noise levels. So this is an underqualified statement as well. Please re-write this sentence or avoid it.

5) Lemma 1 seems trivial to me. You are basically re-deriving what it means to project onto a vector. Is it really worth putting this into a lemma and writing a proof? or am I missing something here?

> S2: FCN on FMNIST and FMNIST

6) Typo

>The fact that rank deficiency of the gradient-space is not a consequence of model complexity or model performance suggests that the gradient-space of state-of-the-art large-scale ML models could be represented using a few principal gradient directions.

7) I was hoping that the authors will further investigate what causes this rank deficiency. That would have been a significant contribution by itself. **[This paper](https://arxiv.org/pdf/2103.11154.pdf)** conjectured that it had something to do with the number of classes in the data-set. Could the authors corroborate or refute that claim?

8) In many experiments (such as figure 2) the stochastic gradients were almost determined by a single PCA gradient. Did the authors investigate what caused this? How is the learning rate being decayed in these figures?

9) Did the authors experiment with layer-wise PCA gradients? Perhaps even a proof of concept experiment could be done by using actual principal components of $S$?

**Summary Of The Paper:**

This paper hypothesizes that,
**(H1)** the subspace $S$ spanned by the stochastic gradients while training through SGD is low rank, and
**(H2)** $S$ is well approximated by a subset of the actual stochastic gradients.

It provides empirical evidence to support these claims by calculating,
(i) the number of principal components that almost explain the entire variance of the stochastic gradients over time, and
(ii) the alignment between the principal components of $S$ and the stochastic gradients, as well as the consecutive stochastic gradients, across different models and data sets.

With these observations, the paper aims to reduce the complexity of the communicated bits between the devices in data-parallel distributed training. Specifically, it proceeds by first estimating the principal components of $S$ through consecutive stochastic gradients. These vectors are then stored at the server as well as each worker. At each communication round the device then only needs to share the projection coefficients for these principal vectors, until the set of the principal vectors needs to be updated and broadcasted to all the machines.

This technique is empirically shown to reduce the communicated bits while almost retaining the performance on certain data sets. A convergence rate is also provided for this algorithm under assumptions that highlight the trade-off between the frequency of updating the principal vectors and the convergence rate to a first-order stationary point.

**Summary Of The Review:**

I think the claims of novelty in the first half of the paper are slightly inflated. The application to federated learning is definitely novel though and should become the focus of the paper. This can be done by improving the discussion of the related work, providing fair and well-rounded evidence for the low-rank hypotheses, and extending the experiments for federated learning to the biggest data-set and model possible. I am open to increasing my score if my concerns are addressed.

---

> ### Author Response · Authors · 2021-11-18
> **Response to Reviewer LE1r (3 of 3)**
>
> --------
>
> > **Comment: What is happening at the 25th iteration ... good to investigate that."**
>
> **Response:**
>
> - We perform a decaying learning rate scheduling using cosine annealing [1] for most experiments. *We have now highlighted this in Sec. 2 of the updated paper.* The details can also be found in our code (https://github.com/PairML/LBGM). The directory `src/sh/` contains all the shell scripts that can be run directly to reproduce our results.
>
> - As pointed out by the reviewer, there might be some effect of scheduling on the overall PCA progression. We indeed observe that as we decrease the learning rate, the model starts exploring newer directions, and thus we see an uptick in the PCA component progression. This may or may not lead to an increase in performance, however. We thank the reviewer for this insightful remark and *we have added this remark in Sec. 2 of the updated paper*.
>
> [1] https://pytorch.org/docs/stable/generated/torch.optim.lr_scheduler.CosineAnnealingLR.html
>
> --------
>
> > **Comment: Can we observe the effect ... that has come up in this area."**
>
> **Response:** We agree with the reviewer's remarks and have changed this to the following more specific question: "Can we observe the effect of overparameterization in gradient-descent based optimization of NNs directly through the PCA decomposition of gradients generated during SGD-based model training?"
>
> --------
>
> > **Comment: The distinction that the authors make ... for the heterogeneous setting?"**
>
> **Response:** As mentioned by the reviewer, the main distinction is in the homogeneous vs. heterogeneous settings. We have added this clarification in Section 4.
>
> -----------
>
> > **Comment:  I don't like the term principal gradients ...  different levels of variance are specified."**
>
> **Response:**
>
> - We understand the concern regarding the terminology principal gradient. We adopted it as a concise way of addressing the concept, as opposed to the longer term "principal components of the generated gradients." *We have changed it to principal gradient directions (PGD) to alleviate the concerns.*
>
> - The components are indeed calculated using SVD decomposition of the concatenation of all the gradients. In the appendix we present the black box function and further point to specific functions in our repository (https://github.com/PairML/LBGM) that defines them. *We have further clarified this process in the algorithm section Appendix D.1.*
>
> --------
>
> > **Comment:  This is a vague statement, first of all ...  re-write this sentence or avoid it."**
>
> **Response:**
>
> - First we would like to clarify that we are not meaning to claim that LBGM noise is similar to SGD noise. We introduced this sentence in the earlier part of the paper to motivate the fact that we do not need to find the exact principal gradient directions to represent the gradients, and that a noisy approximation might work as it is shown to throughout our experiments.
>
> - That being said, we agree that in the current form the sentence might mislead the reader to think that we are comparing SGD noise to LBGM noise. So *we have taken the sentence out as suggested.*
>
> --------
>
> > **Comment: Lemma 1 seems trivial ...  missing something here?"**
>
> **Response:** We agree with the point. We have changed the name Lemma 1 to Definition 1 and removed the proof for brevity.
>
> -----------
>
> > **Comment: I was hoping that the authors will ... corroborate or refute that claim?"**
>
> **Response:** Our experimental findings do not align with the conjecture of the mentioned paper. In our observations we notice that the rank of gradient subspace can vary depending on the choice of the model and dataset (and not just the number of classes). For example, in Fig. 1, CIFAR-10 with CNN has extremely low rank structure unlike CIFAR-10 with Resnet-18. *We have added this discussion in the related work section of the updated paper.*
>
> ------
>
> > **Comment:  In many experiments (such as ... rate being decayed in these figures?"**
>
> **Response:**
>
> - Investigation of this effect is an open question that deserves to be studied further. Fig.~1, 2 \& 3 show different perspectives of the same experiments performed with CIFAR-10 and CelebA.
>
> - The learning rate scheduling is done using cosine annealing implemented in PyTorch; the exact formulation is available in [4]. *We have added this reference in Sec. 2 (as a footnote) of the updated paper.*
>
> [4] https://pytorch.org/docs/stable/generated/torch.optim.lr_scheduler.CosineAnnealingLR.html
>
> ------
>
> > **Comment:   Did the authors experiment with ...actual principal components of S?"**
>
> **Response:** We did not consider the layer-wise PCA gradients. Indeed it is an interesting idea, which deserves to be studied. In general, what discourages the implementation of PCA in a federated learning setting is that the end devices (i.e., workers) are usually low power wireless devices (e.g., cellular phone, tables) for which large matrix decompositions may be prohibitive.

---

> > ### Comment · Reviewer_LE1r · 2021-11-30
> > **Response to authors**
> >
> > I thank the authors for their detailed comments and apologize for my delayed response.
> >
> > I appreciate the changes made to the writing, new experiments, and discussions. I believe the paper has indeed improved over the original submission. Most of my concerns have been addressed and I have reflected this in my new score.

---

> > > ### Author Response · Authors · 2021-11-30
> > > **Thank you**
> > >
> > > Thank you very much for your consideration.

---

> ### Author Response · Authors · 2021-11-18
> **Response to Reviewer LE1r (2 of 3)**
>
> > **Comment:Having said that, I am a bit ... in the light of these papers"**
>
> **Response:** We have refined our summary of contributions in Section 1 and related work in Section 5 to reflect these papers. We focus on highlighting that we are the first to exploit the low-rank characteristics of gradients in improving the training accuracy vs. communication efficiency tradeoff in federated learning.
>
> -------
>
> > **Comment:It is commendable that the authors ... components stabilize and the curve flatten out?"**
>
> **Response:** In our original hypothesis testing (in Fig. 11, 13) we did a PCA progression analysis until a fixed number of epochs or reaching an accuracy saturation. One of the reasons we limit the number of epochs is that each SVD decomposition is incrementally time consuming and often causes out of memory errors for large number of epochs; for example, cumulative truncated SVD decomposition for VGG-19 with 300 epochs takes over 36 hours to process, and the operations are often killed because of memory overflow. However, we agree that it might be insightful to see when the PCA progression stabilizes. As a result, *we have updated Figs. 11 and 13 to run for a longer number of iterations.*
>
> ---------
>
> > **Comment:Similarly in figures 18, 19, 23, 26 ...is presented in the main paper."**
>
> **Response:** *We have added more supporting discussion for the mentioned figures (18, 19, 23, 26, 27, 31)* since we agree that the main takeaways are not clear at a first glance. Using Fig. 18 as an example, we start off by pointing out that the number of principal components is substantially lower than the total number of epochs. We can also see a better picture of gradient overlap with other gradients (inter-gradient overlap) in Fig. 40. This is consistent with the corresponding PCA progression in Fig. 1. Note that a lesser number of principal gradient components (e.g., CNN in Fig. 1) implies a higher overlap among the generated gradients (e.g., CNN in Fig. 3), while a larger number of principal gradient components (e.g., ResNet in Fig. 1) implies a lower overlap among the generated gradients (e.g., ResNet in Fig. 40).
>
> ---------
>
> > **Comment: Similarly, the consecutive ...52, 53 look uninformative."**
>
> **Response:** Fig. 51 and 52 can be considered as examples where large gradient overlaps are not seen as often as in our other experiments. Nevertheless, LBGM still obtains significant communication savings when applied on these datasets and NN models. To observe this, refer to Fig. 60 which has been updated to include LBGM on Resnet18 and MNIST; we see substantial communication benefit while achieving comparable accuracy for $\delta^{\mathsf{threshold}}=0.2$. *We have now highlighted this in Sec. 2 of the updated paper.*
>
> -------
> > **Comment: Finally, why were figures 5-8 not presented  ...  alignment curves and model size."**
>
> **Response:**
>
> - In Section 2, we start by showing that our hypothesis holds for a majority of chosen datasets and models. We then proceed to conduct the federated learning experiments on a smaller subset of datasets that have been commonly considered in FL works. However, to address the concern, *we have added the experiments with PascalVOC and U-Net in Appendix F.1 (Fig. 58 last column) and Appendix F.2 (Fig. 61 last column)*.
>
> - We also remark that we have tested our method on large data sets (e.g., CIFAR-100, CelebA) and large models (e.g., ResNet18). Each of these cases has shown benefit from LBGM and revealed the scalability of our method.

---

> ### Author Response · Authors · 2021-11-18
> **Response to Reviewer LE1r (1 of 3)**
>
> Thank you for taking the time to read through our paper and provide your comments. Below are our point-by-point responses.
>
>
> --------
>
> > **Summary Comment: I think the claims of novelty in the first half of the paper are slightly inflated. The application to federated learning is definitely novel though and should become the focus of the paper. This can be done by improving the discussion of the related work, providing fair and well-rounded evidence for the low-rank hypotheses, and extending the experiments for federated learning to the biggest data-set and model possible. I am open to increasing my score if my concerns are addressed.**
>
> **Response:**
>
> - We have modified the contribution section of the paper and adjusted the text to reflect the reviewer's comments.
> - Also, we have revised the related work section to reflect the reviewer's comments (detailed response below).
> - As per the reviewer's suggestion, we have added an experiment with U-Net architecture and the PascalVOC dataset (see Fig. 58). (detailed response below).
>
> --------
>
> > **Comment:I'd prelude everything by first ... optimization (gradient) landscape."**
>
> **Response:** We agree that the paper [1] (henceforth called DLDR) is a contemporary of ours in directly analyzing the first order optimization landscape. However, there are several major differences:
>
> - Our methodology proposes the use of a single LBG to approximate the current gradient, while DLDR uses 40 orthonormal bases estimated using SVD decomposition.
> - We do not depend on a computationally-intensive technique such as SVD and instead conjecture that it is possible to approximate/replace the principal components with a small subset of gradients generated during the training process.
> - DLDR, which is proposed for centralized setting, cannot be directly applied to federated learning because storing 40 orthonormal bases (each the size of the model parameter) is unrealistic for both the server and the devices.
> - The implementation mechanism (i.e., use of LBC, error measurement using LBP, LBG update, etc.) of our method are fundamentally different than DLDR.
>
> *We have included this paper in our related work section and have toned down the statement "first to study the effect of overparameterization on optimization of NN via the rank-characteristics of the first order optimization (gradient) landscape."*
>
> [1] Li et al. "Low Dimensional Landscape Hypothesis is True: DNNs can be Trained in Tiny Subspaces" 2021.
>
> -------
>
> > **Comment:Furthermore, an older paper ... study the hessian"**
>
> **Response:** We consider [2] to be closer to the works studying Hessians because the work argues that SGD gradients lie in a tiny subspace in terms of their overlap with the Hessian as the training continues. In [2], there are two major steps which lead to the claim that SGD happens in a tiny subspace":
>
> - demonstrating gradient overlaps with the dominant subspace of the Hessian, and
> - demonstrating that the dominant subspace of the Hessian is invariant over training.
>
> However, in their experiments, the authors only demonstrate the overlap of gradient with the Hessian (not the dominant subspace), which calls into question the justification of their method (see pt. 1 by AnonReviewer3 in the official review: https://openreview.net/forum?id=ByeTHsAqtX). We on the other hand investigate the dimensionality of the subspace through comprehensive PCA analysis across several models and datasets. Also, while [2] focuses on an exploration of gradient descent, a key contribution of our paper is in leveraging the low-rank gradient space properties to optimize federated learning. *We have included these discussions in our related work section now.*
>
> [2] Gur-Ari et al. "Gradient descent happens in a tiny subspace." 2018.

---

### Official Review · Reviewer_X6Uf · 2021-11-03

**Correctness:** 3
**Technical Novelty And Significance:** 4
**Empirical Novelty And Significance:** 4
**Recommendation:** 8
**Confidence:** 4

**Main Review:**

In general, I think the proposed algorithm is quite novel and interesting. It explores a new research direction to reduce the communication in FL. The theoretical analysis is also a nice addition to the paper. It provides many useful insights about the proposed algorithm.

In terms of weakness, I mainly have two concerns:
1. In experiments, it seems that in most cases, using LBGM will degrade the performance a lot. For example, in figure 5, on CIFAR-10, the accuracy nearly drops 5-10%. Although the communication is significantly reduced, the performance degradation may prevent people using this algorithm.
2. The algorithm is more suitable for classical distributed learning rather than federated learning. I encourage the authors to change the title and related parts in the paper. The reason is that in FL, at each round, only a few (maybe around 1-10%) clients will participate into training. That means, one client can be inactive for multiple rounds. As a result, the principle component on this client will likely be stale when it participate training again. So the client needs to transmit full gradient again. In this case, the benefits of using LGBM may diminish..

**Summary Of The Paper:**

This paper proposed two hypothesis: (1) the space spanned by gradients generated during training is low-rank; and (2) the principle components can be approximated by the gradients generated during training. The authors first validated these two hypotheses on several datasets. Then, based on these, they proposed a new algorithm called LBGM to reduce the communication costs in federated learning. In the algorithm, the clients local gradients will be treated as the principle components (if there are K clients, then we have K principle components). Then, during training, each clients only needs send a scalar, which represents the coefficient of the corresponding components, to the server. Hence, the communication costs in FL can be significantly reduced.

**Summary Of The Review:**

This paper proposed a novel idea to reduce communication in FL. However, it seems non-trivial to make this algorithm to work with partial client participation. Without this extension, I feel the current algorithm is more suitable for classical distributed training than FL.

---

> ### Author Response · Authors · 2021-11-18
> **Response to Reviewer X6Uf (1 of 1)**
>
> Thank you for your comments and positive assessment. Our point-by-point responses are below.
>
> > **Comment: In general, I think the proposed ... may prevent people using this algorithm."**
>
> **Response:**
>
> - First, note that it is possible to tune the threshold value $\delta^{\mathsf{threshold}}$ of LBGM in Algorithm 1 based on the prioritization of communication savings vs. accuracy. Taking the example of Figure 5, CIFAR-10 uses $\delta^{\mathsf{threshold}} = 0.2$ and realizes a $14\\%$ drop in performance for a $79\\%$ gain in communication savings. If we reduce it to $\delta^{\mathsf{threshold}} = 0.05$, we obtain a performance drop of only $4\\%$ while still retaining a communication savings of $55\\%$. Similarly, for $\delta^{\mathsf{threshold}} = 0.01$, we get $22\\%$ communication savings for a negligible drop in accuracy (by only $0.1\\%$). We are in the process of clarifying this point in the results section. *We have included this discussion in Section 4 of the main paper under "LBGM as a Standalone Algorithm."*
>
> - Second, note that another interpretation of Figures 5-8 is the number of parameters that must be transferred to reach a certain accuracy. For example, in Figure 5, for an accuracy of $50\\%$, FL uses $120\times 10^6$ parameter transfers, while LBGM uses $85\times 10^6$ parameters, which indicates considerable communication savings. To further highlight the performance gain, *we have added the corresponding accuracy vs #params shared subplots for each of the figures (e.g., row 3 in Fig. 5, 6, 7, and 8 of main paper) in the updated paper.*
>
> ----
>
> > **Comment: The algorithm is more suitable ... benefits of using LGBM may diminish."**
>
> **Response:**
>
> - Our methodology has been developed based on several characteristics of federated learning that are reflected in the algorithm design and convergence analysis, including non-i.i.d. data distributions across clients.
> We would therefore hesitate to restate our contributions in terms of classical distributed learning. Nonetheless, we agree that partial device participation is an important practical consideration for federated learning. In fact, our algorithm is compatible with client sampling techniques as well, i.e., it can be applied on top of them to obtain significant performance gains. To demonstrate this, *we have added experiments for LBGM with client sampling for MNIST, FMNIST, CIFAR-10, and CelebA (Fig. 70; non-iid case and Fig. 71: iid case) in Appendix F.5 of the updated paper.* Our results with client sampling are consistent with the rest of our results for the case of full device participation. For example, our results on the MNIST dataset for 50\% client participation shows a 35\% and 55\% improvement in communication efficiency for only 0.2\% and 4\% drop in accuracy for the corresponding i.i.d and non-i.i.d cases (see column 1 of Fig. 71 and 70 respectively).
>
> - In particular, as long as an active client that joins in any step of the algorithm has a fairly good look-back gradient at the server (i.e., its newly generated gradients are close to the look-back gradient), there is no need for transmission of the entire parameter vector. This often happens in practice unless a client engages in the model training after a very long period of inactivity, in which case it would need to transmit its updated LBG before engaging in LBGM communication savings. *Our pseudocode used for device sampling has been added to Appendix D.2.*

---

> > ### Comment · Reviewer_X6Uf · 2021-11-29
> > **Thanks for the response**
> >
> > Thank the authors for the detailed response. While my first concern is addressed, I'm still not convinced that this algorithm can work well with client sampling. The newly added experiments uses a very high client sampling ratio, ie 50%. In practice, the client sampling ratio would be much smaller. For example, in the StackOverflow dataset in [1], the client sampling ratio is only 0.01%. If the authors cannot run experiments in such low client sampling ratio, then I think the authors at least are supposed to provide some discussions/experiments showing how the performance of the proposed algorithm changes along with the client sampling ratio. My main concern is that when the client sampling ratio is lower than some threshold, the proposed algorithm might not provide significant communication reduction.

---

> > > ### Author Response · Authors · 2021-11-29
> > > **Additional Follow-up**
> > >
> > > > **Comment: In practice, the client sampling ratio ... proposed algorithm might not provide significant communication reduction.**
> > >
> > > Thank you very much for the additional feedback. We have found that many recent papers in federated learning (FL) that consider client sampling employ ratios of 10% to 50% on systems with 10 to 1000 nodes [1-6]. Smaller ratios (e.g., 0.01%) may be required in FL systems that are significantly larger than the setup of 100 clients that we have considered. To address the reviewer’s concern, we have conducted another experiment on the MNIST dataset varying the sampling rate between 1% and 50%. Our results are summarized in the tables below:
> > >
> > > For **iid** data distribution:
> > >
> > >
> > > | Dataset | Sampling Rate | Comm. Saving | Accuracy Drop | $\delta^{\mathsf{threshold}}$ |
> > > | ------ | ----------- | ----------- | ----------- | ---------|
> > > | MNIST  | 50\% | 35.36\% | 0.17\%| 0.1 |
> > > | MNIST  | 30\% | 49.72\% | 0.56\%| 0.1 |
> > > | MNIST  | 10\% | 71.36\% | 6.10\%| 0.1 |
> > >
> > > For **non-iid** data distribution:
> > >
> > > | Dataset| Sampling Rate | Comm. Saving | Accuracy Drop | $\delta^{\mathsf{threshold}}$ |
> > > |------|-----------|-----------|-----------|------|
> > > | MNIST  | 50\% | 55.12\% | 0.42\%| 0.1 |
> > > | MNIST  | 30\% | 60.06\% | 1.93\%| 0.1 |
> > > | MNIST  | 10\% | 75.04\% | 7.00\%| 0.1 |
> > >
> > > Overall the results show that it is still possible to obtain communication savings as the sampling ratio decreases; the accuracy drops can be offset by decreasing  $\delta^{\mathsf{threshold}}$ (as we show in Fig. 6; also discussed in Sec. 4 of the main paper). We will complete these experiments for our other datasets as well, add the results to Appendix F.5, and reference them in Section 4 of the camera-ready paper (we are not permitted to upload a new version of the manuscript at this point in the rebuttal process). Additionally, we can consider a case with a higher number of workers (e.g., 1000 workers) and lower sampling ratios (e.g., 0.1\\%) for completeness if the reviewer suggests so.
> > >
> > > [1] Reddi et al. "Adaptive Federated Optimization." ICLR, 2021.
> > >
> > > [2] Wang et al. "Tackling the Objective Inconsistency Problem in Heterogeneous Federated Optimization." NeurIPS, 2020.
> > >
> > > [3] Lin et al. "Semi-decentralized federated learning with cooperative D2D local model aggregations." IEEE JSAC, 2021.
> > >
> > > [4] Li et al. "Fair resource allocation in federated learning". ICLR, 2020.
> > >
> > > [5] Cho et al. "Client Selection in Federated Learning: Convergence Analysis and Power-of-Choice Selection Strategies." ArXiv, 2021.
> > >
> > > [6] Fraboni et al. "On The Impact of Client Sampling on Federated Learning Convergence." ArXiv, 2021.

---

> > > > ### Comment · Reviewer_X6Uf · 2021-11-29
> > > > **Additional response**
> > > >
> > > > Thank the authors for the quick reply! The additional results look good to me and address my concern. But I'm wondering why the communication savings first increase and then decrease when the client sampling rate becomes smaller? Do the authors have any insights on this observation?

---

> > > > > ### Author Response · Authors · 2021-11-30
> > > > > **Additional Follow-up**
> > > > >
> > > > > Thank you for taking our responses into consideration.
> > > > >
> > > > > We apologize for the confusion here. On second thought, we feel it is best for us to remove the result at $1\\%$ from the above table: this translates to a single device participating in each round, and thus no averaging at the server, which requires separate experimental consideration. Over the range $10-50\\%$, we can see the trend that as the sampling ratio is reduced, the corresponding accuracy drop and communication-saving increase for a given LBGM threshold $\delta^{threshold}$. We can further improve on the accuracy drop by tuning the threshold value as discussed in Sec. 4 and Fig. 6 of the main paper: for example, in the $10\\%$ case, if we reduce the threshold to $\delta^{threshold} = 0.01$, then the accuracy drop is only $0.53\\%$ for a communication savings of $32.40\\%$.

---

### Official Review · Reviewer_X8h7 · 2021-11-03

**Correctness:** 3
**Technical Novelty And Significance:** 2
**Empirical Novelty And Significance:** 2
**Recommendation:** 5
**Confidence:** 4

**Main Review:**

**Novelty:** The idea of model updates in a low-rank subspace is not new. In fact, it has been mentioned in one of the pioneering papers on federated learning:
- Konecny et al., Federated learning: Strategies for improving communication efficiency, 2016 (Section 2)

The core idea of low-rank updates in this paper is the same as the low-rank structured update in the above paper by Konecny et al. While the convergence analysis in this paper may be a new result, the proof technique is relatively standard and is similar to other works on delayed/compressed SGD. In general, it is not surprising to achieve a convergence result comparable to existing algorithms under the condition of $\Delta^2 \\leq \\eta$ where $\\eta$ is inversely proportional to $1/\\sqrt{T}$ in Corollary 1. Furthermore, I feel that the current Algorithm 1 may be too specific in the sense that it only allows sending a scalar LBC that is related to the angular difference. The algorithm would be more flexible if an arbitrary (but given) number of scalars is allowed to be transmitted, where the number of scalars depends on the dimension of the constrained subspace of updates.

It is also worth mentioning that similar observations of low-rank gradient subspaces have been made and leveraged in continual learning literature:
- Chaudhry et al., Continual Learning in Low-rank Orthogonal Subspaces, NeurIPS 2020.
- Saha et al., Gradient Projection Memory for Continual Learning, ICLR 2021.

**Practical feasibility:** It is difficult for the proposed algorithm to work in large-scale federated learning systems. The algorithm needs to store copies of look-back gradients (LBGs) for all the clients at the server, which would consume a huge amount of storage in realistic on-device federated learning systems with over millions of clients. Furthermore, in such large-scale systems, it is only feasible for only a small percentage of clients to participate in each federated learning round, i.e., random client sampling needs to be performed. This paper does not seem to consider client sampling/selection at all, not in the algorithm description, analysis, nor in the experiments.

**Experiments:** It is not clear how the hyperparameters were found in the experiments, such as the threshold for LBGM and the value of K for top-K. This is important since the performance of each method could largely depend on the choice of these critical hyperparameters, so the hyperparameters should be found methodologically (e.g., using grid search) instead of chosen arbitrarily. The comparison is only fair if the hyperparameters for each method has been optimized for the method itself. In addition, it is a bit difficult to align the communication savings with the accuracy curves. It would help if figures showing accuracy vs. number of bytes transmitted could be added.

**Minor:**
- It seems that the spacing has been squeezed quite a lot to fit the paper into 9 pages. This should be avoided. For example, some detailed discussion on related work may be moved to the beginning of the appendix.
- In the related work section, Gradient Compression paragraph, there is a sentence saying "These works were originally proposed for inter-GPU communication speedups and not FL." This is incorrect. There has been many papers that discuss gradient compression in the context of federated learning. For example:
  - Haddadpour et al., Federated Learning with Compression: Unified Analysis and Sharp Guarantees, AISTATS 2021.
  - Albasyoni et al., Optimal Gradient Compression for Distributed and Federated Learning, 2020.


**Summary Of The Paper:**

The paper presents an efficient federated learning method which leverages the empirical observation that the gradients used in model updates are usually in a low-rank subspace. Based on this observation, a "look-back gradient multiplier" (LBGM) method is proposed that only updates a scalar look-back coefficient (LBC) as long as the angular difference between the stored gradient and the latest gradient is within a given threshold. A theoretical analysis of this method is given in the paper and the performance is also verified in various experiments.

**Summary Of The Review:**

While communication efficiency is an important aspect in federated learning, this paper appears to have limited novelty and also some issues regarding practical feasibility and experiments.

---

> ### Author Response · Authors · 2021-11-18
> **Response to Reviewer X8h7 (3 of 3)**
>
> > **Comment: It seems that the spacing ... moved to the beginning of the appendix."**
>
> **Response:** While making the other requested additions, we have gone through and condensed descriptions throughout the paper wherever possible, increasing the spacing. Many of the additional experimental results have been deferred to the appendix for this same reason.
>
> ----------
>
> > **Comment: In the related work section, Gradient Compression ... and Federated Learning, 2020."**
>
> **Response:** In the related work, we have removed this statement to avoid confusion. Additionally, we will add reference to [9, 10] for completeness.
>
> [9]  Haddadpour et al., Federated Learning with Compression: Unified Analysis and Sharp Guarantees, AISTATS 2021.
>
> [10] Albasyoni et al., Optimal Gradient Compression for Distributed and Federated Learning, 2020.
>
> ----------
>
> > **Comment: While communication efficiency is ... limited novelty and also some issues regarding practical feasibility and experiments.**
>
> **Response:** Overall, this paper is the first in the literature that proposes reusing/recycling the previously generated gradients during model training to represent the newly generated gradients in federated learning and in the context of distributed machine learning more generally. This opens the door to a new domain of algorithms that explore gradient recycling for optimization communication vs. accuracy trade-off. *We have clarified these points in our contributions in Sec. 1 as well as in our conclusions in Sec. 6.*

---

> ### Author Response · Authors · 2021-11-18
> **Response to Reviewer X8h7 (2 of 3)**
>
> > **Comment: Practical feasibility: It is ... nor in the experiments."**
>
> **Response:**
>
> - While our standalone LBGM method may require more server-side storage space compared with conventional federated learning, this comes with a significant benefit of energy savings for the participant devices, which is a critical area of focus in research on heterogeneous federated learning systems. When storage space on a single server becomes an issue, LBGM can be applied in conjunction with a number of existing techniques for reducing storage burden, e.g., model compression and/or storage offloading between nodes. In particular, large-scale federated learning systems often exhibit a hierarchical structure of node connectivity between the network edge and cloud, which has given rise to techniques for hierarchical federated learning [4].
> In a hierarchical federated learning framework, LBG storage can be distributed across the network hierarchy (e.g., at edge servers, cellular base stations, etc.), which eliminates the burden on the centralized cloud server. Also, using compression techniques such as parameter clustering [5, 6], the LBGs can be clustered at the server to reduce the overall storage requirements. *We have included these discussions in Appendix C.1 of the updated paper.*
>
> - We have not considered client sampling/selection in our work as such techniques can similarly be viewed as orthogonal to LBGM, i.e., our algorithm can be applied in conjunction with partial device participation. Several other recent federated learning works [7,8] have also assumed full device participation for a similar reason, i.e., client sampling does not affect their algorithm directly. To address any concerns here, *we have added experiments for LBGM with client sampling for MNIST, FMNIST, CIFAR-10, and CelebA (Fig. 70; non-iid case and Fig. 71: iid case) in Appendix F.5 of the updated paper (pseudocode in Appendix D.2).* Our results with client sampling are consistent with the rest of our results for the case of full device participation. For example, the MNIST dataset for 50\% client participation shows a 35\% and 55\% improvement in communication efficiency for only 0.2\% and 4\% drop in accuracy for the corresponding i.i.d and non-i.i.d cases (see column 1 of Fig. 71 and 70 respectively).
>
> [4] Hosseinalipour et al. "Multi-Stage Hybrid Federated Learning over Large-Scale D2D-Enabled Fog Networks.", 2020.
>
> [5] Cho et al. "DKM: Differentiable K-Means Clustering Layer for Neural Network Compression." 2021.
>
> [6] Son et al. "Clustering convolutional kernels to compress deep neural networks." ECCV 2018.
>
> [7] Vogels et al. "PowerSGD: Practical low-rank gradient compression for distributed optimization." NeurIPS 2019.
>
> [8] Lin et al. "Semi-decentralized federated learning with cooperative D2D local model aggregations." JSAC 2021.
>
> -----------
>
> > **Comment: Experiments: It is not clear how the hyperparameters ...  transmitted could be added."**
>
> **Response:**
>
> - **Regarding the hyperparameter search:** Most of the compression/communication savings baselines operate on a tradeoff between communication savings and accuracy. Our process for hyperparameter selection first optimizes the hyperparameters for the base algorithm such that we achieve the best possible communication saving subject to constraint that accuracy does not fall off much below the corresponding vanilla federated learning approach. For example, we optimize the value of $K$ by changing it in orders of 10, i.e. $K=10\\%$, $K=1\\%$, $K=0.1\\%$, etc. and choose the value that gives the best tradeoff between the final model accuracy and communication savings (this value is generally around $K=10\\%$). Similarly, for ATOMO we consider rank-1, rank-2, and rank-3 approximations. While rank-1 approximation gives better communication savings, the corresponding accuracy falls off sharply. Rank-3 approximation gives only a marginal accuracy benefit over rank-2 approximation but adds considerably more communication burden. Thus we use rank-2 approximations in ATOMO. In the plug-and-play evaluations, the LBGM algorithm is applied on top of the base algorithms once their hyperparameters are tuned as a final step to show the additional benefits we can attain by exploiting the low-rank characteristic of the gradient subspace. Our choice hyperparameters can be found on GitHub (https://github.com/PairML/LBGM), which is cited in the paper. We can also directly include the hyperparameters in the appendix if the reviewers suggest so. *We have added the discussion on the process of hyperparameter tuning in Appendix C.2 of the revised manuscript.*
>
> - **Regarding the accuracy vs. the number of bytes transferred:** We agree that these plots can provide the reader with a clear understanding of the tradeoff between accuracy and communication savings. *We have added the corresponding accuracy vs #params shared subplots for each of the figures (e.g., row 3 in Figs. 5-8) in the revised manuscript.*

---

> ### Author Response · Authors · 2021-11-18
> **Response to Reviewer X8h7 (1 of 3)**
>
> Thank you for taking the time to read our paper and provide us with comments. Our point-by-point responses are below.
>
> > **Comment: "Novelty: The idea of ... constrained subspace of updates"**
>
> **Response:**
>
> - **Regarding the paper [1] mentioned by the reviewer:** We agree that the idea of updating models within low-rank subspaces has already been introduced in machine learning. However, there is a major difference between our work and the way low-rank is exploited in the work mentioned here. Konency et al. [1] perform a low rank approximation of a given update (weight or gradient), i.e., during an aggregation step $t$, if the device needs to transmit a parameter matrix $\boldsymbol{M}\_t$, we can instead send its low rank approximation $\boldsymbol{M}\_t^\star = \arg \min\_{\hat{\boldsymbol{M}}} \Vert\boldsymbol{M}\_t-\hat{\boldsymbol{M}}\Vert\_F$, s.t., $\mathsf{rank}(\hat{\boldsymbol{M}}) \leq r$.
> By contrast, we consider the rank of the subspace
> spanning the *series of updates* through the model training rounds $1,...,T$. In particular, we observe that if we were to create a matrix of updates stacked from several update rounds, say $\boldsymbol{M}\_1,\boldsymbol{M}\_2,...,\boldsymbol{M}\_T$, then this *matrix of stacked/consecutive updates is low-rank* (equivalently, the updates are linearly dependent). This leads to our technique of recycling the prior gradients to represent the newly generated ones over time, which we have not seen in the literature. *We have included these points in the related works section of the updated paper.*
>
> - **Regarding the flexibility of the algorithm:** Sending multiple scalars instead of a single scalar would require storing a larger number of corresponding look-back gradients (LBGs) at the server and at the devices. This would present scalability challenges in practice, especially as the size of the model and the federated learning system increase; limiting *device-side* computation and storage complexity is critical in the presence of resource heterogeneity.
> As a result, we focus on the single scalar case in this work, which we find obtains significant improvements in communication overhead with only small reductions in model performance compared to vanilla FL. In the absence of such constraints, we agree that adding more LBCs in uplink transmissions could improve the performance of our algorithm even further, since the locally generated gradients would be recovered at the server with lower reconstruction error.
>
> [1] Konecny et al., Federated learning: Strategies for improving communication efficiency, 2016.
>
> ----------
>
> > **Comment: "It is also worth mentioning ... Continual Learning, ICLR 2021"**
>
> **Response:** Thank you for pointing out these references. These two works [2,3] make an observation about rank partitioning of the gradient subspace. In both of the works, the gradient space partitioning is realized by analyzing the gradients with respect to the inputs (i.e., data samples) for the *final trained model*. Our work in comparison is considering the gradient subspace with respect to gradient updates *across the SGD rounds*. In particular, Saha et al. [3] consider the low rank approximation of the matrix consisting of gradients calculated for a random subset of training data at the final trained model checkpoint (refer to Section 5 of [3]). The orthogonal projection technique in Chaudhry et al. [2] also partitions the gradient space of the final model.
> In the context of continual learning, this helps ensure that upon completion of a learning task, the subsequent updates from new task steps do not interfere in a gradient direction which is sensitive/critical to the performance of the prior learned tasks. Our intuitions/insights about the low rank characteristics of SGD optimization are considerably different, since we consider the gradient directions taken during the SGD optimization progress and explore the rank of the consecutive  gradients generated during the training process. *We have added this discussion about these papers into the related work section of the updated manuscript.*
>
> [2] Chaudhry et al., Continual Learning in Low-rank Orthogonal Subspaces, NeurIPS 2020.
>
> [3] Saha et al., Gradient Projection Memory for Continual Learning, ICLR 2021.

---

> ### Comment · Reviewer_X8h7 · 2021-11-20
> **Thank you for your response**
>
> Thanks for the authors' response. I would consider using a subspace that spans a series of updates to be a relatively straightforward extension of approximating a single update. In addition, storing all clients' gradients at the server is a key limitation of this work. Although it can be argued that such a method may be possible in certain scenarios, considering that future cross-device federated learning applications can have over billions of devices, it is quite unlikely that a service provider would consume such a large amount of storage for a marginal performance improvement (most of the accuracy vs # parameters plots in Figure 7 only show a marginal improvement). For these reasons, I would keep my current score, but if another reviewer strongly feels that this paper should be accepted, I would not argue against it either.
>
> It would be useful if the methodology used for hyperparameter search (at the end of the second response) could be included in the appendix.
>
> An additional minor comment is: there are fluctuations in many of the plots, even for vanilla FL in Figures 5 and 6. It would be good to investigate and explain why these fluctuations happen. In many cases, this may be due to improper choice of learning rate or model architecture, which should be fixed if possible.

---

> > ### Author Response · Authors · 2021-11-20
> > **Additional Followup (1 of 2)**
> >
> > Thank you for taking the time to consider our responses. We appreciate the constructive feedback and would request the reviewer to reconsider the following:
> >
> > > **Comment:**  I would consider using a subspace that spans a series of updates to be a relatively straightforward extension of approximating a single update.
> >
> > - Consider a set of rank-1 matrices as follows:
> >
> > \begin{align}
> >     \boldsymbol{A} = \begin{bmatrix}1 & 2\\\\ 1 & 2\end{bmatrix} = \begin{bmatrix}1\\\\ 1\end{bmatrix}\begin{bmatrix}1 \\\\ 2\end{bmatrix}^\top, &\quad
> >     \boldsymbol{B} = \begin{bmatrix}3 & 5\\\\ 6 & 10\end{bmatrix} = \begin{bmatrix}1\\\\ 2\end{bmatrix}\begin{bmatrix}3 \\\\ 5\end{bmatrix}^\top\\
> >     \boldsymbol{C} = \begin{bmatrix}3 & 4\\\\ 6 & 8\end{bmatrix} = \begin{bmatrix}1\\\\ 2\end{bmatrix}\begin{bmatrix}3 \\\\ 4\end{bmatrix}^\top, &\quad
> >     \boldsymbol{D} = \begin{bmatrix}3 & 2\\\\ 12 & 8\end{bmatrix} = \begin{bmatrix}1\\\\ 4\end{bmatrix}\begin{bmatrix}3 \\\\ 2\end{bmatrix}^\top
> > \end{align}
> >
> > If we next consider the auxiliary matrix $\boldsymbol{M}$ obtained by stacking the vectorized form of these low rank matrices, given by,
> > \begin{equation}
> >     \boldsymbol{M} = \begin{bmatrix} \vert & \vert & \vert & \vert\\\\
> >         \mathsf{vec}(\boldsymbol{A}) & \mathsf{vec}(\boldsymbol{B}) & \mathsf{vec}(\boldsymbol{C}) & \mathsf{vec}(\boldsymbol{D}) \\\\
> >         \vert & \vert & \vert & \vert
> >     \end{bmatrix} = \begin{bmatrix} 1 & 3 & 3 & 3\\\\
> >         2 & 5 & 4 & 2 \\\\
> >         1 & 6 & 6 & 12 \\\\
> >         2 & 10 & 8 & 8
> >     \end{bmatrix},
> > \end{equation}
> > this matrix is rank-4. This is a low-dimensional example where we see that while individual updates may have low rank, the subspace spanned by these is not low-rank trivially. This phenomenon would be even more apparent in higher dimensional spaces such as gradient updates during deep neural network training. It is therefore not straightforward to consider the subspace spanned by a series of updates to be an extension of approximating a single update.
> >
> > > **Comment: ... most of the accuracy vs # parameters plots in Figure 7 only show a marginal improvement**
> >
> > - The main benefit that we wish to convey in Fig. 7 is the communication savings we can achieve on top existing state-of-the-art compression. One way to interpret those figures is to consider the communication savings obtained while reaching a fixed accuracy, which we have now added as the third row in each of our plots. As demonstrated in all of these rows, our method outperforms the baselines in terms of communication savings all cases. In addition to this, below we list the percent communication savings along with with their corresponding accuracy loss and LBGM thresholds:
> >
> > | Dataset      | Comm. Saving | Accuracy Drop | $\delta^{\mathsf{threshold}}$ |
> > |-|-|-|-|
> > | MNIST (Top-K)    | 44.28\% | 0.25\% | 0.4 |
> > | FMNIST (Top-K)   | 41.78\% | 0.01\% | 0.4 |
> > | CIFAR-10 (Top-K) | 70.53\% | 14.11\%| 0.2 |
> > | CelebA (Top-K)   | 51.32\% | 0.01\% | 0.4 |
> > | MNIST (ATOMO)    | 54.65\% | 8.30\% | 0.4 |
> > | FMNIST (ATOMO)   | 42.48\% | 5.52\% | 0.4 |
> > | CIFAR-10 (ATOMO) | 56.01\% | 10.76\%| 0.2 |
> > | CelebA (ATOMO)   | 33.70\% | 0.001\%| 0.4 |
> >
> > In most cases, the communication savings is high for a marginal drop in performance. If the accuracy drops on CIFAR-10 are of concern, we can further tune the $\delta^{\mathsf{threshold}}$ to increase the accuracy. Specifically, for the setting in Fig. 5 of CIFAR-10, we obtain the following:
> >
> > | Dataset      | Comm. Saving | Accuracy Drop | $\delta^{\mathsf{threshold}}$ |
> > | -|-|-|-|
> > | CIFAR-10 | 79.43\% | 14.12\%| 0.2 |
> > | CIFAR-10 | 55.29\% | 4.94\%| 0.05 |
> > | CIFAR-10 | 22.16\% | 0.01\%| 0.01 |
> >
> > This, we can obtain 22\% communication savings for negligible accuracy drop.
> >
> > > **Comment: ...considering that future cross-device federated learning applications can have over billions of devices, it is quite unlikely that a service provider would consume such a large amount of storage...**
> >
> > - We would like to reiterate that the purpose of this work is to develop a novel foundational technique using the low-rank hypothesis, which gives rise to several possible extensions for future work. In a billion scale federated learning network, it is realistic to assume that a network hierarchy (e.g., end device-base station-edge server-cloud server) [1] exists which can be leveraged to offload the storage burden across the network. Also, if LBGM is applied on top of existing techniques, it shrinks the size of LBGs to be stored. Moreover, at such a scale it is unrealistic to assume that all billion clients have very different LBGs given the low-rank hypothesis. It should be in fact possible to cluster the LBGs into a smaller number of centroids instead of saving all the individual LBGs from the billion clients. *We have included these discussions in the Appendix C.1 of the updated paper.*
> >
> > [1] Hosseinalipour et al. "Multi-Stage Hybrid Federated Learning over Large-Scale D2D-Enabled Fog Networks.", 2020.

---

> > > ### Author Response · Authors · 2021-11-24
> > > **Additional Followup (2 of 2)**
> > >
> > > > **Comment: An additional minor comment is: there are fluctuations...fixed if possible.**
> > >
> > > As mentioned by the reviewer, the fluctuations seen are mainly caused by a higher learning rate. To alleviate this concern, in the revised manuscript, we have adjusted the learning rate in several subplots to minimize the fluctuation associated with SGD optimization.

---

### Author Response · Authors · 2021-11-19
**Summary of Changes**

Dear Reviewers,

Thank you for taking the time to read through our manuscript and provide detailed feedback. Our point-by-point responses can be found below, with pointers to locations where they have been addressed (or will shortly be addressed) in the revised paper.

The manuscript has been updated with all of the suggested changes:
- additional experiments on device sampling are added to Appendix F.5 (which are consistent with our original claims),
- additional experiments on other datasets/models added to Appendix F.1 and F.2,
- the missing unit vector issue in the convergence analysis proofs (which have not impacted our results) in Appendix A,
- a third row of plots for each experiment which gives the corresponding accuracy vs. communication requirement,
- refinements of our claims in Sec. 1,
- modifications and additions to the related work discussion in Sec. 5,
- additional discussion of results in Sec. 4 and pointers to new results in the appendix, and
- miscellaneous comments from reviewers.

More details are available in the point-by-point responses. Please let us know if you have any additional questions.

Sincerely,

Authors

---

### Decision · Program_Chairs · 2022-01-20

**Decision:**

Accept (Poster)

**Comment:**

The paper shows that most variance of gradients used in FL and distributed learning in general is in very low rank subspaces, an observation also made in Konecny et al 2016 and some other related works in deep learning, though sometimes for a different purpose.
The paper then proposes lightweight updates combining a fresh gradient with old updates. Experiments and a theoretical convergence guarantee complement the results, which are mostly convincing.

The experiments compare against ATOMO but strangely not against the more common PowerSGD, which would also work with partial client participation.

Overall, reviewers all agreed that the paper is interesting, well-motivated and deserves acceptance.
We hope the authors will incorporate the open points as mentioned by the reviewers.